# Online Coresets for Parametric and Non-Parametric Bregman Clustering

**Rachit Chhaya**                                                                          *rachit_chhaya@daiict.ac.in*
*DAIICT- Gandhinagar, India*

**Jayesh Choudhari**                                                          *choudhari.jayesh@alumni.iitgn.ac.in*
*CUBE, England*

**Anirban Dasgupta**                                                                       *anirbandg@iitgn.ac.in*
*IIT Gandhinagar, India*

**Supratim Shit**[*]                                                                       *supratim.shit@gmail.com*
*Technion, Israel*

**Reviewed on OpenReview:** *Review*

## Abstract

We present algorithms that create coresets in an online setting for clustering problems based on a wide subset of Bregman divergences. Notably, our coresets have a small additive error, similar in magnitude to the gap between expected and empirical loss Bachem et al. (2017a), and take update time $O(d)$ for every incoming point where $d$ is the dimension of the point. Our first algorithm gives online coresets of size $\tilde{O}(\text{poly}(k, d, \epsilon, \mu))$ for $k$-clusterings according to any $\mu$-similar Bregman divergence. We further extend this algorithm to show the existence of non-parametric coresets, where the coreset size is independent of $k$, the number of clusters, for the same subclass of Bregman divergences. Our non-parametric coresets also function as coresets for non-parametric versions of the Bregman clustering like DP-Means. While these coresets provide additive error guarantees, they are significantly smaller for high dimensional data than the (relative-error) coresets obtained in Bachem et al. (2015) for DP-Means— for the input of size $n$ our coresets grow as $O(\log n)$ while being independent of $d$ as opposed to $O(d^d)$ for points in $\mathbb{R}^d$ Bachem et al. (2015). We also present experiments to compare the performance of our algorithms with other sampling techniques.

## 1 Introduction

Clustering is a frequently used operation in data processing. A canonical definition of the clustering problem is via the $k$-median, in which $k$ possible centers need to be proposed such that the sum of distances of every point to its closest center is minimized. There has been a plethora of work, both theoretical and practical, devoted to finding efficient and provable clustering algorithms in this $k$-median setting. Most of this literature is devoted towards dissimilarity measures that are algorithmically easier to handle e.g. the various $\ell_p$ norms, especially Euclidean.

A mathematically elegant family of dissimilarity measures that have found wide use in statistics are the *Bregman divergences* which include the squared Euclidean distance, the Mahalanobis distance, Kullbeck-Leibler divergence, Itakuro-Saito dissimilarity and many others. While being mathematically satisfying, the chief drawback of working with Bregman divergences for clustering is algorithmic— most of these divergences do not satisfy either symmetry or triangle inequality conditions. Hence, developing efficient clustering algorithms for these divergences has been a much harder problem to tackle. Banerjee et al. (2005) did a

---

[*]Corresponding author

systematic study of the $k$-median clustering problem under Bregman divergences, and proposed algorithms that are generalizations of Lloyd's iterative algorithm for the Euclidean $k$-means problem. However, scalability remains a major issue. Given that there are no theoretical bounds on the quality of the solution obtained via the Lloyd's algorithm in the general Bregman setting, a decent solution is often achieved via running enough iterations as well as by searching over multiple initializations. This is clearly expensive when the number of data points is large. This problem is further aggravated when the dimension of the input points is also high and the number of clusters is not known.

Coresets, small summaries of data to enable efficient optimization, have been successfully used in many problems in computational geometry and more recently in machine learning. A coreset is a judiciously selected (and reweighted) set of points, often from the input points themselves, such that solving the optimization problem on the coreset gives a guaranteed approximation to the solution of the optimization problem on the full data (Lucic et al., 2016).

In this work we explore two specific goals in creating coresets for Bregman divergence based clustering. First, we wish to create the coresets in an online setting, i.e. the decision about each point should be taken when the point is first consumed by the algorithm from an online stream. Secondly, we show the existence of a single coreset that works for all values of $k$, the number of clusters. We further give an algorithmic version of it under certain assumption. It is not apriori clear that either of these goals are achievable. Coreset creation strategies, e.g. (Lucic et al., 2016), often require a rough approximation in order to construct the importance sampling distribution. This route would seem to preclude taking online decisions.

Yet another issue is the dependence of the coreset size on the number of clusters— $k$, the number of clusters can be large, and more importantly, it can be unknown, to be determined only after exploratory analysis with clustering. When the number of clusters is unknown, even the existence of a coreset of sublinear size is unclear. Recent work by Huang & Vishnoi (2020) shows that for relative error coresets for Euclidean $k$-means, a linear dependence of coreset size on $k$ is both sufficient and inevitable.

In this work, we tackle these questions for Bregman divergences. We develop coresets with *small additive error guarantees*. Such results have also been obtained in the Euclidean setting by Bachem et al. (2018a), and in the online subspace embedding setting by Cohen et al. (2016). We next show that in the case of non-parametric clustering, there exists a coreset whose size is independent of $k$, the parameter representing number of cluster centers. We utilize the *sensitivity* framework of Feldman & Langberg (2011) jointly with the barrier functions method of Batson et al. (2012) in order to achieve this. Using an empirical notion of sensitivity (Baykal et al., 2018) we present an algorithmic version of this result under certain assumptions. To the best of our knowledge this is the first non-parametric coreset for clustering. A non-parametric coreset will be useful in problems such as DP-Means clustering (Bachem et al., 2015) and extreme clustering (Kobren et al., 2017), where number of clusters may not be known apriori. We now formally describe the setup and list our contributions.

Given $\mathbf{A} \in \mathbb{R}^{n \times d}$ where rows are input points in $\mathbb{R}^d$. Let $\varphi$ be the mean of $\mathbf{A}$, i.e., $\varphi = \sum_{i \leq n} (\mathbf{a}_i / n)$. For a query $\mathbf{Q} \subset \mathbb{R}^d$ the clustering cost on $\mathbf{A}$ with respect to $\mathbf{Q}$ is defined as,

$$f_{\mathbf{Q}}(\mathbf{A}) = \sum_{i \leq n} \min_{\mathbf{q} \in \mathbf{Q}} f_{\mathbf{q}}(\mathbf{a}_i).$$

where $f_{\mathbf{q}}(\mathbf{a}_i)$ is a chosen $\mu$-similar Bregman divergence (Definition 2.2). We present algorithms which return coreset $\mathbf{C}$ and set of corresponding weights $\Omega$ such that, $\forall \mathbf{X} \subset \mathbb{R}^d$, $|\mathbf{X}| = k$,

$$|f_{\mathbf{X}}(\mathbf{C}, \Omega) - f_{\mathbf{X}}(\mathbf{A})| \leq \epsilon(f_{\mathbf{X}}(\mathbf{A}) + f_{\varphi}(\mathbf{A})) \tag{1}$$

where, $f_{\mathbf{X}}(\mathbf{C}, \Omega)$ is the weighted cost on points $\mathbf{C}$, i.e., $f_{\mathbf{X}}(\mathbf{C}, \Omega) = \sum_{\mathbf{c}_i \in \mathbf{C}} \omega_i f_{\mathbf{X}}(\mathbf{c}_i)$. The weight $\omega_i \in \Omega$ corresponds to the point $\mathbf{c}_i \in \mathbf{C}$. For points coming in streaming fashion we use $\mathbf{A}_i \in \mathbb{R}^{i \times d}$ to represent the first $i$ points that have arrived. Let $\varphi_i$ denote the mean point of $\mathbf{A}_i$, i.e., $\varphi_i = \sum_{j \leq i} (\mathbf{a}_j / i)$. Notice that the additive term $f_{\varphi}(\mathbf{A})$ can be understood in the following manner—when $f(\cdot)$ is the squared Euclidean metric, then $f_{\varphi}(\mathbf{A}) = \sum_i \|\mathbf{a}_i - \varphi\|_2^2 = n \times \text{avg}_{i,j} \|\mathbf{a}_i - \mathbf{a}_j\|_2^2$.

Our main contributions are as follows,

- We present an online algorithm called `BregmanFilter` (**Algorithm (1)**) which ensures property equation 1 for any $\mathbf{X} \in \mathbb{R}^{k \times d}$ with at least 0.99 probability. `BregmanFilter` takes $O(d)$ both in update time and working space to return a coreset $(\mathbf{C}, \Omega)$ for $\mathbf{A}$. The expected size of the coreset is $O\left(\frac{dk \log k}{\epsilon^2 \mu^2}\left(\log n + \log\left(f_\varphi(\mathbf{A})\right) - \log\left(f_{\varphi_2}(\mathbf{a}_2)\right)\right)\right)$ (Theorem 4.6). For the special case of $k$-means clustering, this implies a online coreset of size $O\left(\frac{dk \log k}{\epsilon^2}\left(\log n + \log\left(f_\varphi(\mathbf{A})\right) - \log\left(f_{\varphi_2}(\mathbf{a}_2)\right)\right)\right)$ (Corollary 4.2) .

- In a non-parametric clustering problem, the number of clusters are unknown. We first show the existence of coreset for non-parametric clustering based on Bregman divergence. Under a mild assumption on the data we also present an algorithmic version, `NonParametricFilter` (**Algorithm (2)**), which creates a coreset $(\mathbf{C}, \Omega)$ based on importance sampling for clustering that ensures equation 1, $\forall \mathbf{X} \in \mathbb{R}^{i \times d}$ where $1 \leq i \leq n$. The coreset has $O\left(\frac{1}{\epsilon^2 \mu^2}\left(\log n + \log\left(f_\varphi(\mathbf{A})\right) - \log\left(f_{\varphi_2}(\mathbf{a}_2)\right)\right)\right)$ expected points (Theorem 5.1). This coreset can be used for DP-Means clustering (Theorem 5.12). For $d \log(d) > \log(\log(n))$, the expected coreset size is smaller than $O(d^d k^* \epsilon^{-2})$, the current best known coreset size for DP-means obtained by Bachem et al. (2015), where $k^*$ is the optimal number of centers for DP-Means clustering. Further for the special case of $k$-means clustering, the non-parametric coreset $(\mathbf{C}, \Omega)$ for $\mathbf{A}$ will have an expected size of $O\left(\frac{1}{\epsilon^2}\left(\log n + \log\left(f_\varphi(\mathbf{A})\right) - \log\left(f_{\varphi_2}(\mathbf{a}_2)\right)\right)\right)$ (Corollary 5.2).

- We present experimental results and compare the performance of our coresets with other known coreset building techniques. The comparison is done on real-world datasets to support our theoretical claims.

For $\mathbf{A}_i$, the `BregmanFilter` maintains an online coreset $\mathbf{C}_i$ with corresponding weights $\Omega_i$. The coreset is online in the sense that for every incoming point $\mathbf{a}_i$ the algorithm either samples or discards the point before getting the next point. With this we can also ensure an online guarantee, i.e., $|f_\mathbf{X}(\mathbf{C}_i) - f_\mathbf{X}(\mathbf{A}_i)| \leq \epsilon(f_\mathbf{X}(\mathbf{A}_i) + f_{\varphi_i}(\mathbf{A}_i))$, with constant probability $\forall i \in [n]$, by taking a union bound over all $i$. Note that this is a stronger guarantee and in this case the expected sample size has an excess multiplicative factor of $O(\log n)$.

**Outline:** In section 2 we present all the notations and definitions that we use in the rest of the paper. In section 3 we discuss the previous works related to the results in this paper. In section 4 we present our first result, which is an online algorithm for building coresets clustering based on $\mu$-similar Bregman divergences. In the next section 5 we discuss coreset results for non-parametric clustering for same class of divergences. In section 6 we show a bound on the uniform deviation for the same class of divergences. Finally we present some experimental results in section 7 on real datasets.

## 2   Preliminary

Here we define the notation and the common terms that we use in rest of the paper. The set of the first $n$ natural number is represented by $[n]$. A bold lower case letter denotes a vector or a point for e.g. $\mathbf{a}$, and a bold upper case letter denotes a matrix or set of points as defined by the context for e.g. $\mathbf{A}$. Unless it is stated otherwise, the matrix $\mathbf{A}$ is used to represent $n$ points each in $\mathbb{R}^d$. $\mathbf{a}_i$ denotes the $i^{th}$ row of matrix $\mathbf{A}$ and $\mathbf{a}^j$ denotes its $j^{th}$ column. We use the notation $\mathbf{A}_i$ to denote the matrix or a set, formed by the first $i$ rows or points of $\mathbf{A}$ seen till a time in the streaming setting. Given $\mathbf{A}$, the smallest and the largest absolute values are defined as $\|\mathbf{A}\|_{\min} = \min_{i,j} |a_{i,j}|$ and $\|\mathbf{A}\|_{\max} = \max_{i,j} |a_{i,j}|$.

In a clustering problem, depending on the type of the input points a function is used from a wide range of divergence measure called *Bregman divergence.*

**Definition 2.1. *Bregman divergence:*** *For any strictly convex, differentiable function $\Phi : \mathcal{Z} \to \mathbb{R}$, the Bregman divergence with respect to $\Phi$, $\forall \mathbf{x}, \mathbf{y} \in \mathcal{Z}$ is,*

$$d_\Phi(\mathbf{y}, \mathbf{x}) = \Phi(\mathbf{y}) - \Phi(\mathbf{x}) - \nabla\Phi(\mathbf{x})^T(\mathbf{y} - \mathbf{x})$$

We also denote $f_{\mathbf{x}}(\mathbf{y}) = d_\Phi(\mathbf{y}, \mathbf{x})$. Throughout the paper for some set of centers $\mathbf{X}$ in $\mathbb{R}^d$ and point $\mathbf{a} \in \mathbb{R}^d$ we consider $f_{\mathbf{X}}(\mathbf{a})$ as a cost function based on Bregman divergence. Such $\mathbf{X}$ are also called query set. We define it as $f_{\mathbf{X}}(\mathbf{a}) = \min_{\mathbf{x} \in \mathbf{X}} f_{\mathbf{x}}(\mathbf{a}) = \min_{\mathbf{x} \in \mathbf{X}} d_\Phi(\mathbf{a}, \mathbf{x})$, where $d_\Phi(\cdot)$ is some Bregman divergence as defined above. If the set of points in $\mathbf{A}$ have corresponding weights $\{w_a\}$ then $\forall \mathbf{a} \in \mathbf{A}$ we define $f_{\mathbf{x}}(\mathbf{a}) = w_{\mathbf{a}} d_\Phi(\mathbf{a}, \mathbf{x})$.

Unlike squared euclidean distance, not all Bregman divergences follow metric properties. However there is a wide sub class called $\mu$-similar Bregman divergence which can relate to distance measure that follows metric properties.

**Definition 2.2.** *A Bregman divergence $d_\Phi$ on domain $\mathcal{Z}$ is called a $\mu$-similar Bregman divergence for some $\mu > 0$ iff there exists a positive definite matrix $\mathbf{M}$ such that, for each $\mathbf{x}, \mathbf{y} \in \mathcal{Z}$*

$$\mu d_{\mathbf{M}}(\mathbf{y}, \mathbf{x}) \le d_\Phi(\mathbf{y}, \mathbf{x}) \le d_{\mathbf{M}}(\mathbf{y}, \mathbf{x})$$

*where $d_{\mathbf{M}}(\mathbf{y}, \mathbf{x}) = (\mathbf{y} - \mathbf{x})^T \mathbf{M} (\mathbf{y} - \mathbf{x})$ is the squared Mahalanobis distance.*

Going forward, we also denote $f_{\mathbf{x}}^{\mathbf{M}}(\mathbf{a}) = d_{\mathbf{M}}(\mathbf{a}, \mathbf{x})$, and hence, we have $\mu f_{\mathbf{x}}^{\mathbf{M}}(\mathbf{a}) \le f_{\mathbf{x}}(\mathbf{a}) \le f_{\mathbf{x}}^{\mathbf{M}}(\mathbf{a})$, $\forall \mathbf{x}$ and $\forall \mathbf{a} \in \mathbf{A}$. Due to this we say $f_{\mathbf{x}}(\cdot)$ and $f_{\mathbf{x}}^{\mathbf{M}}(\cdot)$ are $\mu$-similar. For Euclidean $k$-means clustering $\mathbf{M}$ is just an identity matrix and $\mu = 1$. It is known that a large set of Bregman divergences is $\mu$-similar, including KL-divergence, Itakura-Saito, Relative Entropy, Harmonic etc (Ackermann & Blömer, 2009). In Table 1, we list the most common $\mu$-similar Bregman divergences, their corresponding $\mathbf{M}$ and the $\mu$. In each case the $\lambda$ and $\nu$ refer to the minimum and maximum values of all coordinates over all points, i.e. the input is a subset of $[\lambda, \nu]^d$.

Table 1: $\mu$-similar Bregman divergences (Lucic et al., 2016).

| **Divergence** | Domain | $\mu$ | $\mathbf{M}$ |
|---|---|---|---|
| Squared-Euclidean | $\mathbb{R}^d$ | 1 | $\mathbf{I}_d$ |
| Mahalanobis$_N$ | $\mathbb{R}^d$ | 1 | $\mathbf{N}$ |
| Exponential-Loss | $[\lambda, \nu]^d \subset \mathbb{R}_+^d$ | $e^{-(\nu - \lambda)}$ | $\frac{e^\nu}{2} \mathbf{I}_d$ |
| Kullback-Leibler | $[\lambda, \nu]^d \subset \mathbb{R}_+^d$ | $\frac{\lambda}{\nu}$ | $\frac{1}{2\lambda} \mathbf{I}_d$ |
| Itakura-Saito | $[\lambda, \nu]^d \subset \mathbb{R}_+^d$ | $\frac{\lambda^2}{\nu^2}$ | $\frac{1}{2\lambda^2} \mathbf{I}_d$ |
| Harmonic$_\alpha$ ($\alpha > 0$) | $[\lambda, \nu]^d \subset \mathbb{R}_+^d$ | $\frac{\lambda^{\alpha+2}}{\nu^{\alpha+2}}$ | $\frac{\alpha(1-\alpha)}{2\lambda^{\alpha+2}} \mathbf{I}_d$ |
| Norm-Like$_\alpha$ ($\alpha > 2$) | $[\lambda, \nu]^d \subset \mathbb{R}_+^d$ | $\frac{\lambda^{\alpha-2}}{\nu^{\alpha-2}}$ | $\frac{\alpha(1-\alpha)}{2} \nu^{\alpha-2} \mathbf{I}_d$ |
| Hellinger-Loss | $[-\nu, \nu]^d \subset (-1, 1)^d$ | $2(1 - \nu^2)^{3/2}$ | $2(1 - \nu)^{-3/2} \mathbf{I}_d$ |

*Mahalanobis distance is also a $\mu$-similar Bregman divergence with $\mu = 1$ and $\mathbf{M}$ is the inverse of the covariance matrix.

There are two types of clustering, hard and soft clustering for Bregman divergence (Banerjee et al., 2005). In this work, by the term clustering, we only refer to the hard clustering problem.

**Coresets:** A coreset Har-Peled & Mazumdar (2004); Agarwal et al. (2005); Badoiu & Clarkson (2003) acts as a small proxy for the original data in the sense that it can be used in place of the original data for a given optimization problem in order to obtain a provably accurate approximate solution to the problem. Let $\epsilon > 0$. For a non-negative cost function, say $f_{\mathbf{X}}(\mathbf{a})$, where $\mathbf{X}$ is a query and $\mathbf{a} \in \mathbf{A}$, a set of subsampled and appropriately reweighted points $(\mathbf{C}, \Omega)$ is an $\epsilon$-coreset if $\forall \mathbf{X}$,

$$|\sum_{\mathbf{a} \in \mathbf{A}} f_{\mathbf{X}}(\mathbf{a}) - \sum_{(\tilde{\mathbf{a}}, \omega_{\tilde{\mathbf{a}}}) \in (\mathbf{C}, \Omega)} \omega_{\tilde{\mathbf{a}}} f_{\mathbf{X}}(\tilde{\mathbf{a}})| \le \epsilon \sum_{\mathbf{a} \in \mathbf{A}} f_{\mathbf{X}}(\mathbf{a}).$$

Typically, the samples that we will construct will satisfy this condition with a desired probability.

While coresets are typically defined for relative errors, additive error coresets can also be defined similarly.

For $\epsilon, \gamma > 0$, $(\mathbf{C}, \Omega)$ is an additive $(\epsilon, \gamma)$ coreset of $\mathbf{A}$ if $\mathbf{C}$ contains points from $\mathbf{A}$ with corresponding weights in $\Omega$, and $\forall \mathbf{X}$, $|\sum_{\mathbf{a} \in \mathbf{A}} f_{\mathbf{X}}(\mathbf{a}) - \sum_{(\tilde{\mathbf{a}}, \omega_{\tilde{\mathbf{a}}}) \in (\mathbf{C}, \Omega)} \omega_{\tilde{\mathbf{a}}} f_{\mathbf{X}}(\tilde{\mathbf{a}})| \le \epsilon \sum_{\mathbf{a} \in \mathbf{A}} f_{\mathbf{X}}(\mathbf{a}) + \gamma$. The coresets that are presented here satisfy such additive guarantees. For ease of representation, sometime we will just use $f_{\mathbf{X}}(\mathbf{C})$ instead of $\sum_{(\tilde{\mathbf{a}}, \omega_{\tilde{\mathbf{a}}}) \in (\mathbf{C}, \Omega)} \omega_{\tilde{\mathbf{a}}} f_{\mathbf{X}}(\tilde{\mathbf{a}})$ and $f_{\mathbf{X}}(\mathbf{A})$ instead of $\sum_{\mathbf{a} \in \mathbf{A}} f_{\mathbf{X}}(\mathbf{a})$.

**Sensitivity Score:** Given an input and an optimization function, a *sensitivity score* for each input point measures the importance of the point for that optimization function. For a dataset $\mathbf{A}$, a query space $\mathcal{X}$ that denotes candidate solutions to an optimization problem, and a cost function $f_{\mathbf{X}}(\cdot)$, Langberg & Schulman (2010) define sensitivity scores as follows— the sensitivity of a point $\mathbf{a}$ is defined as $s_{\mathbf{a}} = \sup_{\mathbf{X} \in \mathcal{X}} \frac{f_{\mathbf{X}}(\mathbf{a})}{\sum_{\mathbf{a}' \in \mathbf{A}} f_{\mathbf{X}}(\mathbf{a}')}$. Note that for all points $\mathbf{a}$, $s_{\mathbf{a}} \in [0, 1]$, and can be treated as a probability score. The sensitivity based coresets Langberg & Schulman (2010) are created by sampling points according to these probabilities (or their upper bounds).

While the above definition is standard, we also define the following variant of sensitivity scores as a useful tool in our results.

**Empirical Sensitivity Score:** In certain cases, for the query space $\mathcal{X}$, it will be challenging to compute a reasonable upper bound to the sensitivity scores. In such cases, we will use empirical sensitivity scores $s_{\mathbf{a}} = \max_{\mathbf{X} \in \mathcal{Y}} \frac{f_{\mathbf{X}}(\mathbf{a})}{\sum_{\mathbf{a}' \in \mathbf{A}} f_{\mathbf{X}}(\mathbf{a}')}$ where $\mathcal{Y}$ is a finite set of queries such that $\mathcal{Y} \subset \mathcal{X}$.

**Online Sensitivity Scores:** For inputs coming in a streaming fashion, i.e., the point $\mathbf{a}_i$ arriving at the $i^{th}$ instance and so far the algorithm has received $\mathbf{A}_{i-1}$; now for a query space $\mathcal{X}$ and a cost function $f_{\mathbf{X}}(\cdot)$ we define the online sensitivity score for every such point $\mathbf{a}_i$ as, $s_{\mathbf{a}_i} = \sup_{\mathbf{X} \in \mathcal{X}} \frac{f_{\mathbf{X}}(\mathbf{a}_i)}{\sum_{j \leq i} f_{\mathbf{X}}(\mathbf{a}_j)}$.

We focus on creating coresets for clustering. Our first results are on $k$-median clustering, where the query space $\mathcal{X}$ satisfies $\mathcal{X} \subseteq \mathbb{R}^{k \times d}$ [1]. One can also define a clustering problem when the number of clusters are unknown. We call this as *Non-Parametric Clustering*. We define $(\mathbf{C}, \Omega)$ to be an $(\epsilon, \gamma)$-additive error *coreset* for non-parametric clustering if its size is independent of $k$ (number of centres) and ensures $|f_{\mathbf{X}}(\mathbf{C}) - f_{\mathbf{X}}(\mathbf{A})| \leq \epsilon f_{\mathbf{X}}(\mathbf{A}) + \gamma$ for all $k \leq n$ and for all query $\mathbf{X} \in \mathbb{R}^{k \times d}$.

**DP-Means:** The DP-Means problem, studied in Kulis & Jordan (2012), formalizes the clustering problem when the number of clusters is unknown. It can be considered to be a specific case of the well-known facility location problem. Given a dataset $\mathbf{A} \in \mathbb{R}^{n \times d}$ and a parameter $\lambda > 0$, the goal of the problem is to find a $k \in (0, n]$ and an $\mathbf{X} \in \mathbb{R}^{k \times d}$ that minimizes the $cost_{DP}(\mathbf{A}, \mathbf{X})$, which is defined as follows,

$$cost_{DP}(\mathbf{A}, \mathbf{X}, k) = \sum_{i \leq n} \min_{\mathbf{x} \in \mathbf{X}} \|\mathbf{a}_i - \mathbf{x}\|^2 + \lambda k.$$

In this paper we consider an obvious extension of the cost function that depends on the Bregman divergences. Given a dataset $\mathbf{A} \in \mathbb{R}^{n \times d}$ and a parameter $\lambda$, the goal of the problem is to find $\mathbf{X} \in \mathbb{R}^{k \times d}$ that minimizes the $cost_{DP}(\mathbf{A}, \mathbf{X})$, defined as follows,

$$cost_{DP}(\mathbf{A}, \mathbf{X}) = f_{\mathbf{X}}(\mathbf{A}) + \lambda k.$$

**Uniform Deviation:** Given a distribution $\mathcal{D}$, input set $\mathbf{A} = \{\mathbf{a}_i, \ldots, \mathbf{a}_m\}$ where each $\mathbf{a}_i$ is an independent and identically distributed sample from $\mathcal{D}$, a query $\mathbf{X}$ and a function $f_{\mathbf{X}}(.)$, the *uniform deviation* is defined as

$$\left| \mathbb{E}_{\mathbf{a} \in \mathcal{D}} f_{\mathbf{X}}(\mathbf{a}) - \frac{1}{m} f_{\mathbf{X}}(\mathbf{A}) \right|.$$

**Definition 2.3** (Pseudo-dimension Haussler (1992))**.** *For a function family $\mathcal{F}$ mapping from an arbitrary input space $\mathcal{X}$ to $\mathbb{R}_{\geq 0}$ and a distribution $\mathcal{P}$ on $\mathcal{X}$, the pseudo-dimension of $\mathcal{F}$, denoted by $Pdim(\mathcal{F})$, is the largest $d$ such there is a sequence $x_1, \ldots, x_d$ of domain elements from $\mathcal{X}$ and a sequence $r_1, \ldots, r_d$ of reals such that for each $b_1, \ldots, b_d \in \{above, below\}$, there is an $f \in \mathcal{F}$ such that for all $i = 1, \ldots, d$, we have $f(x_i) \geq r_i$ if and only if $b_i = above$.*

---

[1]Certain Bregman divergences have non-negativity constraints on the input.

## 3 Related Work

The initial coresets were used for making the computational geometric algorithms more efficient Badoiu & Clarkson (2003), as well as to improve the running times for various clustering problems Har-Peled & Mazumdar (2004); Agarwal et al. (2005). Since then there has been a significant amount of work on coresets. Interested readers can look at (Woodruff et al., 2014; Bachem et al., 2017b) and the references therein. Using sensitivities to construct coresets was introduced in (Langberg & Schulman, 2010) and further generalized by (Feldman & Langberg, 2011). Coresets for clustering problems such as $k$-means clustering has been extensively studied (Har-Peled & Mazumdar, 2004; Cohen et al., 2015; Braverman et al., 2016; Feldman et al., 2016; Bachem et al., 2018a;b; Barger & Feldman, 2020; Feldman et al., 2020). In (Cohen et al., 2015) the authors reduce the $k$-means problem to a constrained low rank approximation problem. They show that a constant factor approximation can be achieved by just $O(\epsilon^{-2} \log k)$ size coreset and for $(1 \pm \epsilon)$ relative error approximation they get coreset of size $O(k\epsilon^{-2})$. In (Barger & Feldman, 2020; Feldman et al., 2016), the authors discuss a deterministic algorithm for creating coresets for clustering problem which ensures a relative error approximation. The streaming version of (Barger & Feldman, 2020) returns a coreset of size $O(k^{\epsilon^{-2}} \epsilon^{-2} \log n)$ which ensures a $(1 \pm \epsilon \log n)$ relative error approximation. Feldman et al. (2016) reduce the problem of $k$-means clustering to $\ell_2$ frequent item approximation. The streaming version of the algorithm returns a coreset of size $O(k^2 \epsilon^{-2} \log^2 n)$. In (Bachem et al., 2018b), the authors give an algorithm which returns a one shot coreset for all $\ell_p$ $k$-clustering problem, where $p \in [1, p_{\max}]$. Their algorithm creates a grid over the range $[1, p_{\max}]$ and based on the sensitivity at each grid point the coreset is built. It returns a coreset of size $\tilde{O}(16^{p_{\max}} dk^2)$ for which it takes $\tilde{O}(ndk)$ ensuring $(1 \pm \epsilon)$ relative error approximation. In (Boutsidis & Magdon-Ismail, 2013; Cohen et al., 2015), authors show that one can use spectral approximation technique (Batson et al., 2012) for deterministic feature selection for $k$-means problem. In (Bachem et al., 2018a), the authors give an algorithm to create a coreset which only takes $O(nd)$ time and returns a coreset of size $O(dk\epsilon^{-2} \log k)$ at a cost of small additive error approximation. Their algorithm can further be extended for clustering based Bregman divergences which are $\mu$-similar to squared Mahalanobis distance. In (Lucic et al., 2016) the authors give algorithm to create such coresets for both hard and soft clustering based on $\mu$-similar Bregman Divergence. In this paper we present an online algorithm which is returns just a $\tilde{O}(dk\epsilon^{-2} \log k)$ size coreset with similar guarantees.

There are several online algorithms for $k$-means clustering (Liberty et al., 2016; Lattanzi & Vassilvitskii, 2017; Bhaskara & Rwanpathirana, 2020). These algorithms do not create coresets, rather focus on giving online algorithms for the clustering problem. In (Liberty et al., 2016) the authors give an online algorithm that maintains a set of centers such that the $k$-means cost on these centers is $\tilde{O}(W^*)$ where $W^*$ is the optimal $k$-means cost. Lattanzi et.al., Lattanzi & Vassilvitskii (2017) improve this result and give a robust algorithm that can also handle outliers in the dataset. We further explore the relation of these algorithms with our online coresets empirically in the experimental section.

The coreset building methods can also be analyzed from the point of view of generalization of the resulting clustering. There are several results on uniform bound deviation for $k$-means clustering problem (Biau et al., 2008; Bachem et al., 2017a). The result in Biau et al. (2008) shows that the difference between the optimal empirical loss and the optimal clustering loss (i.e., the expected excess risk) can be bounded in terms of the radius of the input data and inverse of the input size. As our main contribution in this paper is about building strong coresets, we extend the results in Bachem et al. (2017a), which ensure that the difference between empirical loss and clustering loss for all possible queries are bounded.

We use Theorem 3.2 from Chhaya et.al., Chhaya et al. (2020b), where the authors show that the coreset built using sensitivity framework has a sampling complexity that only depends on $O(S)$ instead of $O(S \log(S))$ in (Braverman et al., 2016), where $S$ is the sum of sensitivity scores. Our coreset size for clustering based on $\mu$-similar Bregman divergence only has a dependence of $O(1/\mu)$, unlike in (Lucic et al., 2016; Bachem et al., 2018a) where the dependence is $O(1/\mu^2)$.

---

**Algorithm 1** `BregmanFilter`

---

**Require:** Streaming points $\mathbf{a}_i, i = 1, 2, \ldots, n; r > 0$
**Ensure:** $(\mathbf{C}, \Omega)$

   $\mathbf{C}_0 = \Omega_0 = \varphi_0 = \emptyset; S = 0$
   $\lambda = \|\mathbf{a}_1\|_{\min}; \quad \nu = \|\mathbf{a}_1\|_{\max}$
   **while** $i \leq n$ **do**
      $\lambda = \min\{\lambda, \|\mathbf{a}_i\|_{\min}\}; \nu = \max\{\nu, \|\mathbf{a}_i\|_{\max}\}$
      Update $\mu_i = \lambda/\nu$; Update $\mathbf{M}_i$ based on (Table 1)
      $\varphi_i = ((i-1)\varphi_{i-1} + \mathbf{a}_i)/i; S = S + f_{\varphi_i}^{\mathbf{M}_i}(\mathbf{a}_i)$
      **if** $i = 1$ **then**
         $p_i = 1$
      **else**
         $l_i = \frac{2f_{\varphi_i}^{\mathbf{M}_i}(\mathbf{a}_i)}{\mu_i S} + \frac{8}{\mu_i(i-1)}; p_i = \min\{1, rl_i\}$
      **end if**
      $(\mathbf{c}_i, \omega_i) = \begin{cases} (\mathbf{a}_i, 1/p_i) & \text{w. p. } p_i \\ (\emptyset, 0) & \text{else} \end{cases}$
      $(\mathbf{C}_i, \Omega_i) = (\mathbf{C}_{i-1}, \Omega_{i-1}) \cup (\mathbf{c}_i, \omega_i)$
   **end while**
   Return $(\mathbf{C}, \Omega)$

---

## 4 Online Coresets for Clustering

We consider that the input points are coming in a streaming manner, i.e., each point on arrival is either selected to be in the coreset or discarded. We present our first algorithm called `BregmanFilter`. It creates a coreset in an online manner for clustering based on Bregman divergences. The coreset is built via importance sampling, which is based on sensitivity framework, i.e., we use sensitivity scores to define the sampling probability of each point. The algorithm starts with the knowledge of Bregman divergence $d_\Phi$. It is important to note that for a fixed $d_\Phi$ if the domain of the input changes, then both the parameters $\mathbf{M}$ and $\mu$ also change (Ackermann & Blömer, 2009; Lucic et al., 2016) (table 1). This is exactly what happens in our case, because we only have access to the stream of input. These parameters are important because they are used to compute an upper bound on the sensitivity scores of every points.

**Overview:** Under a fixed $\mu$-similar Bregman divergence, the input to the algorithm is a stream of points and an user defined parameter $r$ that depends on $\epsilon$ and the VC dimension of query space of the problem, such that it returns $(\mathbf{C}, \Omega)$ which is an $\epsilon$ coreset. Here on arrival of every input point say $\mathbf{a}_i$, the algorithm first updates the smallest and largest absolute values as $\lambda$ and $\nu$, such that $\lambda = \|\mathbf{A}_i\|_{\min}$ and $\nu = \|\mathbf{A}_i\|_{\max}$. It also updates the mean $\varphi_i$. Next it computes both the Mahalanobis matrix $\mathbf{M}_i$ as well as $\mu_i$. Fortunately, computing $\mathbf{M}_i$ and $\mu_i$ requires maintaining only two simple statistic of the data (Table 1). Using these terms it computes an upper bound for the sensitivity score, which is then used to decide whether $\mathbf{a}_i$ should be stored in the coreset. If selected, the point $\mathbf{a}_i$ is stored with an appropriate weight $\omega_i$. At the end of the stream we get $(\mathbf{C}, \Omega)$. Note that this algorithm is online in nature because for every point its sampling decision is taken before looking at the next incoming point. We present `BregmanFilter` as algorithm 1.

Now we present some supporting lemmas based on which we show the correctness and the corresponding guarantees of the algorithm. Note that for $\mathbf{A}_i$ formed by first $i$ data points. The algorithm `BregmanFilter` maintains $\mathbf{M}_i$ and $\mu_i$ (as per Table 1) such that, $\mu_i f_{\mathbf{x}}^{\mathbf{M}_i}(\mathbf{a}_j) \leq f_{\mathbf{x}}(\mathbf{a}_j) \leq f_{\mathbf{x}}^{\mathbf{M}_i}(\mathbf{a}_j)$, $\forall \mathbf{x}$ and $\forall \mathbf{a}_j \in \mathbf{A}_i$. Using this we show a useful observation that is immediate, based on $\mathbf{M}$ and $\mu$ defined in Table 1.

**Lemma 4.1.** *For all Bregman divergences in Table 1, for $j \leq i$, $\mu_j \geq \mu_i$ and $\mathbf{M}_j \preceq \mathbf{M}_i$.*

*Proof.* At any $i^{th}$ point we have $\lambda = \|\mathbf{A}_i\|_{\min}$ and $\nu = \|\mathbf{A}_i\|_{\max}$, i.e., the smallest and largest absolute values in $\mathbf{A}_i$. Further we have $\|\mathbf{A}_j\|_{\min} \geq \|\mathbf{A}_i\|_{\min}$ and $\|\mathbf{A}_j\|_{\max} \leq \|\mathbf{A}_i\|_{\max}$ for $j \leq i$. By using the formula for $\mathbf{M}$ for all Bregman divergences given in Table 1 we have $\mathbf{M}_j \preceq \mathbf{M}_i$ and $\mu_j \geq \mu_i$ to be always true for $j \leq i$. □

For any Bergman divergence the mean of a set of points always minimizes the sum of Bergman divergences between the set of points and any other point. Now recall that for all $i \in [n]$, we use $\varphi_i$ to represent the mean of the first $i$ points, i.e., $\varphi_i = \sum_{j \leq i} \mathbf{a}_j / i$. Hence we have the following important observation.

**Lemma 4.2.** *For points arriving in streaming manner, $\forall i > j$ we have, $f_{\varphi_i}(\mathbf{A}_i) \geq f_{\varphi_j}(\mathbf{A}_j)$.*

*Proof.* We have,

$$f_{\varphi_i}(\mathbf{A}_i) = \sum_{r \leq j} f_{\varphi_i}(\mathbf{a}_r) + \sum_{j < r \leq i} f_{\varphi_i}(\mathbf{a}_r) \overset{(i)}{\geq} \sum_{r \leq j} f_{\varphi_j}(\mathbf{a}_r) + \sum_{j < r \leq i} f_{\varphi_i}(\mathbf{a}_r) \geq f_{\varphi_j}(\mathbf{A}_j).$$

In $(i)$ we use the fact that $\varphi_j$ minimizes the sum of Bregman divergence between any point and first $j$ points. $\square$

Due to lemma 4.1 and lemma 4.2, we have the following lemma which is then used to upper bound the online sensitivity scores with $l_i$ as defined in `BregmanFilter`.

**Lemma 4.3.** *For a fixed $\mu$-similar Bregman divergence with streaming inputs, let $\varphi_i = \sum_{j \leq i} (\mathbf{a}_j / i)$ and $\mathbf{M}_i$ is the p.s.d. Mahalanobis matrix for $\mathbf{A}_i$ (as Table 1). If $j \leq i$, then $f_{\varphi_i}^{\mathbf{M}_i}(\mathbf{A}_i) \geq \sum_{j \leq i} f_{\varphi_j}^{M_j}(\mathbf{a}_j)$*

*Proof.*

$$
\begin{aligned}
f_{\varphi_i}^{\mathbf{M}_i}(\mathbf{A}_i) &= \sum_{\mathbf{j} \leq i} (\mathbf{a}_j - \varphi_i)^T \mathbf{M}_i (\mathbf{a}_j - \varphi_i) \\
&\overset{(i)}{\geq} (\mathbf{a}_j - \varphi_i)^T \mathbf{M}_i (\mathbf{a}_j - \varphi_i) + \sum_{j \leq i-1} (\mathbf{a}_j - \varphi_i)^T \mathbf{M}_{i-1} (\mathbf{a}_j - \varphi_i) \\
&\geq (\mathbf{a}_j - \varphi_i)^T \mathbf{M}_i (\mathbf{a}_j - \varphi_i) + \sum_{j \leq i-1} (\mathbf{a}_j - \varphi_i)^T \mathbf{M}_j (\mathbf{a}_j - \varphi_i) \\
&\overset{(ii)}{\geq} (\mathbf{a}_j - \varphi_i)^T \mathbf{M}_i (\mathbf{a}_j - \varphi_i) + \sum_{j \leq i-1} (\mathbf{a}_j - \varphi_{i-1})^T \mathbf{M}_j (\mathbf{a}_j - \varphi_{i-1}) \\
&\geq (\mathbf{a}_j - \varphi_i)^T \mathbf{M}_i (\mathbf{a}_j - \varphi_i) + \sum_{j \leq i-1} (\mathbf{a}_j - \varphi_j)^T \mathbf{M}_j (\mathbf{a}_j - \varphi_j) \\
&= \sum_{j \leq i} f_{\varphi_j}^{\mathbf{M}_j}(\mathbf{a}_j)
\end{aligned}
$$

In $(i)$ we used the property that $\mathbf{M}_{i-1} \preceq \mathbf{M}_i$. In $(ii)$ we used the fact that for $\mathbf{A}_{i-1}$, its mean $\varphi_{i-1}$ minimizes the cost. $\square$

As we use $l_i$ for building our coreset, the expected coreset size depends on the sum of $l_i$'s. In the next lemma we prove that the scores $l_i$'s defined in `BregmanFilter` upper bound the online sensitivity scores of $\mathbf{a}_i$, i.e., $\sup_{\mathbf{X} \in \mathbb{R}^{k \times d}} \frac{f_{\mathbf{X}}(\mathbf{a}_i)}{f_{\mathbf{X}}(\mathbf{A}_{i-1}) + f_{\varphi_i}(\mathbf{A}_i)}$ and we also show that the sum of $l_i$'s is also bounded.

**Lemma 4.4.** *For every incoming points $\mathbf{a}_i$ the $l_i$ as defined in `BregmanFilter`, upper bounds the online sensitivity score. I.e., $\forall i \in [n]$,*

$$\sup_{\mathbf{X} \in \mathbb{R}^{k \times d}} \frac{f_{\mathbf{X}}(\mathbf{a}_i)}{f_{\mathbf{X}}(\mathbf{A}_{i-1}) + f_{\varphi_i}(\mathbf{A}_i)} \leq l_i \tag{2}$$

*Furthermore,*

$$\sum_{i \leq n} l_i \leq \left( 8 \log n + 4 \log \left( f_{\varphi}^{\mathbf{M}}(\mathbf{A}) \right) - 4 \log \left( f_{\varphi_2}^{\mathbf{M}_2}(\mathbf{a}_2) \right) \right) / \mu.$$

*Proof.* At step $i$, let $(\mu_i, \mathbf{M}_i)$ be the parameters such that, $\forall \mathbf{a}_i \in \mathbf{A}$ and $\forall \mathbf{X}, \mu_i f_{\mathbf{x}}^{\mathbf{M}_i}(\mathbf{a}_j) \leq f_{\mathbf{x}}(\mathbf{a}_j) \leq f_{\mathbf{x}}^{\mathbf{M}_i}(\mathbf{a}_j)$ and $\varphi_i = \frac{\sum_{j \leq i} \mathbf{a}_j}{i}$. Now for any query $\mathbf{X} \in \mathbb{R}^{k \times d}$, each point $\mathbf{a}_j \in \mathbf{A}_{i-1}$ has some closest point $\mathbf{x}_l \in \mathbf{X}$. Now for such pair $\{\mathbf{a}_j, \mathbf{x}_l\}$, using the property $(\|a+b\|)^2 \leq 2(\|a\|^2 + \|b\|^2)$ we have $f_{\mathbf{x}_l}^{\mathbf{M}_i}(\varphi_i) \leq 2 f_{\mathbf{x}_l}^{\mathbf{M}_i}(\mathbf{a}_j) + 2 f_{\varphi_i}^{\mathbf{M}_i}(\mathbf{a}_j)$. So by taking into account for all the points in $\mathbf{A}_{i-1}$ we get $(i-1) f_{\mathbf{X}}^{\mathbf{M}_i}(\varphi_i) \leq 2 \sum_{\mathbf{a}_j \in \mathbf{A}_{i-1}} (f_{\mathbf{X}}^{\mathbf{M}_i}(\mathbf{a}_j) + f_{\varphi_i}^{\mathbf{M}_i}(\mathbf{a}_j)) = 2 f_{\mathbf{X}}^{\mathbf{M}_i}(\mathbf{A}_{i-1}) + 2 f_{\varphi_i}^{\mathbf{M}_i}(\mathbf{A}_{i-1})$. We use this triangle inequality in the following analysis, which holds $\forall \mathbf{X} \in \mathbb{R}^{k \times d}$,

$$
\begin{aligned}
\frac{f_{\mathbf{X}}(\mathbf{a}_i)}{f_{\mathbf{X}}(\mathbf{A}_{i-1}) + f_{\varphi_i}(\mathbf{A}_i)} &\overset{(i)}{\leq} \frac{f_{\mathbf{X}}^{\mathbf{M}_i}(\mathbf{a}_i)}{f_{\mathbf{X}}(\mathbf{A}_{i-1}) + f_{\varphi_i}(\mathbf{A}_i)} \\
&\leq \frac{2 f_{\varphi_i}^{\mathbf{M}_i}(\mathbf{a}_i) + 2 f_{\mathbf{X}}^{\mathbf{M}_i}(\varphi_i)}{f_{\mathbf{X}}(\mathbf{A}_{i-1}) + f_{\varphi_i}(\mathbf{A}_i)} \\
&\leq \frac{2 f_{\varphi_i}^{\mathbf{M}_i}(\mathbf{a}_i) + \frac{4}{i-1} f_{\varphi_i}^{\mathbf{M}_i}(\mathbf{A}_{i-1}) + \frac{4}{i-1} f_{\mathbf{X}}^{\mathbf{M}_i}(\mathbf{A}_{i-1})}{f_{\mathbf{X}}(\mathbf{A}_{i-1}) + f_{\varphi_i}(\mathbf{A}_i)} \\
&\overset{(ii)}{\leq} \frac{2 f_{\varphi_i}^{\mathbf{M}_i}(\mathbf{a}_i) + \frac{4}{i-1} f_{\varphi_i}^{\mathbf{M}_i}(\mathbf{A}_{i-1}) + \frac{4}{i-1} f_{\mathbf{X}}^{\mathbf{M}_i}(\mathbf{A}_{i-1})}{\mu_i (f_{\mathbf{X}}^{\mathbf{M}_i}(\mathbf{A}_{i-1}) + f_{\varphi_i}^{\mathbf{M}_i}(\mathbf{A}_i))} \\
&= \frac{2 f_{\varphi_i}^{\mathbf{M}_i}(\mathbf{a}_i) + \frac{4}{i-1} f_{\varphi_i}^{\mathbf{M}_i}(\mathbf{A}_{i-1})}{\mu_i (f_{\mathbf{X}}^{\mathbf{M}_i}(\mathbf{A}_{i-1}) + f_{\varphi_i}^{\mathbf{M}_i}(\mathbf{A}_i))} + \frac{\frac{4}{i-1} f_{\mathbf{X}}^{\mathbf{M}_i}(\mathbf{A}_{i-1})}{\mu_i (f_{\mathbf{X}}^{\mathbf{M}_i}(\mathbf{A}_{i-1}) + f_{\varphi_i}^{\mathbf{M}_i}(\mathbf{A}_i))} \\
&\leq \frac{2 f_{\varphi_i}^{\mathbf{M}_i}(\mathbf{a}_i) + \frac{4}{i-1} f_{\varphi_i}^{\mathbf{M}_i}(\mathbf{A}_{i-1})}{\mu_i (f_{\mathbf{X}}^{\mathbf{M}_i}(\mathbf{A}_{i-1}) + f_{\varphi_i}^{\mathbf{M}_i}(\mathbf{A}_i))} + \frac{4}{\mu_i (i-1)} \\
&\leq \frac{2 f_{\varphi_i}^{\mathbf{M}_i}(\mathbf{a}_i)}{\mu_i f_{\varphi_i}^{\mathbf{M}_i}(\mathbf{A}_i)} + \frac{\frac{4}{i-1} f_{\varphi_i}^{\mathbf{M}_i}(\mathbf{A}_{i-1})}{\mu_i f_{\varphi_i}^{\mathbf{M}_i}(\mathbf{A}_i)} + \frac{4}{\mu_i (i-1)} \\
&\overset{(iii)}{\leq} \frac{2 f_{\varphi_i}^{\mathbf{M}_i}(\mathbf{a}_i)}{\mu_i f_{\varphi_i}^{\mathbf{M}_i}(\mathbf{A}_i)} + \frac{8}{\mu_i (i-1)} \\
&\leq \frac{2 f_{\varphi_i}^{\mathbf{M}_i}(\mathbf{a}_i)}{\mu_i \sum_{j \leq i} f_{\varphi_j}^{\mathbf{M}_j}(\mathbf{a}_j)} + \frac{8}{\mu_i (i-1)}
\end{aligned}
$$

The inequality $(i)$ is due to $\mu_i$ similarity, i.e., $f_{\mathbf{X}}(\mathbf{a}_i) \leq f_{\mathbf{X}}^{\mathbf{M}_i}(\mathbf{a}_i)$. Next couple of inequalities are by applying triangle inequality on the numerator. In the $(ii)$ inequality we use the $\mu_i$ similarity to get a lower bound on the denominator term. We get $(iii)$ inequality by upper bounding the second and third term by $4/(\mu_i(i-1))$. By the property of Bregman divergence we know that $\varphi_{i-1} = \arg\min_{\mathbf{x}} f_{\mathbf{x}}(\mathbf{A}_{i-1})$ and, so we have $f_{\varphi_i}(\mathbf{A}_i) = f_{\varphi_i}(\mathbf{A}_{i-1}) + f_{\varphi_i}(\mathbf{a}_i) \geq f_{\varphi_{i-1}}(\mathbf{A}_{i-1}) + f_{\varphi_i}(\mathbf{a}_i)$. So by induction we get $f_{\varphi_i}(\mathbf{A}_i) \geq \sum_{j \leq i} f_{\varphi_j}(\mathbf{a}_i)$.

As, $f_{\varphi_i}^{\mathbf{M}_i}(\mathbf{A}_i) \geq \sum_{j \leq i} f_{\varphi_j}^{\mathbf{M}_j}(\mathbf{a}_j)$ which can be maintained as a running sum hence $\forall i$, the score $l_i$ can be computed in just one pass.

Next, in order to upper bound $\sum_{i \leq n} l_i$, consider the denominator term of $l_i$ as follows,

$$
\begin{aligned}
\sum_{j \leq i} f_{\varphi_j}^{\mathbf{M}_j}(\mathbf{a}_j) &= \sum_{j \leq i-1} f_{\varphi_j}^{\mathbf{M}_j}(\mathbf{a}_j) + f_{\varphi_i}^{\mathbf{M}_i}(\mathbf{a}_i) \\
&= \sum_{j \leq i-1} f_{\varphi_j}^{\mathbf{M}_j}(\mathbf{a}_j) \left( 1 + \frac{f_{\varphi_i}^{\mathbf{M}_i}(\mathbf{a}_i)}{\sum_{j \leq i-1} f_{\varphi_j}^{\mathbf{M}_j}(\mathbf{a}_j)} \right) \\
&\geq \sum_{j \leq i-1} f_{\varphi_j}^{\mathbf{M}_j}(\mathbf{a}_j) \left( 1 + \frac{f_{\varphi_i}^{\mathbf{M}_i}(\mathbf{a}_i)}{\sum_{j \leq i} f_{\varphi_j}^{\mathbf{M}_j}(\mathbf{a}_j)} \right) \\
&= \sum_{j \leq i-1} f_{\varphi_j}^{\mathbf{M}_j}(\mathbf{a}_j)(1 + q_i) \\
&\overset{(i)}{\geq} \exp(q_i/2) \sum_{j \leq i-1} f_{\varphi_j}^{\mathbf{M}_j}(\mathbf{a}_j)
\end{aligned}
$$

$$\exp(q_i/2) \quad \leq \quad \frac{\sum_{j \leq i} f_{\varphi_j}^{\mathbf{M}_j}(\mathbf{a}_j)}{\sum_{j \leq i-1} f_{\varphi_j}^{\mathbf{M}_j}(\mathbf{a}_j)}$$

where for inequality $(i)$ we used that $q_i = \frac{f_{\varphi_i}^{\mathbf{M}_i}(\mathbf{a}_i)}{\sum_{j \leq i} f_{\varphi_j}^{\mathbf{M}_j}(\mathbf{a}_j)} \leq 1$ and hence we have $(1 + q_i) \geq \exp(q_i/2)$. Now as we know that $\sum_{j \leq i} f_{\varphi_j}^{\mathbf{M}_j}(\mathbf{a}_j) \geq \sum_{j \leq i-1} f_{\varphi_j}^{\mathbf{M}_j}(\mathbf{a}_j)$ hence the following product results into a telescopic product and we get,

$$\prod_{2 \leq i \leq n} \exp(q_i/2) \quad \leq \quad \frac{\sum_{j \leq n} f_{\varphi_j}^{\mathbf{M}_j}(\mathbf{a}_j)}{f_{\varphi_2}^{\mathbf{M}_2}(\mathbf{a}_2)}$$

So by taking logarithm of both sides we get $\sum_{2 \leq i \leq n} q_i \leq 2 \log\left(f_\varphi^{\mathbf{M}}(\mathbf{A})\right) - 2 \log\left(f_{\varphi_2}^{\mathbf{M}_2}(\mathbf{a}_2)\right)$. Further incorporating the terms $\frac{8}{\mu_i(i-1)}$ we have $l_i = \frac{2q_i}{\mu_i} + \frac{8}{\mu_i(i-1)}$. Hence, $\sum_{2 \leq i \leq n} l_i \leq 4\mu^{-1}(\log n + \log\left(f_\varphi^{\mathbf{M}}(\mathbf{A})\right) - \log\left(f_{\varphi_2}^{\mathbf{M}_2}(\mathbf{a}_2)\right))$. Where $\mu = \mu_n \leq \mu_i$ and $\mathbf{M} \succeq \mathbf{M}_n \succeq \mathbf{M}_i$ for all $i \leq n$. $\qquad \square$

Note that the upper bounds and the sum are independent of $k$ (#clusters). Now in the next Lemma we show that by sampling enough points based on $l_i$, we get equation 1.

**Lemma 4.5.** *In* `BregmanFilter`*, setting* $r = O\left(\frac{dk(\log k)\log(1/\epsilon)}{\epsilon^2}\right)$*, the returned coreset* $(\mathbf{C}, \Omega)$ *satisfies the following guarantee with at least $0.99$ probability $\forall \mathbf{X} \in \mathbb{R}^{k \times d}$*

$$|f_{\mathbf{X}}(\mathbf{C}, \Omega) - f_{\mathbf{X}}(\mathbf{A})| \leq \epsilon(f_{\mathbf{X}}(\mathbf{A}) + f_\varphi(\mathbf{A})) \tag{3}$$

We proof this lemma by applying Bernstein's inequality on the following random variables

$$w_i = \begin{cases} (1/p_i - 1)f_{\mathbf{X}}(\mathbf{a}_i) & \text{with probability } p_i \\ -f_{\mathbf{X}}(\mathbf{a}_i) & \text{with probability } (1 - p_i). \end{cases}$$

A detailed proof is discussed in the appendix (A.1.1).

Now using the Lemmas 4.4 and 4.5, we have the following main theorem of this section.

**Theorem 4.6.** *For points coming in streaming fashion,* `BregmanFilter` *returns a coreset* $(\mathbf{C}, \Omega)$ *for the clustering based on Bregman divergence such that for all $\mathbf{X} \in \mathbb{R}^{k \times d}$, with at least $0.99$ probability it ensures the following guarantee.*

$$|f_{\mathbf{X}}(\mathbf{C}, \Omega) - f_{\mathbf{X}}(\mathbf{A})| \leq \epsilon(f_{\mathbf{X}}(\mathbf{A}) + f_\varphi(\mathbf{A}))$$

*Such a coreset has* $O\left(\frac{dk(\log k)\log(1/\epsilon)}{\mu\epsilon^2}\left(\log n + \log\left(f_\varphi^{\mathbf{M}}(\mathbf{A})\right) - \log\left(f_{\varphi_2}^{\mathbf{M}_2}(\mathbf{a}_2)\right)\right)\right)$ *expected samples.* `BregmanFilter` *takes $O(d)$ update time and uses $O(d)$ working space.*

The expected sample size of the coreset $(\mathbf{C}, \Omega)$ returned by `BregmanFilter` is bounded by $r \sum_{i \leq n} l_i$. Using Lemma 4.4 and Lemma 4.5 we obtain the expected sample size to be $O\left(\frac{dk(\log k)\log(1/\epsilon)}{\mu\epsilon^2}\left(\log n + \log\left(f_\varphi^{\mathbf{M}}(\mathbf{A})\right) - \log\left(f_{\varphi_2}^{\mathbf{M}_2}(\mathbf{a}_2)\right)\right)\right)$. Further, by using the $\mu$-similarity expected sample size is also $O\left(\frac{dk(\log k)\log(1/\epsilon)}{\mu^2\epsilon^2}\left(\log n + \log\left(f_\varphi(\mathbf{A})\right) - \log\left(f_{\varphi_2}(\mathbf{a}_2)\right)\right)\right)$.

The algorithm requires a working space of $O(d)$ which is to maintain the mean(centre) $\varphi_i$, $\mu_i$, $\mathbf{M}_i$(sparse matrix) and the running sum $S$. Further for every incoming point `BregmanFilter` only needs to compute the distance between the point and the current mean hence the update time is just $O(d)$. So the running time of the entire algorithm is $O(nd)$, which is why it is easy to scale for large $n$. Note that although Theorem 4.6 gives the guarantees at the last instance, but using the same analysis technique one can ensure an equivalent guarantee at every $i^{th}$ instance by taking an union bound– this requires an extra multiplicative factor of $\log(n)$ in the sample size. By doing so, $(\mathbf{C}_i, \Omega_i)$ satisfies the following for $\mathbf{A}_i$, $\forall i \in [n]$ and $\forall \mathbf{X} \in \mathbb{R}^{k \times d}$ with at least $0.99$ probability,

$$|f_{\mathbf{X}}(\mathbf{C}_i, \Omega_i) - f_{\mathbf{X}}(\mathbf{A}_i)| \leq \epsilon(f_{\mathbf{X}}(\mathbf{A}_i) + f_{\varphi_i}(\mathbf{A}_i)) \tag{4}$$

Note that `BregmanFilter` returns a smaller coreset $\mathbf{C}$ compare to offline coresets Bachem et al. (2018b); Lucic et al. (2016) but at a cost of additive factor approximation that depends on the structure of the data. Further unlike Bachem et al. (2018a); Lucic et al. (2016) our sampling complexity only depends on $1/\mu$. `BregmanFilter` can be easily generalized to create coresets for weighted clustering where each point $\mathbf{a}_i$ has some weight $w_i$ such that $f_{\mathbf{X}}(\mathbf{a}_i) = w_i \min_{\mathbf{x} \in \mathbf{X}} d_\Phi(\mathbf{a}_i, \mathbf{x})$. While sampling point (say $\mathbf{a}_i$) the algorithm `BregmanFilter` sets $\mathbf{c}_i = \mathbf{a}_i$ and $\omega_i = w_i/p_i$ with probability $p_i$.

Notice that the additive term (second term) in the guarantee of Theorem 4.6 depends on the variance of the input data. So it can be significantly large compare to the relative term (first term). One way to address this issue is to use an user defined parameter $\tau \in (0, 1)$ to reduce the effect of additive term. In the algorithm `BregmanFilter`, one needs to update $l_i = \frac{2f_{\varphi_i}^{\mathbf{M}_i}(\mathbf{a}_i)}{\mu_i \tau S} + \frac{4}{\mu_i(i-1)}(\frac{1}{\tau} + 1)$. In the following corollary we state our guarantee.

**Corollary 4.1.** *For points coming in streaming fashion and with the above change in `BregmanFilter`, the algorithm returns a coreset $(\mathbf{C}, \Omega)$ for the clustering based on Bregman divergence such that for all $\mathbf{X} \in \mathbb{R}^{k \times d}$, with at least $0.99$ probability it ensures the following guarantee.*

$$|f_{\mathbf{X}}(\mathbf{C}, \Omega) - f_{\mathbf{X}}(\mathbf{A})| \leq \epsilon f_{\mathbf{X}}(\mathbf{A}) + \epsilon' f_{\varphi}(\mathbf{A})$$

*where $\epsilon' = \epsilon \cdot \tau$. Such a coreset has $O\left(\frac{dk(\log k)\log(1/\epsilon)}{\mu \epsilon^2}\left(\log n + \frac{\log\left(f_{\varphi}^{\mathbf{M}}(\mathbf{A})\right) - \log\left(f_{\varphi_2}^{\mathbf{M}_2}(\mathbf{a}_2)\right)}{\tau}\right)\right)$ expected samples. It takes $O(d)$ update time and uses $O(d)$ working space.*

The proof is similar to the proof of Theorem 4.6. Here, an important change in the analysis is that instead of equation 2 we use following sensitivity function,

$$\sup_{\mathbf{X} \in \mathbb{R}^{k \times d}} \frac{f_{\mathbf{X}}(\mathbf{a}_i)}{f_{\mathbf{X}}(\mathbf{A}_{i-1}) + \tau f_{\varphi_i}(\mathbf{A}_i)}. \tag{5}$$

In the appendix we first present the upper on these scores followed by the proof of the above corollary. Note that with an increase in the coreset size by a factor of $\frac{1}{\tau}$ one can improve the additive error guarantee by a factor of $\tau$.

## 4.1 Online Coresets for $k$-means Clustering

In $k$-means clustering, most commonly used Bregman divergence measure is the squared euclidean distance. Now for squared euclidean distance we know that, $\mathbf{M}_i = \mathbf{I}_d$ and $\mu_i = 1, \forall i \in [n]$. So in this case the algorithm `BregmanFilter` does not need to maintain $\mathbf{M}_i$ and $\mu_i$. In the following corollary we state the guarantee of `BregmanFilter` for k-means clustering.

**Corollary 4.2.** *Let $\mathbf{A} \in \mathbb{R}^{n \times d}$ such that the points are coming in streaming manner and fed to `BregmanFilter`, it returns a coreset $(\mathbf{C}, \Omega)$ which ensures the guarantee equation 1 $\forall \mathbf{X} \in \mathbb{R}^{k \times d}$ with probability at least $0.99$ for the k-means problem. Such a coreset has $O\left(\frac{dk(\log k)\log(1/\epsilon)}{\epsilon^2}\left(\log n + \log\left(f_{\varphi}(\mathbf{A})\right) - \log\left(f_{\varphi_2}(\mathbf{a}_2)\right)\right)\right)$ expected samples. The update time is $O(d)$ time and uses $O(d)$ as working space.*

*Proof.* The proof follows by combining Lemma 4.4 and Lemma 4.5. As k-means clustering has $\mathbf{M}_i = \mathbf{I}_d$ and $\mu_i = 1$ for all $i \leq n$, the upper bound on the sensitivity scores is just,

$$l_i = \frac{f_{\varphi_i}(\mathbf{a}_i)}{\sum_{j \leq i} f_{\varphi_j}(\mathbf{a}_j)} + \frac{8}{i-1}$$

It can be verified by a similar analysis as in the proof of Lemma 4.4. The proof of the second part of Lemma 4.4 and Lemma 4.5 will follow as it is. Further note that k-means is a hard clustering hence the $\epsilon$-net size is $O(\epsilon^{-dk \log k})$. Hence the expected size of $\mathbf{C}$ returned by `BregmanFilter` is $O\left(\frac{dk(\log k)\log(1/\epsilon)}{\epsilon^2}\left(\log n + \log\left(f_{\varphi}(\mathbf{A})\right) - \log\left(f_{\varphi_2}(\mathbf{a}_2)\right)\right)\right)$. Further, similar to other $\mu$-similar Bregman divergences, here in this case also the update time is $O(d)$ and uses a working space of $O(d)$. $\square$

# 5 Coresets for Non-Parametric Clustering

For data coming in an streaming fashion, it is extremely challenging to anticipate the value of $k$ with having some domain knowledge. For such an input one requires to solve a non-parametric clustering problem, i.e., the problem is find a clustering as well as a value of $k$.

Our previous algorithm `BregmanFilter` requires such $k$ and the size of the coreset returned by it depends on such $k$ (#clusters). It is mainly due to the union bound we need to take over a set in the query space ($\epsilon$-net). Here we explore the possibility of building a coreset non-parametric clustering problem. Naturally the size of such coresets are independent of any $k$. However, it ensures a provable guarantee for any query $\mathbf{X}$ with at most $n$ centers.

Even showing the existence of such a coreset is not obvious. It is not difficult to realize that it is impossible to get a *sublinear sized* coreset for non-parametric clustering which ensures a relative error approximation. The following example illustrates this. Let $\mathbf{A}$ be set of $n$ points and let $\mathbf{C}$ be a set of size $m$ with weights $\Omega$, where $m = o(n)$. Now for $(\mathbf{C}, \Omega)$ to be a *relative error* coreset for non-parametric clustering, it must ensure the following for all $\mathbf{X}$ with at most $n$ rows,

$$|f_{\mathbf{X}}(\mathbf{C}, \Omega) - f_{\mathbf{X}}(\mathbf{A})| \leq \epsilon f_{\mathbf{X}}(\mathbf{A}).$$

However, notice that no matter what the set $\mathbf{C}$ is, the above claim is false when $\mathbf{X} = \mathbf{C}$. This is because for such a query $\mathbf{X}$ the total cost $f_{\mathbf{X}}(\mathbf{C}, \Omega) = 0$, whereas $f_{\mathbf{X}}(\mathbf{A}) \neq 0$ since $m = o(n)$.

## 5.1 Existence of sublinear sized coresets for non-parametric clustering

In this subsection, we show the existence of coreset for non-parametric clustering that ensures an additive error approximation guarantee. The proof for this will proceed via the *probabilistic method*– we will design a set of sampling probabilities that will be used to show the existence of the claimed coreset. The main novelty here is that the sampling scores are **not** the usual sensitivity scores Feldman & Langberg (2011), which are defined for all the data points all at once. In contrast, we define the scores iteratively— for every step $i$, we define a score for the $i^{th}$ point, and this score depends on the coreset maintained by the algorithm till this step, i.e., these scores are also random variables themselves. We will then show a bound on the expected sample size.

At each step $i$ we define two sensitivity scores called *barrier sensitivity scores*. We start by defining barrier sensitivity scores.

**Definition 5.1** (Barrier Sensitivity Scores). *Let $\mathbf{A}_{i-1}$ be the set first set of $(i-1)$ points and $\varphi_i = \sum_{j \leq i} \mathbf{a}_j / i$ be the mean of $\mathbf{A}_i$. Let $(\mathbf{C}_{i-1}, \Omega_{i-1})$ be a weighted subsample of $\mathbf{A}_{i-1}$. At step $i$, the upper barrier sensitivity $s_i^u = \sup_{\mathbf{X}} r_i^u(\mathbf{X})$ and the lower barrier sensitivity $s_i^l = \sup_{\mathbf{X}} r_i^l(\mathbf{X})$, where:*

$$r_i^u(\mathbf{X}) = \frac{f_{\mathbf{X}}(\mathbf{a}_i)}{(1+\epsilon)f_{\mathbf{X}}(\mathbf{A}_{i-1}) - f_{\mathbf{X}}(\mathbf{C}_{i-1}, \Omega_{i-1}) + \epsilon f_{\varphi_i}(\mathbf{A}_i)} \tag{6}$$

$$r_i^l(\mathbf{X}) = \frac{f_{\mathbf{X}}(\mathbf{a}_i)}{f_{\mathbf{X}}(\mathbf{C}_{i-1}, \Omega_{i-1}) - (1-\epsilon)f_{\mathbf{X}}(\mathbf{A}_{i-1}) + \epsilon f_{\varphi_i}(\mathbf{A}_i)} \tag{7}$$

*here* $\sup$ *is over all* $\mathbf{X}$ *with at most $n$ centers.*

Note that unlike the regular sensitivity scores, the above scores are neither guaranteed to be non-negative nor bounded by 1, if the sets $\mathbf{C}_{i-1}$ and weights $\Omega_{i-1}$ are arbitrary. Our algorithm will, however, maintain an invariant (which is effectively the "coreset-property" of $(\mathbf{C}_{i-1}, \Omega_{i-1})$) that guarantees the non-negativity of the scores.

We will sample points based on the upper bounds on the above barrier sensitivity, and show that the resulting set is a coreset for non-parametric clustering. A similar technique has been used in (Cohen et al., 2016) to build coresets for spectral approximation for matrices. In case of general cost functions, especially Bregman divergences, getting nontrivial upper bounds (i.e., bounds less than 1) on $s_i^u$ and $s_i^l$ is very challenging.

In order to show the existence result, we assume that we have access to an oracle that returns a upper bound on these sensitivity scores. Utilizing these upper bounds, we show the existence of coreset for the non-parametric clustering problem. We later show that under some assumption we can get an upper bound on these scores and thereby getting an algorithm that returns a desired coreset the problem.

### 5.1.1 Algorithm Overview:

For every point $\mathbf{a}_i$, we assume that the oracle returns $\tilde{s}_i^u$ and $\tilde{s}_i^l$ such that $\tilde{s}_i^u \geq s_i^u$ and $\tilde{s}_i^l \geq s_i^l$. Define $p_i$ and $\omega_i$ as follows– $p_i = \min\{1, c^u \tilde{s}_i^u + c^l \tilde{s}_i^l\}$, where $c^u = \frac{2}{\epsilon} + 1$ and $c^l = \frac{2}{\epsilon} - 1$. Define $\omega_i = \frac{1}{p_i}$. We sample $\mathbf{a}_i$ to be in the coreset with probability $p_i$ and give it a weight $\omega_i$ if included. Such a sampled set of points are going to ensure the guarantee in the following theorem.

**Theorem 5.1.** *Let $\mathbf{A} \in \mathbb{R}^{n \times d}$, for every Bregman divergence $d_\Phi$ as in table (1) the above sampling technique returns a coreset $(\mathbf{C}, \Omega)$ such that $\forall \mathbf{X}$ with at most $n$ centers it ensures.*

$$\left| f_{\mathbf{X}}(\mathbf{C}, \Omega) - f_{\mathbf{X}}(\mathbf{A}) \right| \leq \epsilon(f_{\mathbf{X}}(\mathbf{A}) + f_\varphi(\mathbf{A})). \tag{8}$$

*The expected size of such a coreset is $O\left(\frac{1}{\mu \epsilon^2}\left(\log n + \log\left(f_\varphi^{\mathbf{M}}(\mathbf{A})\right) - \log\left(f_{\varphi_2}^{\mathbf{M}_2}(\mathbf{a}_2)\right)\right)\right)$.*

Note that the above guarantee equation 8 is deterministic, but the coreset size is a random variable. To prove the above theorem we use the following supporting lemmas. We first show that for each point $\mathbf{a}_i$, if $\tilde{s}_i^u$ and $\tilde{s}_i^l$ upper bound the sensitivity scores $s_i^u$ and $s_i^l$, then the sampled coreset would guarantee that equation 4 holds for all $\mathbf{X}$ with at most $n$ centers.

**Lemma 5.2.** *Suppose the oracle returns $\tilde{s}_i^u$ and $\tilde{s}_i^l$ such that, $\forall i \in [n]$.*

$$\sup_{\mathbf{X}} \frac{f_{\mathbf{X}}(\mathbf{a}_i)}{(1 + \epsilon) f_{\mathbf{X}}(\mathbf{A}_{i-1}) - f_{\mathbf{X}}(\mathbf{C}_{i-1}, \Omega_{i-1}) + \epsilon f_{\varphi_i}(\mathbf{A}_i)} \leq \tilde{s}_i^u$$

$$\sup_{\mathbf{X}} \frac{f_{\mathbf{X}}(\mathbf{a}_i)}{f_{\mathbf{X}}(\mathbf{C}_{i-1}, \Omega_{i-1}) - (1 - \epsilon) f_{\mathbf{X}}(\mathbf{A}_{i-1}) + \epsilon f_{\varphi_i}(\mathbf{A}_i)} \leq \tilde{s}_i^l$$

*Let $\epsilon > 0$. The sampling probability for first point is $p_1 = 1$, and for the $i^{th}$ point is $p_i = \min\{\tilde{s}_i, 1\}$ where $\tilde{s}_i = c^u \tilde{s}_i^u + c^l \tilde{s}_i^l$, such that $c^u = \frac{2}{\epsilon} + 1$ and $c^l = \frac{2}{\epsilon} - 1$.*

*It maintains,*

$$(\mathbf{C}_i, \Omega_i) = \begin{cases} (\mathbf{C}_{i-1}, \Omega_{i-1}) \cup (\mathbf{a}_i, \frac{1}{p_i}) & \text{if sampled,} \\ (\mathbf{C}_{i-1}, \Omega_{i-1}), & \text{else.} \end{cases}$$

*such that $\forall \mathbf{X}$ with at most $n$ centers it guarantees*

$$\left| f_{\mathbf{X}}(\mathbf{C}_i, \Omega_i) - f_{\mathbf{X}}(\mathbf{A}_i) \right| \leq \epsilon(f_{\mathbf{X}}(\mathbf{A}_i) + f_{\varphi_i}(\mathbf{A}_i)).$$

*Proof.* We show this by induction. At $i = 1$ this is true, as we have $p_1 = 1$. So we get,

$$(1 - \epsilon) f_{\mathbf{X}}(\mathbf{a}_1) \leq f_{\mathbf{X}}(\mathbf{c}_1, \omega_1) \leq (1 + \epsilon) f_{\mathbf{X}}(\mathbf{a}_1) \tag{9}$$

where $\mathbf{c}_1 = \mathbf{a}_1$ and $\omega_1 = 1$. Now consider that at $i - 1$ the tuple $(\mathbf{C}_{i-1}, \Omega_{i-1})$ ensures

$$|f_{\mathbf{X}}(\mathbf{A}_{i-1}) - f_{\mathbf{X}}(\mathbf{C}_{i-1}, \Omega_{i-1})| \leq \epsilon(f_{\mathbf{X}}(\mathbf{A}_{i-1}) + f_{\varphi_{i-1}}(\mathbf{A}_{i-1})) \tag{10}$$

We show that $(\mathbf{C}_i, \Omega_i)$ also holds a similar guarantee for $\mathbf{A}_i$. Recall that we need to show that upon creating the sampling probability (and weight) by using an upper bound on the barrier sensitivity scores, no matter whether the random process samples the point $\mathbf{a}_i$ or not, the tuple $(\mathbf{C}_i, \Omega_i)$ will ensure the desired guarantee. Recall that the sampling probability $p_i = \min\{1, c^u \tilde{s}_i^u + c^l \tilde{s}_i^l\}$. We first take the case when $p_i = 1$, and hence $\mathbf{c}_i = \mathbf{a}_i$ and $\omega_i = 1$. So we have,

$$|f_{\mathbf{X}}(\mathbf{A}_{i-1}) - f_{\mathbf{X}}(\mathbf{C}_{i-1}, \Omega_{i-1})| \quad \leq \quad \epsilon(f_{\mathbf{X}}(\mathbf{A}_{i-1}) + f_{\varphi_{i-1}}(\mathbf{A}_{i-1}))$$

$$|f_{\mathbf{X}}(\mathbf{A}_i) - f_{\mathbf{X}}(\mathbf{C}_i, \Omega_i)| \stackrel{(i)}{\leq} \epsilon(f_{\mathbf{X}}(\mathbf{A}_{i-1}) + f_{\varphi_{i-1}}(\mathbf{A}_{i-1}))$$

$$|f_{\mathbf{X}}(\mathbf{A}_i) - f_{\mathbf{X}}(\mathbf{C}_i, \Omega_i)| \stackrel{(ii)}{\leq} \epsilon(f_{\mathbf{X}}(\mathbf{A}_i) + f_{\varphi_i}(\mathbf{A}_i))$$

Here, $(i)$ is true because $f_{\mathbf{X}}(\mathbf{a}_i) = f_{\mathbf{X}}(\mathbf{c}_i, \omega_i)$. In $(ii)$ we only increase the RHS, because $f_{\varphi_i}(\mathbf{A}_i) \geq f_{\varphi_{i-1}}(\mathbf{A}_{i-1})$ (Observation 4.2) and $f_{\mathbf{X}}(\mathbf{a}_i) \geq 0$. So, finally we have,

$$(1-\epsilon)f_{\mathbf{X}}(\mathbf{A}_i) - \epsilon f_{\varphi_i}(\mathbf{A}_i) \leq f_{\mathbf{X}}(\mathbf{C}_i, \Omega_i) \leq (1+\epsilon)f_{\mathbf{X}}(\mathbf{A}_i) + \epsilon f_{\varphi_i}(\mathbf{A}_i)$$

Next when $p_i < 1$ then for the upper bound we use the upper barrier sensitivity definition i.e., $s_i^u$,

$$p_i \geq \tilde{s}_i^u \geq s_i^u$$

Using the definition of $s_i^u$, we have that $\forall \mathbf{X}$,

$$p_i \stackrel{(i)}{\geq} \frac{f_{\mathbf{X}}(\mathbf{a}_i)}{(1+\epsilon)f_{\mathbf{X}}(\mathbf{A}_{i-1}) - f_{\mathbf{X}}(\mathbf{C}_{i-1}, \Omega_{i-1}) + \epsilon f_{\varphi_i}(\mathbf{A}_i)}$$

$$(1+\epsilon)f_{\mathbf{X}}(\mathbf{A}_{i-1}) - f_{\mathbf{X}}(\mathbf{C}_{i-1}, \Omega_{i-1}) + \epsilon f_{\varphi_i}(\mathbf{A}_i) \geq \frac{f_{\mathbf{X}}(\mathbf{a}_i)}{p_i}$$

$$(1+\epsilon)f_{\mathbf{X}}(\mathbf{A}_{i-1}) + \epsilon f_{\varphi_i}(\mathbf{A}_i) \geq f_{\mathbf{X}}(\mathbf{C}_{i-1}, \Omega_{i-1}) + \frac{f_{\mathbf{X}}(\mathbf{a}_i)}{p_i}$$

Now, if $\mathbf{a}_i$ is actually sampled, $f_{\mathbf{X}}(\mathbf{C}_{i-1}, \Omega_{i-1}) + \frac{f_{\mathbf{X}}(\mathbf{a}_i)}{p_i} = f_{\mathbf{X}}(\mathbf{C}_i, \Omega_i)$, and hence,

$$(1+\epsilon)f_{\mathbf{X}}(\mathbf{A}_{i-1}) + \epsilon f_{\varphi_i}(\mathbf{A}_i) \geq f_{\mathbf{X}}(\mathbf{C}_i, \Omega_i)$$
$$(1+\epsilon)f_{\mathbf{X}}(\mathbf{A}_i) + \epsilon f_{\varphi_i}(\mathbf{A}_i) \geq f_{\mathbf{X}}(\mathbf{C}_i, \Omega_i)$$

This shows that the upper bound claim is true with $\mathbf{a}_i$ is sampled in $(\mathbf{C}_i, \Omega_i)$. In case that $p_i < 1$ and $\mathbf{a}_i$ is not actually sampled, we have $f_{\mathbf{X}}(\mathbf{C}_i, \Omega_i) = f_{\mathbf{X}}(\mathbf{C}_{i-1}, \Omega_{i-1})$ and hence follows immediately from above. So, when $p_i < 1$, irrespective of $\mathbf{a}_i$ being sampled in $(\mathbf{C}_i, \Omega_i)$ or not, the desired upper bound claim for $\mathbf{A}_i$ holds.

For the lower bound we use the lower barrier sensitivity definition i.e., $s_i^l$ in a similar manner–

$$1 > \tilde{s}_i^l$$
$$1 > s_i^l$$
$$1 \geq \frac{f_{\mathbf{X}}(\mathbf{a}_i)}{f_{\mathbf{X}}(\mathbf{C}_{i-1}, \Omega_{i-1}) - (1-\epsilon)f_{\mathbf{X}}(\mathbf{A}_{i-1}) + \epsilon f_{\varphi_i}(\mathbf{A}_i)}$$
$$f_{\mathbf{X}}(\mathbf{C}_{i-1}, \Omega_{i-1}) - (1-\epsilon)f_{\mathbf{X}}(\mathbf{A}_{i-1}) + \epsilon f_{\varphi_i}(\mathbf{A}_i) \geq f_{\mathbf{X}}(\mathbf{a}_i)$$
$$f_{\mathbf{X}}(\mathbf{C}_{i-1}, \Omega_{i-1}) \geq (1-\epsilon)f_{\mathbf{X}}(\mathbf{A}_{i-1}) - \epsilon f_{\varphi_i}(\mathbf{A}_i) + f_{\mathbf{X}}(\mathbf{a}_i)$$
$$f_{\mathbf{X}}(\mathbf{C}_{i-1}, \Omega_{i-1}) \geq (1-\epsilon)f_{\mathbf{X}}(\mathbf{A}_i) - \epsilon f_{\varphi_i}(\mathbf{A}_i)$$

When the point $\mathbf{a}_i$ is not sampled in $(\mathbf{C}_i, \Omega_i)$ we have $(\mathbf{C}_{i-1}, \Omega_{i-1}) = (\mathbf{C}_i, \Omega_i)$ and hence we have our inductive step. If $\mathbf{a}_i$ does get sampled than $f_{\mathbf{X}}(\mathbf{C}_i, \Omega_i) \geq f_{\mathbf{X}}(\mathbf{C}_{i-1}, \Omega_{i-1})$. Hence in both the cases we have that

$$f_{\mathbf{X}}(\mathbf{C}_i, \Omega_i) \geq (1-\epsilon)f_{\mathbf{X}}(\mathbf{A}_i) - \epsilon f_{\varphi_i}(\mathbf{A}_i)$$

This completes the proof of both the lower and upper bound for the $i^{th}$ step. □

There are two important points to note from the above lemma.

- Since, the oracle is giving an upper bound on the barrier sensitivity scores, so the coreset guarantees are deterministic. That is when, $p_i < 1$ then irrespective of $\mathbf{a}_i$ being sampled in the $(\mathbf{C}_i, \Omega_i)$ or not, it always ensures the guarantee. It implies that if we run this process multiple times on the same data stream, then each time we get a different coreset but each coreset ensures the desired guarantee with probability 1.

- Due to the barrier sensitivity score based sampling, the coreset guarantee holds $\forall \mathbf{X}$, and we do not require the knowledge of pseudo dimension of the query space. This intuition is similar to the deterministic spectral sparsification claim in (Batson et al., 2012).

Next we need to analyze the expected sample size. Note that, unlike the the online sensitivity scores in `BregmanFilter`, the barrier sensitivity scores themselves are random variables, and depend both on the order of the input stream as well as the previous sampling decisions while maintaining $(\mathbf{C}_{i-1}, \Omega_{i-1})$. Next, we show that the expectation of these scores can be bounded, hence giving a bound on the expected coreset size. We first present a supporting lemma.

**Lemma 5.3.** *Given non-negative scalars $q, r, s, u, v$ and $w$, where $q, r, s$ and $w$ are positive, we define a random variable $t$ as,*

$$
t = \begin{cases} q - u \cdot r & \text{with probability } p, \\ q - v \cdot r & \text{with probability } (1 - p). \end{cases}
$$

*If $\frac{r}{q+w} = 1$ then we get,*

$$
\mathbb{E}\left[ \frac{s}{t+w} - \frac{s}{q+w} \right] = \frac{pu + (1-p)v - uv}{(1-u)(1-v)} \left( \frac{s}{q+w} \right)
$$

The proof is discussed in the appendix (A.2.1). Let $\Pi_{i-1} \in \{0, 1\}^{i-1}$ be the sampling decisions (0 and 1 indicating the not-sampled/sampled decision respectively) that the algorithm made while creating $(\mathbf{C}_{i-1}, \Omega_{i-1})$, based on which the sampling probability of the next point $\mathbf{a}_i$ is being computed. The $i$ coordinates of $\Pi_i$ will be denoted as $\pi_1, \ldots, \pi_i$.

Before bounding the expected barrier sensitivity scores we first show that the at any step $i$, the upper bound on the expected sensitivity barrier score is independent of the sampling choice made by the algorithm in the previous step for the point $\mathbf{a}_{i-1}$. Recall, $c^u = \frac{2}{\epsilon} + 1$ and $c^u = \frac{2}{\epsilon} - 1$, for which we show a helpful lemma.

**Lemma 5.4.** *For any step $j \geq 1$,*

$$
\frac{f_{\mathbf{X}}(\mathbf{a}_i)}{p_{j+1}((\epsilon/2)f_{\mathbf{X}}(\mathbf{A}_{i-1}) + (1 + \epsilon/2)f_{\mathbf{X}}(\mathbf{A}_j) - f_{\mathbf{X}}(\mathbf{C}_j, \Omega_j))} \leq \frac{1}{c^u}
$$

$$
\frac{f_{\mathbf{X}}(\mathbf{a}_i)}{p_{j+1}(f_{\mathbf{X}}(\mathbf{C}_j, \Omega_j)) - (\epsilon/2)f_{\mathbf{X}}(\mathbf{A}_{i-1}) - (1 - \epsilon/2)f_{\mathbf{X}}(\mathbf{A}_j)} \leq \frac{1}{c^l}.
$$

Again the proof is delegated to the appendix (A.2.2). Next we show the following lemma which provides a bound on the expected barrier sensitivity score. The detailed proof is discussed in the appendix (A.2.3).

**Lemma 5.5.** *For points that are coming in a stream, let $i \geq j + 1$, and let $\pi_{j+1}$ denote the sampling choice that the algorithm has made at $(j+1)^{th}$ step. Then we have,*

$$
\mathbb{E}_{\pi_{j+1}} \left[ \frac{f_{\mathbf{X}}(\mathbf{a}_i)}{\gamma_{i-1, j+1}^u + \epsilon f_{\varphi_i}(\mathbf{A}_i)} \right] \leq \frac{f_{\mathbf{X}}(\mathbf{a}_i)}{\gamma_{i-1, j}^u + \epsilon f_{\varphi_i}(\mathbf{A}_i)}
$$

$$
\mathbb{E}_{\pi_{j+1}} \left[ \frac{f_{\mathbf{X}}(\mathbf{a}_i)}{\gamma_{i-1, j+1}^l + \epsilon f_{\varphi_i}(\mathbf{A}_i)} \right] \leq \frac{f_{\mathbf{X}}(\mathbf{a}_i)}{\gamma_{i-1, j}^l + \epsilon f_{\varphi_i}(\mathbf{A}_i)}
$$

In the following lemma we bound the expected barrier sensitivity score with a term independent of any sampling choice made by the algorithm until then. Finally, it yields the expected sample size.

**Lemma 5.6.** *Let $\mathbf{A}$ be a set of $n$ points each in $\mathbb{R}^d$. For point $\mathbf{a}_i$ and for all $\mathbf{X} \in \mathbb{R}^{k \times d}$ we have following bound $\forall i \in [n]$,*

$$
\mathbb{E}_{\Pi_{i-1}} \left[ \frac{f_{\mathbf{X}}(\mathbf{a}_i)}{(1+\epsilon)f_{\mathbf{X}}(\mathbf{A}_{i-1}) - f_{\mathbf{X}}(\mathbf{C}_{i-1}, \Omega_{i-1}) + \epsilon f_{\varphi_i}(\mathbf{A}_i)} \right] \leq \frac{2 f_{\varphi_i}^{\mathbf{M}_i}(\mathbf{a}_i)}{\mu_i \epsilon \sum_{j \leq i} f_{\varphi_j}^{\mathbf{M}_j}(\mathbf{a}_j)} + \frac{12}{\mu_i \epsilon (i-1)}
$$

$$
\mathbb{E}_{\Pi_{i-1}} \left[ \frac{f_{\mathbf{X}}(\mathbf{a}_i)}{f_{\mathbf{X}}(\mathbf{C}_{i-1}, \Omega_{i-1}) - (1-\epsilon)f_{\mathbf{X}}(\mathbf{A}_{i-1}) + \epsilon f_{\varphi_i}(\mathbf{A}_i)} \right] \leq \frac{2 f_{\varphi_i}^{\mathbf{M}_i}(\mathbf{a}_i)}{\mu_i \epsilon \sum_{j \leq i} f_{\varphi_j}^{\mathbf{M}_j}(\mathbf{a}_j)} + \frac{12}{\mu_i \epsilon (i-1)}
$$

*Proof.* Due to lemma 5.5 we show that expected sensitivity score for any $\mathbf{a}_i$ is independent of the sampling choice made by the algorithm for $\mathbf{a}_{i-1}$ point. We show the result for the upper barrier sensitivity score. The analysis for the lower barrier sensitivity score is very similar which can be found in the appendix A.2.4.

Let $(1 + \epsilon)f_{\mathbf{X}}(\mathbf{A}_{i-1}) - f_{\mathbf{X}}(\mathbf{C}_{i-1}, \Omega_{i-1}) = f_{\mathbf{X}}(\mathbf{A}_{i-1}^u)$ where $f_{\mathbf{X}}(\mathbf{A}_{i-1}^u) = \sum_{j \leq i-1} f_{\mathbf{X}}(\mathbf{a}_j^u)$. Here each term $f_{\mathbf{X}}(\mathbf{a}_j^u) = (1 + \epsilon - p_j^{-1})f_{\mathbf{X}}(\mathbf{a}_j)$ if $\mathbf{a}_j$ is present in $\mathbf{C}_{i-1}$ else $f_{\mathbf{X}}(\mathbf{a}_j^u) = (1 + \epsilon)f_{\mathbf{X}}(\mathbf{a}_j)$. Now the expected upper bound on the upper barrier sensitivity score can be bounded as follows.

$$
\begin{aligned}
\mathbb{E}_{\Pi_{i-1}}\left[\frac{f_{\mathbf{X}}(\mathbf{a}_i)}{f_{\mathbf{X}}(\mathbf{A}_{i-1}^u) + \epsilon f_{\varphi_i}(\mathbf{A}_i)}\right] &\overset{(i)}{\leq} \mathbb{E}_{\Pi_{i-1}}\left[\frac{f_{\mathbf{X}}^{\mathbf{M}_i}(\mathbf{a}_i)}{f_{\mathbf{X}}(\mathbf{A}_{i-1}^u) + \epsilon f_{\varphi_i}(\mathbf{A}_i)}\right] \\[2mm]
&\overset{(ii)}{\leq} \mathbb{E}_{\pi_{i-1}}\left[\frac{\left[2f_{\varphi_i}^{\mathbf{M}_i}(\mathbf{a}_i) + \frac{4}{i-1}\sum_{\mathbf{a}_j \in \mathbf{A}_{i-1}}[f_{\varphi_i}^{\mathbf{M}_i}(\mathbf{a}_j) + f_{\mathbf{X}}^{\mathbf{M}_i}(\mathbf{a}_j)]\right]}{(1+\epsilon)f_{\mathbf{X}}(\mathbf{A}_{i-1}) - f_{\mathbf{X}}(\mathbf{C}_{i-1}, \Omega_{i-1}) + \epsilon f_{\varphi_i}(\mathbf{A}_i)}\bigg| \Pi_{i-2}\right] \\[2mm]
&= \mathbb{E}_{\pi_{i-1}}\left[\frac{\left[2f_{\varphi_i}^{\mathbf{M}_i}(\mathbf{a}_i) + \frac{4}{i-1}\sum_{\mathbf{a}_j \in \mathbf{A}_{i-1}} f_{\varphi_i}^{\mathbf{M}_i}(\mathbf{a}_j)\right]}{(1+\epsilon)f_{\mathbf{X}}(\mathbf{A}_{i-1}) - f_{\mathbf{X}}(\mathbf{C}_{i-1}, \Omega_{i-1}) + \epsilon f_{\varphi_i}(\mathbf{A}_i)}\bigg| \Pi_{i-2}\right] \\[2mm]
&\quad + \mathbb{E}_{\pi_{i-1}}\left[\frac{\frac{4}{i-1}f_{\mathbf{X}}^{\mathbf{M}_i}(\mathbf{A}_{i-1})}{(1+\epsilon)f_{\mathbf{X}}(\mathbf{A}_{i-1}) - f_{\mathbf{X}}(\mathbf{C}_{i-1}, \Omega_{i-1}) + \epsilon f_{\varphi_i}(\mathbf{A}_i)}\bigg| \Pi_{i-2}\right] \\[2mm]
&\overset{(iii)}{=} \mathbb{E}_{\pi_{i-1}}\left[\frac{2f_{\varphi_i}^{\mathbf{M}_i}(\mathbf{a}_i) + \frac{4}{i-1}f_{\varphi_i}^{\mathbf{M}_i}(\mathbf{A}_{i-1})}{\gamma_{i-1,i-1}^u + \epsilon f_{\varphi_i}(\mathbf{A}_i)}\bigg| \Pi_{i-2}\right] \\[2mm]
&\quad + \mathbb{E}_{\pi_{i-1}}\left[\frac{\frac{4}{i-1}f_{\mathbf{X}}^{\mathbf{M}_i}(\mathbf{A}_{i-1})}{\gamma_{i-1,i-1}^u + \epsilon f_{\varphi_i}(\mathbf{A}_i)}\bigg| \Pi_{i-2}\right] \\[2mm]
&\overset{(iv)}{\leq} \mathbb{E}_{\pi_{i-2}}\left[\frac{2f_{\varphi_i}^{\mathbf{M}_i}(\mathbf{a}_i) + \frac{4}{i-1}f_{\varphi_i}^{\mathbf{M}_i}(\mathbf{A}_{i-1})}{\gamma_{i-1,i-2}^u + \epsilon f_{\varphi_i}(\mathbf{A}_i)}\bigg| \Pi_{i-3}\right] \\[2mm]
&\quad + \mathbb{E}_{\pi_{i-2}}\left[\frac{\frac{4}{i-1}f_{\mathbf{X}}^{\mathbf{M}_i}(\mathbf{A}_{i-1})}{\gamma_{i-1,i-2}^u + \epsilon f_{\varphi_i}(\mathbf{A}_i)}\bigg| \Pi_{i-3}\right] \\[2mm]
&\overset{(v)}{\leq} \mathbb{E}_{\pi_0}\left[\frac{2f_{\varphi_i}^{\mathbf{M}_i}(\mathbf{a}_i) + \frac{4}{i-1}f_{\varphi_i}^{\mathbf{M}_i}(\mathbf{A}_{i-1})}{\gamma_{i-1,0}^u + \epsilon f_{\varphi_i}(\mathbf{A}_i)}\right] + \mathbb{E}_{\pi_0}\left[\frac{\frac{4}{i-1}f_{\mathbf{X}}^{\mathbf{M}_i}(\mathbf{A}_{i-1})}{\gamma_{i-1,0}^u + \epsilon f_{\varphi_i}(\mathbf{A}_i)}\right] \\[2mm]
&= \frac{2f_{\varphi_i}^{\mathbf{M}_i}(\mathbf{a}_i) + \frac{4}{i-1}f_{\varphi_i}^{\mathbf{M}_i}(\mathbf{A}_{i-1})}{\epsilon/2 f_{\mathbf{X}}(\mathbf{A}_{i-1}) + \epsilon f_{\varphi_i}(\mathbf{A}_i)} + \frac{\frac{4}{(i-1)}f_{\mathbf{X}}^{\mathbf{M}_i}(\mathbf{A}_{i-1})}{\epsilon/2 f_{\mathbf{X}}(\mathbf{A}_{i-1}) + \epsilon f_{\varphi_i}(\mathbf{A}_i)} \\[2mm]
&\overset{(vi)}{\leq} \frac{2f_{\varphi_i}^{\mathbf{M}_i}(\mathbf{a}_i) + \frac{4}{i-1}f_{\varphi_i}^{\mathbf{M}_i}(\mathbf{A}_{i-1})}{\mu_i \epsilon(0.5 f_{\mathbf{X}}^{\mathbf{M}_i}(\mathbf{A}_{i-1}) + f_{\varphi_i}^{\mathbf{M}_i}(\mathbf{A}_i))} + \frac{\frac{4}{(i-1)}f_{\mathbf{X}}^{\mathbf{M}_i}(\mathbf{A}_{i-1})}{\mu_i \epsilon(0.5 f_{\mathbf{X}}^{\mathbf{M}_i}(\mathbf{A}_{i-1}) + f_{\varphi_i}^{\mathbf{M}_i}(\mathbf{A}_i))} \\[2mm]
&\leq \frac{2f_{\varphi_i}^{\mathbf{M}_i}(\mathbf{a}_i)}{\mu_i \epsilon f_{\varphi_i}^{\mathbf{M}_i}(\mathbf{A}_i)} + \frac{4f_{\varphi_i}^{\mathbf{M}_i}(\mathbf{A}_{i-1})}{\mu_i \epsilon(i-1)f_{\varphi_i}^{\mathbf{M}_i}(\mathbf{A}_i)} + \frac{8f_{\mathbf{X}}^{\mathbf{M}_i}(\mathbf{A}_{i-1})}{\mu_i \epsilon(i-1)f_{\mathbf{X}}^{\mathbf{M}_i}(\mathbf{A}_{i-1})} \\[2mm]
&\overset{(vii)}{\leq} \frac{2f_{\varphi_i}^{\mathbf{M}_i}(\mathbf{a}_i)}{\mu_i \epsilon f_{\varphi_i}^{\mathbf{M}_i}(\mathbf{A}_i)} + \frac{4}{\mu_i \epsilon(i-1)} + \frac{8}{\mu_i \epsilon(i-1)} \\[2mm]
&\leq \frac{2f_{\varphi_i}^{\mathbf{M}_i}(\mathbf{a}_i)}{\mu_i \epsilon \sum_{j \leq i} f_{\varphi_j}^{\mathbf{M}_j}(\mathbf{a}_j)} + \frac{12}{\mu_i \epsilon(i-1)}
\end{aligned}
$$

The inequality $(i)$ is by upper bounding Bregman divergence by squared Mahalanobis distance. The inequality $(ii)$ is due to applying triangle inequality on the numerator, $(a^2 + b^2) \leq 2(a^2 + b^2)$. The $(iii)$ equality is by replacing the denominator with the above definition. The $(iv)$ inequality is by applying the supporting Lemma 5.3. By recursively applying Lemma 5.3 we get the inequality $(v)$ which is independent of the random choices made by the algorithm. The inequality $(vi)$ is by using the lower bound on the denominator. The inequality $(vii)$ an upper bound on the second and the third term. In the final inequality we use the fact that for any $\mu$ similar Bregman divergence from Ackermann & Blömer (2009); Lucic et al. (2016) we

have $\mathbf{M}_j \preceq \mathbf{M}_i$ for $j \leq i$ (Lemma 4.1). Further by the property of Bregman divergence we know that $f_{\varphi_i}(\mathbf{A}_{i-1}) \geq f_{\varphi_{i-1}}(\mathbf{A}_{i-1})$. Hence we have $f_{\varphi_i}^{\mathbf{M}_i}(\mathbf{A}_i) \geq \sum_{j \leq i} f_{\varphi_j}^{\mathbf{M}_j}(\mathbf{a}_j)$. Notice that the above analysis is also true for all $\mathbf{X}$. Hence we have this upper bound for all $\mathbf{X}$ with at most $n$ centers. So we have the,

$$\mathbb{E}_{\Pi_{i-1}}\left[\frac{f_{\mathbf{X}}(\mathbf{a}_i)}{(1+\epsilon)f_{\mathbf{X}}(\mathbf{A}_{i-1}) - f_{\mathbf{X}}(\mathbf{C}_{i-1}, \Omega_{i-1}) + f_{\varphi_i}(\mathbf{A}_i)}\right] \leq \frac{2f_{\varphi_i}(\mathbf{a}_i)}{\epsilon \sum_{j \leq i} f_{\varphi_j}(\mathbf{a}_j)} + \frac{12}{\epsilon(i-1)}$$

$\square$

Note that the analysis holds for all $\mathbf{X}$. Hence we get the expected upper bound on both the barrier sensitivity scores. Further, these upper bounds are independent of $k$, which is number of centers in $\mathbf{X}$ or any bicreteria approximation that usually depends on $k$. Now as the oracle returns a tight upper bound on the sensitivity scores which is used to compute the sampling probability of a streaming point, so we can comment about the expected sample size by bounding the sum of upper bounds of barrier sensitivity scores.

**Lemma 5.7.** *The algorithm returns* $(\mathbf{C}, \Omega)$ *such that the expected size of* $\mathbf{C}$ *is* $O\left(\frac{1}{\mu\epsilon^2}\left(\log n + \log\left(f_{\varphi}^{\mathbf{M}}(\mathbf{A})\right) - \log\left(f_{\varphi_2}^{\mathbf{M}_2}(\mathbf{a}_2)\right)\right)\right)$.

The detailed proof is discussed in the appendix (A.2.5). Now we are ready to prove our main theorem.

*Proof. of Theorem 5.1.* Using lemma 5.2 and 5.6 we can now show that the any coreset created by the algorithm satisfies the guarantees in Theorem 5.1. Since the coreset guarantees hold deterministically, there must exist a coreset with size less than the expected bound that satisfies the guarantees. This completes the existential guarantee stated by Theorem 5.1. $\square$

## 5.2 Coreset algorithm for non-parametric clustering based on empirical sensitivity

Now as it is not known how to get an oracle that upper bounds the barrier sensitivity scores, we present a heuristic which works well in practice. We estimate the sensitivity scores using sampling, and call these *empirical sensitivity scores.*

We first give an example to show that the sampling probabilities in the online setting need to be upper bounds to the barrier sensitivity scores.

**Example 5.1.** *Fix any* $\epsilon \in (0, 1)$. *Consider a simple setup where the first point is* $\mathbf{a} \in \mathbb{R}^d$. *Now the algorithm samples the point in the coreset with probability* 1.

*Suppose the second point is* $\mathbf{b} \in \mathbb{R}^d$ *and suppose that the upper barrier sensitivity for it, achieved by the query* $\mathbf{X}^*$, *equals* $s$. *That is,*

$$s = \sup_{\mathbf{X}} r_2^u(\mathbf{X}) = r_2^u(\mathbf{X}^*)$$

*Now, suppose we sample b with some probability* $p < s$. *Thus,*

$$p < r_2^u(\mathbf{X}^*) = \frac{f_{\mathbf{X}^*}(\mathbf{b})}{(1+\epsilon)f_{\mathbf{X}^*}(\mathbf{a}) - f_{\mathbf{X}^*}(\mathbf{a}) + \epsilon(f_{\varphi}(\mathbf{a}) + f_{\varphi}(\mathbf{b}))}$$

$$\epsilon f_{\mathbf{X}^*}(\mathbf{a}) + \epsilon(f_{\varphi}(\mathbf{a}) + f_{\varphi}(\mathbf{b})) < \frac{f_{\mathbf{X}^*}(\mathbf{b})}{p}$$

$$\epsilon f_{\mathbf{X}^*}(\mathbf{a}) + \epsilon(f_{\varphi}(\mathbf{a}) + f_{\varphi}(\mathbf{b})) < f_{\mathbf{X}^*}(\mathbf{a}) + \frac{f_{\mathbf{X}^*}(\mathbf{b})}{p}.,$$

*which violates the coreset property. Hence, when we sample the second point, which happens with probability* $p$, *we do not have a coreset with the desired guarantee.*

We next present an example which shows the difficulty of estimating the barrier sensitivity scores by sampling queries. This will motivate us to make some assumptions about our data.

**Example 5.2.** *For a fixed point* $\mathbf{a}_i$ *and a query space, let* $r_i^u(\mathbf{X})$ *be a random variable defined as equation 6, where* $\mathbf{X}$ *is chosen randomly from the query space. Consider that we do not have access to the sensitivity score* $s_i = \sup_{\mathbf{X}} r_i^u(\mathbf{X})$. *Our proposed algorithm is to sample a random set of queries* $\mathcal{Y}$ *uniformly at random such that the maximum value of* $r_i^u(\cdot)$ *in* $\mathcal{Y}$ *well approximates* $s_i$.

*Define the empirical upper sensitivity score to be* $\tilde{s}_i = \max_{\mathbf{X} \in \mathcal{Y}} r_i^u(\mathbf{X})$. *Note that if* $\mathbf{X}^*$ *belongs to* $\mathcal{Y}$, *then* $\tilde{s}_i = s_i$, *but this is a very low probability event in general. Consider the set* $\mathcal{B} = \{\mathbf{X} \mid r_i^u(\mathbf{X}) \geq s_i/K\}$ *for some* $K > 1$. *If the sampled queries do not contain any element in* $\mathcal{B}$, *then* $\tilde{s}_i < s_i/K$. *Thus the probability that this sampling based algorithm obtains an estimate* $\tilde{s}_i$ *that satisfies* $\tilde{s}_i \geq s_i/K$ *is exactly the same as the probability mass of the set* $\mathcal{B}$. *Note that the first example shows that with* $s_i = 1$, *if* $\tilde{s}_i < 1/K$, *then using* $K\tilde{s}_i$ *as sampling probabilities, we fail to have a coreset.*

From this we conclude that, in order to use an empirically obtained upper bounds on the sensitivity scores (called empirical sensitivity scores), we need additional assumptions to show that we get a valid coreset with desired guarantees. We now state the assumption, originally in (Baykal et al., 2018).

Let $\mathcal{X}$ represent the query space where each query $\mathbf{X} \in \mathcal{X}$ has at most $n$ centers. Now, for each point $\mathbf{a}_i$ and $\mathcal{X}$, we consider the following two random variables, $r_i^u(\mathbf{X})$ and $r_i^l(\mathbf{X})$ defined as follows,

$$r_i^u(\mathbf{X}) = \frac{f_{\mathbf{X}}(\mathbf{a}_i)}{(1+\epsilon)f_{\mathbf{X}}(\mathbf{A}_{i-1}) - f_{\mathbf{X}}(\mathbf{C}_{i-1}, \Omega_{i-1}) + \epsilon f_{\varphi_i}(\mathbf{A}_i)}$$

$$r_i^l(\mathbf{X}) = \frac{f_{\mathbf{X}}(\mathbf{a}_i)}{f_{\mathbf{X}}(\mathbf{C}_{i-1}, \Omega_{i-1}) - (1-\epsilon)f_{\mathbf{X}}(\mathbf{A}_{i-1}) + \epsilon f_{\varphi_i}(\mathbf{A}_i)},$$

where the randomness is over $\mathbf{X} \in \mathcal{X}$. We assume that the CDF of both $r_i^u(\cdot)$ and $r_i^l(\cdot)$ are bounded. The assumption stated below is for $r_i^u(\cdot)$. A similar assumption is also considered for $r_i^l(\cdot)$.

**Assumption 5.1.** *There is a pair of universal constants* $K$ *and* $K'$ *such that for each* $i \in [n]$, *the CDF of the random variable* $r_i^u(\mathbf{X})$ *for* $\mathbf{X} \in \mathcal{X}$ *denoted by* $G_i()$ *satisfies,*

$$G_i(x^*/K) \leq \exp(-1/K')$$

*where* $x^* = \min\{y \in [0,1] : G_i(y) = 1\}$.

Further, we consider the above assumption is true for all $\mathbf{a}_i \in \mathbf{A}$. Now the following two lemmas are similar to lemma 6 and 7 in (Baykal et al., 2018), using which we get the upper bounds on the sensitivity scores. Here we state them for completeness. Lemma 5.8 is stated for all $i \leq n$.

**Lemma 5.8.** *Let* $K, K' > 0$ *be universal constants and let* $\mathcal{X}$ *be the query space as defined above with CDF* $G_i(\cdot)$ *satisfying the assumption 5.1. Let* $\mathcal{Y} = \{\mathbf{X}_1, \mathbf{X}_2, \ldots, \mathbf{X}_m\}$ *be a set of* $m$ *i.i.d. samples each drawn from* $\mathcal{X}$. *Let* $\mathbf{X}_{m+1}$ *be an i.i.d. sample from* $\mathcal{X}$ *then,*

$$\mathbb{P}\big(K \max_{\mathbf{X} \in \mathcal{Y}} r_i^u(\mathbf{X}) \leq r_i^u(\mathbf{X}_{m+1})\big) \leq \exp(-m/K') \mathbb{P}\big(K \max_{\mathbf{X} \in \mathcal{Y}} r_i^l(\mathbf{X}) \leq r_i^l(\mathbf{X}_{m+1})\big) \leq \exp(-m/K')$$

*Proof.* Let $\mathbf{X}_{\max} = \arg\max_{\mathbf{X} \in \mathcal{Y}_j} r_i^u(\mathbf{X})$, then

$$
\begin{aligned}
\mathbb{P}(K \max_{\mathbf{X} \in \mathcal{Y}} r_i^u(\mathbf{X}) \leq r_i^u(\mathbf{X}_{m+1})) &= \int_0^{x^*} \mathbb{P}(r_i^u(\mathbf{X}_{\max}) \leq y/K | r_i^u(\mathbf{X}_{m+1}) = y) d\mathbb{P}(y) \\
&\overset{(i)}{=} \int_0^{x^*} \mathbb{P}(r_i^u(\mathbf{X}_{\max}) \leq y/K)^m d\mathbb{P}(y) \\
&\leq \int_0^{x^*} G_i(y/K)^m d\mathbb{P}(y) \\
&\overset{(ii)}{\leq} G_i(x^*/K)^m \int_0^{x^*} d\mathbb{P}(y) \\
&= G_i(x^*/K)^m \\
&\leq \exp(-m/K')
\end{aligned}
$$

Here $(i)$ is because $\{\mathbf{X}_1, \mathbf{X}_2, \ldots, \mathbf{X}_m\}$ are i.i.d. from $\mathcal{Y}$. Further $(ii)$ is due to the assumption 5.1. $\qquad \square$

---

**Algorithm 2** `NonParametricFilter`

---

**Require:** $\mathbf{a}_i, i = 1, \ldots n; t > 1; \epsilon \in (0,1); \mathcal{Y} = \{\mathbf{X}_1, \mathbf{X}_2, \ldots, \mathbf{X}_{O(\log(n/\delta))}\}$
**Ensure:** $(\mathbf{C}, \Omega)$

$\quad c^u = 2/\epsilon + 1; c^l = 2/\epsilon - 1; \varphi_0 = \emptyset; S = 0; \mathbf{C}_0^1 = \ldots = \Omega_0^t = \emptyset$
$\quad \lambda = \|\mathbf{a}_1\|_{\min}; \quad \nu = \|\mathbf{a}_1\|_{\max}$
$\quad \textbf{while } i \leq n \textbf{ do}$
$\quad\quad \lambda = \min\{\lambda, \|\mathbf{a}_i\|_{\min}\}; \nu = \max\{\nu, \|\mathbf{a}_i\|_{\max}\}$
$\quad\quad \text{Update } \mathbf{M}_i; \; \mu_i = \lambda/\nu$
$\quad\quad \varphi_i = ((i-1)\varphi_{i-1} + \mathbf{a}_i)/i; S = S + f_{\varphi_i}^{\mathbf{M}_i}(\mathbf{a}_i)$
$\quad\quad \textbf{if } i = 1 \textbf{ then}$
$\quad\quad\quad p_i = 1$
$\quad\quad \textbf{else}$
$\quad\quad\quad \tilde{r}_i^u = \max_{\mathbf{X} \in \mathcal{Y}} r_i^u(\mathbf{X})$
$\quad\quad\quad \tilde{r}_i^l = \max_{\mathbf{X} \in \mathcal{Y}} r_i^l(\mathbf{X})$
$\quad\quad\quad p_i = \min\{1, (c^u \tilde{r}_i^u + c^l \tilde{r}_i^l\}$
$\quad\quad \textbf{end if}$
$\quad\quad (\mathbf{c}_i, \omega_i) = \begin{cases} (\mathbf{a}_i, 1/(p_i)) & \text{if } \mathbf{a}_i \text{ is sampled} \\ (\emptyset, 0) & \text{else} \end{cases}$
$\quad\quad (\mathbf{C}_i, \Omega_i) = (\mathbf{C}_{i-1}, \Omega_{i-1}) \cup (\mathbf{c}_i, \omega_i)$
$\quad \textbf{end while}$
$\quad \text{Return } (\mathbf{C}, \Omega)$

---

Similarly, it is also proved for $r_i^l()$. Let there is a finite set $\mathcal{Y} \subset \mathcal{X}$ from which we get *empirical sensitivity scores* $\tilde{r}_i^u = \max_{\mathbf{X} \in \mathcal{Y}} r_i^u(\mathbf{X})$ and $\tilde{r}_i^l = \max_{\mathbf{X} \in \mathcal{Y}} r_i^l(\mathbf{X})$. Our algorithm uses these scores. Now with the following lemma we establish that empirical sensitivity scores are good approximations to the true sensitivity scores $s_i^u$ and $s_i^l$. It is also used to decide the size of the finite set $\mathcal{Y}$.

**Lemma 5.9.** *Let $\delta \in (0,1)$, consider the set $\mathcal{Y} \subset \mathcal{X}$ of size $|\mathcal{Y}| \geq \lceil K' \log(n/\delta) \rceil$, then*

$$\mathbb{P}_{\mathbf{X} \in \mathcal{X}} \big( \exists i \in [n] : K\tilde{r}_i^u \leq r_i^u(\mathbf{X}) \big) \leq \delta$$
$$\mathbb{P}_{\mathbf{X} \in \mathcal{X}} \big( \exists i \in [n] : K\tilde{r}_i^l \leq r_i^l(\mathbf{X}) \big) \leq \delta$$

*Proof.* The proof follows from lemma 5.8. Let the event $\mathcal{E}_i$ be the event that $K \cdot \max_{\mathbf{X}' \in \mathcal{Y}} r_i^u(\mathbf{X}') \leq r_i^u(\mathbf{X})$. Now,

$$\mathbb{P}(\mathcal{E}_i) = \mathbb{P}_{\mathbf{X} \in \mathcal{X}} (K \max_{\mathbf{X}' \in \mathcal{Y}} r_i^u(\mathbf{X}') \leq r_i^u(\mathbf{X})) \leq \exp(-|\mathcal{Y}|/K')$$

Hence, by taking union bound over all $i \in [n]$, we have that $\mathbb{P}[\neg(\cup_i \mathcal{E}_i)] \geq 1 - n\exp(-|\mathcal{Y}|/K')$. By choosing $|\mathcal{Y}| \geq \lceil K' \log(n/\delta) \rceil$, we get that $\mathbb{P}[\neg(\cup_i \mathcal{E}_i)] \geq 1 - \delta$. Hence with probability at least $1 - \delta$, the score $\tilde{r}_i^u$ upper bounds the true score. We can show a similar claim for $\tilde{r}_i^l$.

$\square$

So with high probability we have an upper bound on $(s_i^u, s_i^l)$ using empirical sensitivity scores $(\tilde{r}_i^u, \tilde{r}_i^l)$. Although the above two lemmas are stated for upper barrier sensitivity scores, they are also true for lower barrier sensitivity scores. Now we present our second algorithm (2) called `NonParametricFilter`, which returns a coreset for non-parametric clustering via Bregman divergence. Note that without the above assumption 5.1 our algorithm acts as a heuristic. The algorithm requires the query set $\mathcal{Y}$ which has $O(\log(n/\delta))$ queries.

The coreset from the above algorithm ensures the guarantee.

**Theorem 5.10.** *Let $\mathbf{A} \in \mathbb{R}^{n \times d}$, for every Bregman divergence $d_\Phi$ as in table (1) `NonParametricFilter` returns a coreset $(\mathbf{C}, \Omega)$ for non parametric clustering based on $d_\Phi$ such that $\forall \mathbf{X}$ with at most $n$ centres in $\mathbb{R}^d$ it ensures equation 1 with at least $1 - \delta$ probability.*

$$\big| f_{\mathbf{X}}(\mathbf{C}) - f_{\mathbf{X}}(\mathbf{A}) \big| \leq \epsilon(f_{\mathbf{X}}(\mathbf{A}) + f_\varphi(\mathbf{A})). \tag{11}$$

*The size of such a coreset is $O\left(\frac{1}{\mu\epsilon^2}\left(\log n + \log\left(f_\varphi^\mathbf{M}(\mathbf{A})\right) - \log\left(f_{\varphi_2}^{\mathbf{M}_2}(\mathbf{a}_2)\right)\right)\right)$ expected samples.*

*Proof.* This is proved by combining the claims of Theorem 5.1 and Lemma 5.9. □

Again the additive error factor can be further improved using an user defined parameter $\tau \in (0,1)$. In the barrier sensitivity scores equation 6 and equation 7 we multiply $\tau$ with the additive term, i.e., $\epsilon f_{\varphi_i}(A_i)$ is replaced with $\tau\epsilon f_{\varphi_i}(A_i)$. We get the following corollary.

**Corollary 5.1.** *Let $\mathbf{A} \in \mathbb{R}^{n\times d}$, for every Bregman divergence $d_\Phi$ as in table (1) `NonParametricFilter` returns a coreset $(\mathbf{C}, \Omega)$ for non parametric clustering based on $d_\Phi$ such that $\forall \mathbf{X}$ with at most $n$ centres in $\mathbb{R}^d$ it ensures equation 1 with at least $1 - \delta$ probability.*

$$\left|f_\mathbf{X}(\mathbf{C}) - f_\mathbf{X}(\mathbf{A})\right| \le \epsilon f_\mathbf{X}(\mathbf{A}) + \epsilon' f_\varphi(\mathbf{A}), \tag{12}$$

*where $\epsilon' = \epsilon \cdot \tau$. The size of such a coreset is $O\left(\frac{1}{\mu\epsilon^2}\left(\log n + \frac{\log\left(f_\varphi^\mathbf{M}(\mathbf{A})\right) - \log\left(f_{\varphi_2}^{\mathbf{M}_2}(\mathbf{a}_2)\right)}{\tau}\right)\right)$ expected samples.*

This can be proved by a similar analysis as of Corollary 4.1.

### 5.2.1 Non-Parametric Clustering via Squared Euclidean Distance

As `NonParametricFilter` returns a coreset for non-parametric clustering, hence it can also be used for $k$-means clustering. In this case our algorithm (2) returns a coreset $(\mathbf{C}, \Omega)$, for squared euclidean Bregman divergence. The algorithm simply uses $\mathbf{I}_d$ as $\mathbf{M}_i$ and 1 as $\mu_i$. In the following corollary we state the guarantees that `NonParametricFilter` ensures for $k$-means clustering that follows from Theorem 5.1.

**Corollary 5.2.** *Let $\mathbf{A} \in \mathbb{R}^{n\times d}$ be the points fed to `NonParametricFilter` for the $k$-means problem. It returns a coreset $(\mathbf{C}, \Omega)$ which ensures the guarantee equation 1 for any $\mathbf{X}$ with at most $n$ centers. The coresets has $O\left(\frac{1}{\epsilon^2}\left(\log n + \log\left(f_\varphi(\mathbf{A})\right) - \log\left(f_{\varphi_2}(\mathbf{a}_2)\right)\right)\right)$ expected samples.*

This directly follows from Theorem 5.1, and discussed in the appendix A.2.6.

### 5.2.2 Coresets for DP-Means Clustering

DP-Means clustering is a non-parametric clustering. Here we discuss how `NonParametricFilter` can also be used to approximate DP-Means clustering Bachem et al. (2015). The problem was originally defined for squared euclidean distances which has a valid generalization in other Bregman divergences. For an input $\mathbf{A}$, query $\mathbf{X}$ with at most $n$ cost of DP-Means cost is defined as follows,

$$cost_{DP}(\mathbf{A}, \mathbf{X}) = f_\mathbf{X}(\mathbf{A}) + \lambda|\mathbf{X}|.$$

Here $f_\mathbf{X}(\mathbf{A})$ is the cost on the entire $\mathbf{A}$ based on some Bregman divergences introduced earlier. It is not difficult to see that the coreset from `NonParametricFilter` ensures an additive error approximation for this definition of DP-Means clustering based on $\mu$-similar Bregman divergence. Now we claim that by allowing a small additive error approximation our coreset size significantly improves upon the coresets for relative error approximation for DP-Means clustering Bachem et al. (2015), as in practice $O(d^d) \gg O(\log n)$. We first the state the following result.

**Lemma 5.11.** *The coreset $(\mathbf{C}, \Omega)$ from `NonParametricFilter` ensures the following for all $\mathbf{X}$ with at most $n$ centers in $\mathbb{R}^d$,*

$$|cost_{DP}(\mathbf{C}, \mathbf{X}) - cost_{DP}(\mathbf{A}, \mathbf{X})| \le \epsilon(f_\mathbf{X}(\mathbf{A}) + f_\varphi(\mathbf{A}))$$

*Proof.* Notice that if one applies DP-Means on the coreset from `NonParametricFilter` we get the following,

$$\left|\text{cost}_{DP}(\mathbf{C}, \mathbf{X}) - \text{cost}_{DP}(\mathbf{A}, \mathbf{X})\right| = \left|f_\mathbf{X}(\mathbf{C}) - f_\mathbf{X}(\mathbf{A})\right| \le \epsilon(f_\mathbf{X}(\mathbf{A}) + f_\varphi(\mathbf{A}))$$

The last inequality is by Theorem 5.1. □

Now due to Lemma 5.11 we have the following Theorem.

**Theorem 5.12.** *For $\epsilon \in (0,1)$ the coreset $(\mathbf{C}, \Omega)$ ensures, $cost_{DP}(\mathbf{A}, \mathbf{X_C}) \le cost_{DP}(\mathbf{A}, \mathbf{X_A}) + \epsilon(f_{\mathbf{X_C}}(\mathbf{A}) + f_{\mathbf{X_A}}(\mathbf{A}) + 2f_{\varphi}(\mathbf{A}))$. with high probability, where $\mathbf{X_C}$ and $\mathbf{X_A}$ are the optimal cluster centers for the DP-Means clustering on $(\mathbf{C}, \Omega)$ and $\mathbf{A}$. The expected size of the coreset is $O\left(\frac{1}{\mu \epsilon^2}\left(\log n + \log\left(f_{\varphi}^{\mathbf{M}}(\mathbf{A})\right) - \log\left(f_{\varphi_2}^{\mathbf{M_2}}(\mathbf{a_2})\right)\right)\right)$.*

*Proof.* Let $\mathbf{X_C}$ and $\mathbf{X_A}$ are the optimal centers for DP-Means clustering on $\mathbf{C}$ and $\mathbf{A}$ respectively. Now we know that,

$$
\begin{aligned}
cost_{DP}(\mathbf{A}, \mathbf{X_C}) - \epsilon(f_{\mathbf{X_C}}(\mathbf{A}) + f_{\varphi}(\mathbf{A})) &\le cost_{DP}(\mathbf{C}, \mathbf{X_C}) \\
&\le cost_{DP}(\mathbf{C}, \mathbf{X_A}) \\
&\le cost_{DP}(\mathbf{A}, \mathbf{X_A}) + \epsilon(f_{\mathbf{X_A}}(\mathbf{A}) + f_{\varphi}(\mathbf{A})) \\
cost_{DP}(\mathbf{A}, \mathbf{X_C}) &\le cost_{DP}(\mathbf{A}, \mathbf{X_A}) + \epsilon(f_{\mathbf{X_C}}(\mathbf{A}) + f_{\mathbf{X_A}}(\mathbf{A}) + 2f_{\varphi}(\mathbf{A})).
\end{aligned}
$$

$\square$

# 6 Uniform Deviation

Our first result showed an additive error coreset for Bregman clustering. It is a natural question to ask whether an additive error that is a function of $f_{\varphi}(\mathbf{A})$ is useful. In this section we provide evidence that when looking at Bregman clustering from the generalization perspective, the generalization error obtained is also a function of $f_{\varphi}(\mathbf{A})$. This implies that clustering by using coresets of the training data (rather than the full data) does not increase the generalization gap.

For a given distribution of input sets and a learnt model, the uniform deviation is the difference between the expected loss and the average empirical loss over a set of samples from the distribution of inputs. It is used to understand the generalization error by a trained model. In this section we show that for clustering based on any $\mu$-similar Bregman divergence the uniform deviation is bounded. For a distribution $\mathcal{D}$ over an input space in $\mathbb{R}^d$ and some $\mu$-similar Bregman divergence (Table 1), let $\xi = \mathbb{E}_{\mathbf{a} \in \mathcal{D}}[\mathbf{a}]$ and $\sigma^2 = \mathbb{E}_{\mathbf{a} \in \mathcal{D}}[f_{\xi}(\mathbf{a})]$. Bachem et al. (2017a) showed that for $k$-means clustering using Euclidean distance, if $\frac{\mathbb{E}_{\mathbf{a} \in \mathcal{D}}[f_{\xi}(\mathbf{a})^2]}{\sigma^4} \le t < \infty$, then with sufficiently large number of samples $\mathbf{A} = \{\mathbf{a}_1, \dots, \mathbf{a}_m\}$ where each $\mathbf{a}_i$ is an i.i.d. sample from $\mathcal{D}$ and $m = \Omega\left(\frac{tkd\log k}{\epsilon^2}\right)$ we get the following $\forall \mathbf{X} \in \mathbb{R}^{k \times d}$ with a constant probability,

$$\left|\frac{1}{m}\sum_{i=1}^{m} f_{\mathbf{X}}(\mathbf{a}_i) - \mathbb{E}[f_{\mathbf{X}}(\mathbf{a})]\right| \le \epsilon(\sigma^2 + \mathbb{E}[f_{\mathbf{X}}(\mathbf{a})]) \tag{13}$$

This essentially implies that there is an additive error approximation on the generalization error by a model trained on a large input set. We show that a similar result is also true for other $\mu$-similar Bregman divergence. The additive term $(\epsilon \cdot \sigma^2)$ in the generalization error is very similar to the additive term $(\epsilon \cdot f_{\varphi}(\mathbf{A}))$ in our coresets guarantees equation 1.

Given the $\mathcal{D}$, the randomness is always over the samples from $\mathcal{D}$. So we use $\mathbb{E}[\cdot]$ instead of $\mathbb{E}_{\mathbf{a} \in \mathcal{D}}[\cdot]$. We consider that for any $\mu$-similar Bregman divergence we know the parameters $\mathbf{M}$ and $\mu$ (1) for $\mathcal{D}$ such that for all $\mathbf{a} \sim \mathcal{D}$ and $\mathbf{x} \in \mathbb{R}^d$, $\mu f_{\mathbf{x}}^{\mathbf{M}}(\mathbf{a}) \le f_{\mathbf{x}}(\mathbf{a}) \le f_{\mathbf{x}}^{\mathbf{M}}(\mathbf{a})$. We also consider the following assumption is true.

$$\frac{\mathbb{E}\left[f_{\xi}^{\mathbf{M}}(\mathbf{a})^2\right]}{\mu^2 \sigma_{\mathbf{M}}^4} + \frac{1}{\mu^2} < \infty \tag{14}$$

Here, $\xi = \mathbb{E}[\mathbf{a}], \sigma_{\mathbf{M}}^2 = \mathbb{E}[(\mathbf{a} - \xi)^T \mathbf{M}(\mathbf{a} - \xi)]$ and for all $\mathbf{a} \sim \mathcal{D}$, $f_{\xi}^{\mathbf{M}}(\mathbf{a}) = (\mathbf{a} - \xi)^T \mathbf{M}(\mathbf{a} - \xi)$. Consider $\mathbf{A} = \{\mathbf{a}_1, \dots, \mathbf{a}_m\}$ be $m$ independent samples from $\mathcal{D}$.

We analyze a family of functions $\mathcal{G}$ mapping from an input space $\mathbb{R}^d$ to $\mathbb{R}_{\ge 0}$ that captures a measure between the cost of a point $\mathbf{a} \sim \mathcal{D}$ and a statistical cost of $\mathcal{D}$. We first state the following lemma, whose proof is discussed in the appendix A.3.1.

**Lemma 6.1.** *Let $k \in \mathbb{N}$. Let $\mathcal{D}$ be a distribution on $\mathbb{R}^d$ with $\mu$ and $\sigma^2$ same as defined above. For a $\mu$-similar Bregman divergence, for any point $\mathbf{a} \sim \mathcal{D}$, and any $\mathbf{X} \in \mathbb{R}^{k \times d}$ we define $g_{\mathbf{X}}(\mathbf{a})$ as,*

$$g_{\mathbf{X}}(\mathbf{a}) = \frac{f_{\mathbf{X}}(\mathbf{a})}{\sigma^2 + \mathbb{E}[f_{\mathbf{X}}(\mathbf{a})]}. \tag{15}$$

*Let $s(\mathbf{a}) = \frac{2f_\xi(\mathbf{a})}{\mu \sigma_{\mathbf{M}}^2} + \frac{8}{\mu}$. Then we have $g_{\mathbf{X}}(\mathbf{a}) \leq s(\mathbf{a})$ for all $\mathbf{X} \in \mathbb{R}^{k \times d}$ and $\mathbb{E}[s(\mathbf{a})^2] = O(t)$.*

For the family of function $\mathcal{G} = \{g_{\mathbf{X}}(\cdot) \mid \mathbf{X} \in \mathbb{R}^{k \times d}\}$, let $\text{Pdim}(\mathcal{G}) \leq \rho < \infty$. Now due to above lemma we can use the main framework of Bachem et al. (2017a) which is stated as follows,

**Theorem 6.2.** *Let $\epsilon \in (0, 1)$ and $\mathcal{G}$ be a family of functions from $\mathbf{a}$ to $\mathbb{R}_{\geq 0}$, where $\mathbf{a} \sim \mathcal{D}$. Let $\rho$ be such that $Pdim(\mathcal{G}) \leq \rho < \infty$ and $t$ be such that $\mathbb{E}[s(\mathbf{a})] = O(t)$. For every $\mathbf{a}$ and $\mathbf{X} \in \mathbb{R}^{k \times d}$, let $s(\mathbf{a})$ be a function such that, $\sup_{\mathbf{X}} g_{\mathbf{X}}(\mathbf{a}) \leq s(\mathbf{a})$. Let $\mathbf{a}_1, \ldots, \mathbf{a}_{2m}$ be $2m$ i.i.d. samples from $\mathcal{D}$ where $m = O\left(\frac{t\rho}{\epsilon^2}\right)$ such that $\frac{4\mathbb{E}[f_\xi^{\mathbf{M}}(\mathbf{a})^2]}{\mu^2 \sigma_{\mathbf{M}}^4} + \frac{96}{\mu^2} \leq t$ and $\mathbb{P}(\frac{1}{2m} \sum_{i=1}^{2m} s(\mathbf{a}_i)^2 > t) \leq 1/4$. Then for all $\mathbf{X} \in \mathbb{R}^{k \times d}$ with at least $0.99$ probability we have,*

$$\left| \frac{1}{m} \sum_{i=1}^{m} f_{\mathbf{X}}(\mathbf{a}_i) - \mathbb{E}[f_{\mathbf{X}}(\mathbf{a})] \right| \leq \epsilon(\sigma^2 + \mathbb{E}[f_{\mathbf{X}}(\mathbf{a})]) \tag{16}$$

The proof of can be found in the appendix A.3.2.

# 7 Experiments

We now show the empirical performance of `BregFilter` (our `BregmanFilter`) as well as a heuristic version of `NonParametricFilter` (`NP-Filter`). Since there are no existing *online* baselines for this problem, we compare it against the *offline* coreset algorithms on real datasets. It is important to note that theoretically our coreset can never perform better than the offline coreset algorithm. Intuitively this simply is because an offline algorithm has more knowledge from the complete data before deciding the sampling probability. Here we investigate whether the performance of the `BregFilter` and the `NP-Filter` matches the performance of the offline algorithms. We compare against the following coreset algorithms:

1) `TwoPass`: It is a two pass algorithm similar to `BregmanFilter`. Here the algorithm knows $\varphi$ from one pass and in the second pass it computes the sampling probability by using $\varphi$ instead of $\varphi_i$ for all $i \in [n]$.

2) `LWCS`: It is the offline lightweight coreset algorithm as stated in Bachem et al. (2015).

3) `RelCoreset`: It is an Offline sampling method that uses bi-criteria approximation Lucic et al. (2016). The resultant coreset ensures relative error approximation.

4) `Leverage`: It is just a heuristic online sampling algorithm. Here we do Online leverage score sampling method as in Chhaya et al. (2020a). Note that leverage score sampling is only guaranteed to preserve the rank of the matrix, not the cluster structure.

5) `Online-K-Means`: This is also a heuristic online sampling algorithm. We run the Algorithm-3 in Liberty et al. (2016) which returns $\tilde{O}(k)$ for $k$ centers. We consider the returned set as our subset of data and compare its performance with our coreset. In order to make this subset meaningful for the comparison we reweigh the selected points with inverse of the sampling probability (just like other sampling techniques). Notice that while the $\tilde{O}(k)$ points returned by the algorithm have a bicriteria approximation guarantee, there is no theoretical guarantee that is available after doing a $k$-clustering of these points.

Please note that except for `Leverage` and `Online-K-Means`, none of the other sampling method qualify as a baseline for comparison with our method `BregFilter`. Further, note that even though any offline sampling method can be treated as a streaming sampling method using merge-and-reduce technique, however as these method allows the algorithm to discard a sample which might have been selected in the coreset at some previous step, so these sampling methods do not qualify as baselines for our online sampling method. As there are very limited baselines, so we compare with other offline methods such as `LWCS` and `RelCoreset`.

We compare the performance on the following datasets:

1) `KDD(BIO-TRAIN)`: $145,751$ points with $74$ features. We consider $k = \{100, 200\}$ and squared Euclidean as the Bregman divergence (see Figure 1). We further consider that the data points are arriving in a streaming fashion. On this dataset we compare the performance of `BregFilter` with other algorithms.

2) `MNIST`: $60,000$ points in $784$ dimension digits dataset. Here we consider $k = \{5, 10, 25, 50\}$ and relative entropy as Bregman divergence (see Figure 2). On this dataset we compare a heuristic version of `NonParametricFilter` with `Uniform` and `TwoPass` sampling algorithms.

Using each of the above described algorithm, we first subsample coresets of different sizes. Once we have the coreset, we run the weighted $k$-means++ Arthur & Vassilvitskii (2007) on them to obtain the centers. We then use these centers and compute the quantization error ($C_s$) on the full data set. We also compute quantization error by running k-means++ on the full data set ($C_f$). Finally we report the *Relative-Error* $\eta = |C_s - C_f|/C_f$.

In the figure 1 the Y-axis represents the relative error $\eta$ and the X-axis represents the expected sample size which is in terms of percentage of the full data. For every expected sample size, we run 10 random instances. Using a parameter that controls the sample size, we ensure that the expected sample size of the 10 random instances are equal. Based on these we compute the average $\eta$ of these 10 events and report it in the plot. The figure shows the change in $\eta$ with the increase in the coreset size for $k = \{100, 200\}$ on `KDD(BIO-TRAIN)` datasets for $k$-means clustering. As the coreset size increases the $\eta$ decreases for all the algorithms. As expected, the offline methods `LWCS` and `RelCoreset` perform relatively better than our `BregFilter`. `BregFilter` clearly outperforms the baseline `Online-K-Means` in terms of relative error at all sample sizes. We also compare with online version of `Leverage` score sampling which, empirically, appears to be competitive with our method. However, recall that the coresets from `Leverage` do not have any provable guarantee and it is not difficult to show a bad input where `Leverage` will fail to return reasonable clusters, e.g. if the data spans a low rank space but has a large number of clusters. We present a toy example in the appendix.

Also note that `Leverage` and `Online-k-Means` have update times of $O(d^2)$ and $O(kd)$ respectively, as opposed to the $O(d)$ update time of `BregFilter`. On `KDD(BIO-TRAIN)` for $k = 100$ and expected coreset size as $1\%$ of the data, the average running time of `BregFilter` is 2 seconds, `Online-k-Means` is 379 seconds and `Leverage` is 5264 seconds.

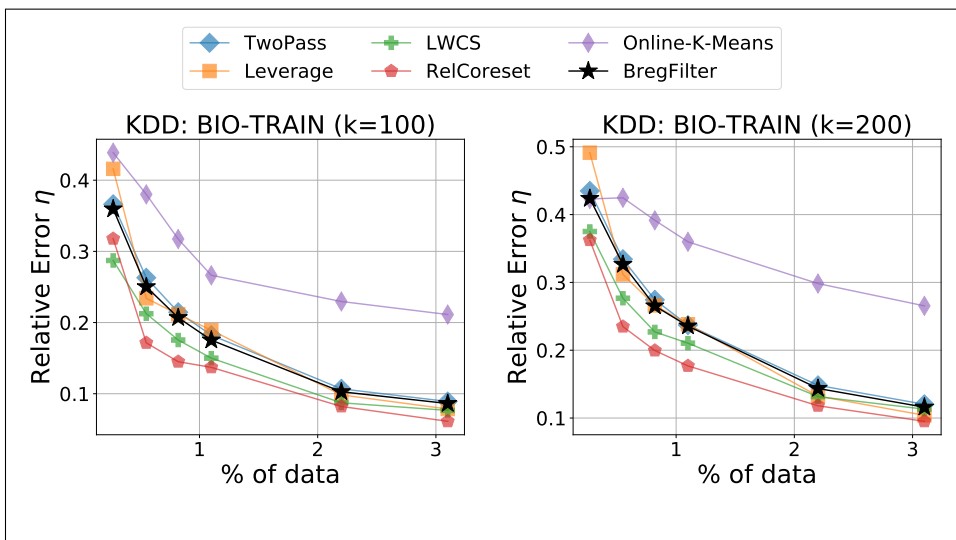

Figure 1: Relative error v/s coreset size for squared Euclidean k-means clustering. We do not hope to beat the offline methods. However, we are at par with them. We are one pass.

For comparing the performance of non-parametric coreset[2]. We run `TwoPass`, `Uniform` and `NP-Filter` on `MNIST`. To the best of our knowledge there are no other baselines to compare with. In `Uniform` we sample each point with probability $r/n$, where $r$ is a parameter used to control the coreset size and $n$ is the number of input points. Now to capture the notion of coreset for non-parametric clustering we run $k$-means++ on every coreset from each method for various values of $k = \{5, 10, 25, 50\}$. Finally we compute the relative error $\eta$ as described above. The computation of empirical sensitivity scores makes the `NonParametricFilter` computationaly expensive. The running time is $\tilde{O}(n^2)$. So we run `NP-Filter` (heuristic version of `NonParametricFilter`) where we use the upper bound of the expected barrier sensitivity scores as in Lemma 5.6 to sample every point, i.e., $\tilde{r}_i^u = \tilde{r}_i^u = \frac{2f_{\varphi_i}^{\mathbf{M}_i}(\mathbf{a}_i)}{\mu_i \epsilon \sum_{j \leq i} f_{\varphi_j}^{\mathbf{M}_j}(\mathbf{a}_j)} + \frac{12}{\mu_i \epsilon(i-1)}$ for all $i \in [n]$. Hence the running time of `NP-Filter` is just $O(nd)$ and the sampling complexity of the returned coreset is controlled by the distortion parameter $\epsilon$. Now for every value of $\epsilon$ we run 5 instances of the algorithms and report the average $\eta$ for every value of $k$. Notice that as we increase $\epsilon$, the $\eta$ also increases. This is due the fact that the coreset size inversely depends on $\epsilon$, so a high $\epsilon$ results to a smaller coreset and as a result it incurs higher $\eta$. It is evident from figure 2 that even our heuristic outperforms the `Uniform` and performs equivalent to `TwoPass`.

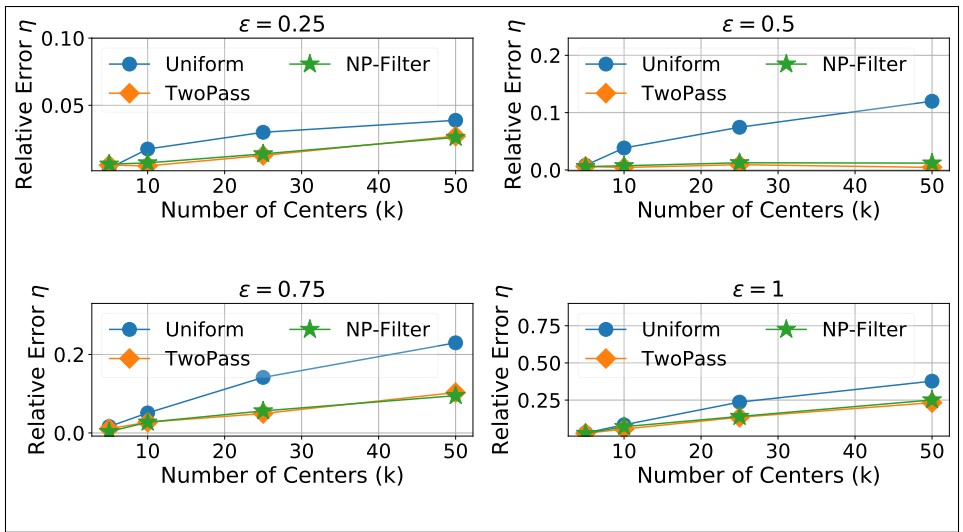

Figure 2: Change in $\eta$ with respect to #centers $k$ for various $\epsilon$.

Please note here we compare the performance in terms of relative error approximation, which is stronger than our actual additive error theoretical guarantees. The plot shows that even with small coreset sizes we get tight relative error approximations and thus supporting the theoretical guarantees.

# 8    Conclusion

Here we presented online coreset for clustering based on Bregman divergences. We also present the first algorithm for non-parametric coreset for the same problem. The algorithm leverages upon additive error approximation, and uses barrier functions and empirical sensitivity scores.

**Broader Impact Statement**

We do not foresee any potential negative impact.

---

[2]coreset for non-parametric clustering.

## Acknowledgments

Anirban would like to acknowledge the following grants and the corresponding funding agencies— Google India Faculty Award, Cisco University grant, SERB-MATRICS grant, SERB-CORE research grant. Supratim acknowledges the generous funding from the European Union's Horizon 2020 research and innovation programmed under grant agreement No 682203 -ERC-[ Inf-Speed-Tradeoff].

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

## A    Appendix

In this paper we use the following theorems in our analysis.

**Theorem A.1** (Bernstein's inequality 2009). *Let the scalar random variables $x_1, x_2, \ldots, x_n$ be independent that satisfy $\forall i \in [n]$, $|x_i - \mathbb{E}[x_i]| \leq b$. Let $X = \sum_i x_i$ and let $\sigma^2 = \sum_i \sigma_i^2$ be the variance of $X$. Then for any $t > 0$,*

$$Pr(X > \mathbb{E}[X] + t) \leq \exp\left(\frac{-t^2}{2\sigma^2 + bt/3}\right)$$

**Theorem A.2** (2020b). *Let $\mathbf{A}$ be the dataset, $\mathbf{X}$ be the query space of dimension $D$, and for $\mathbf{x} \in \mathbf{X}$, let $f_{\mathbf{x}}(\cdot)$ be the cost function. Let $s_j$ be the sensitivity of the $j^{th}$ row of $\mathbf{A}$, and the sum of sensitivities be $S$. Let $(\epsilon, \delta) \in (0, 1)$. Let $r$ be such that*

$$r \geq O\left(\frac{S}{\epsilon^2}(D \log \frac{1}{\epsilon} + \log \frac{1}{\delta})\right)$$

*$\mathbf{C}$ be a matrix of $r$ rows, each sampled i.i.d from $\mathbf{A}$ such that each $\tilde{\mathbf{a}}_i \in \mathbf{C}$ is chosen to be $\mathbf{a}_j$, with weight $\frac{S}{rs_j}$, with probability $\frac{s_j}{S}$, for $j \in [n]$. Then $\mathbf{C}$ is an $\epsilon$-coreset of $\mathbf{A}$ for function $f()$, with probability at least $1 - \delta$.*

We use the above Theorem to bound our coreset size. Note that the Theorem considers a multinomial sample where a point $\tilde{\mathbf{a}}_i$ in coreset $\mathbf{C}$ is $\mathbf{a}_j$ and weight $\frac{S}{rs_j}$ for $j \in [n]$ with probability $\frac{s_j}{S}$. Instead in our approach we get $\tilde{\mathbf{a}}_i$ as $\mathbf{a}_i$, with weight $1/\min\{1, rs_i\}$, with probability $\min\{1, rs_i\}$ or it is $\emptyset$, with weight 0, with probability $1 - \min\{1, rs_i\}$. However, the same Theorem as above applies.

### A.1    Online Coresets for Clustering

Here we discuss the proofs of the supporting lemmas to claim of the main theorem

#### A.1.1    Proof of Lemma 4.5

*Proof.* For some fixed (query) $\mathbf{X} \in \mathbb{R}^{k \times d}$ consider the following random variable.

$$w_i = \begin{cases} (1/p_i - 1)f_{\mathbf{X}}(\mathbf{a}_i) & \text{with probability } p_i \\ -f_{\mathbf{X}}(\mathbf{a}_i) & \text{with probability } (1 - p_i) \end{cases}$$

Note that $\mathbb{E}[w_i] = 0$ and with $p_i = 1$ we get $|w_i| = 0$. The algorithm uses the sampling probability $p_i = \min\{rl_i, 1\}$. Now we bound the term $|w_i|$. In the case when $p_i < 1$ and $\mathbf{a}_i$ is sampled we have,

$$
\begin{aligned}
|w_i| &\leq & \frac{1}{p_i} f_{\mathbf{X}}(\mathbf{a}_i) \\
&=& \frac{f_{\mathbf{X}}(\mathbf{a}_i)}{rl_i} \\
&\overset{(i)}{\leq}& \frac{(f_{\mathbf{X}}(\mathbf{A}_{i-1}) + f_{\varphi_i}(\mathbf{A}_i))f_{\mathbf{X}}(\mathbf{a}_i)}{rf_{\mathbf{X}}(\mathbf{a}_i)} \\
&=& \frac{(f_{\mathbf{X}}(\mathbf{A}_{i-1}) + f_{\varphi_i}(\mathbf{A}_i))}{r} \\
&\leq& \frac{(f_{\mathbf{X}}(\mathbf{A}) + f_{\varphi}(\mathbf{A}))}{r}
\end{aligned}
$$

$(i)$ is by replacing $l_i$ with a smaller term, $\frac{f_{\mathbf{X}}(\mathbf{a}_i)}{f_{\mathbf{X}}(\mathbf{A}_{i-1} + f_{\varphi_i}(\mathbf{A}_i))}$. In the last inequality is because $f_{\varphi}(\mathbf{A}) \geq f_{\varphi_i}(\mathbf{A}_i)$. Here $\varphi = \varphi_n$ is the mean of the entire data $\mathbf{A}$. Next if the point $\mathbf{a}_i$ is not sampled then we know for sure that $p_i < 1$, hence we have that,

$$1 \quad > \quad rl_i$$

$$\geq \frac{r f_{\mathbf{X}}(\mathbf{a}_i)}{(f_{\mathbf{X}}(\mathbf{A}_{i-1}) + f_{\varphi_i}(\mathbf{A}_i))}$$

$$\therefore f_{\mathbf{X}}(\mathbf{a}_i) \leq \frac{(f_{\mathbf{X}}(\mathbf{A}) + f_{\varphi}(\mathbf{A}))}{r}$$

Say, $b = (f_{\mathbf{X}}(\mathbf{A}) + f_{\varphi}(\mathbf{A}))/r$, so $|w_i| \leq b$. Next we bound the $\mathrm{var}(\sum_{i \leq n} w_i) = \sum_{i \leq n} \mathbb{E}[w_i^2]$. Note that for a single term when $p_i < 1$, $\mathbb{E}[w_i^2]$ is,

$$
\begin{aligned}
\mathbb{E}[w_i^2] &= \left(p_i(1/p_i - 1)^2 + (1 - p_i)\right) f_{\mathbf{X}}(\mathbf{a}_i)^2 \\
&\leq \frac{1}{p_i} f_{\mathbf{X}}(\mathbf{a}_i)^2 \\
&= \frac{f_{\mathbf{X}}(\mathbf{a}_i)^2}{r l_i} \\
&\leq \frac{(f_{\mathbf{X}}(\mathbf{A}_{i-1}) + f_{\varphi_i}(\mathbf{A}_i)) f_{\mathbf{X}}(\mathbf{a}_i)^2}{r f_{\mathbf{X}}(\mathbf{a}_i)} \\
&= \frac{(f_{\mathbf{X}}(\mathbf{A}_{i-1}) + f_{\varphi_i}(\mathbf{A}_i)) f_{\mathbf{X}}(\mathbf{a}_i)}{r} \\
&\leq \frac{f_{\mathbf{X}}(\mathbf{a}_i)(f_{\mathbf{X}}(\mathbf{A}) + f_{\varphi}(\mathbf{A}))}{r}
\end{aligned}
$$

So we get,

$$
\begin{aligned}
\mathrm{var}\left(\sum_{i \leq n} w_i\right) &= \sum_{i \leq n} \mathbb{E}[w_i^2] \\
&\leq \sum_{i \leq n} \frac{f_{\mathbf{X}}(\mathbf{a}_i)(f_{\mathbf{X}}(\mathbf{A}) + f_{\varphi}(\mathbf{A}))}{r} \\
&= \frac{f_{\mathbf{X}}(\mathbf{A})(f_{\mathbf{X}}(\mathbf{A}) + f_{\varphi}(\mathbf{A}))}{r} \\
&\leq \frac{(f_{\mathbf{X}}(\mathbf{A}) + f_{\varphi}(\mathbf{A}))^2}{r}
\end{aligned}
$$

Now by applying Bernstein's inequality (A.1) on $\sum_{i \leq n} w_i$ with $t = \epsilon(f_{\mathbf{X}}(\mathbf{A}) + f_{\varphi}(\mathbf{A}))$ we bound the probability $\mathbb{P} = \mathrm{Pr}\left(|f_{\mathbf{X}}(\mathbf{C}, \Omega) - f_{\mathbf{X}}(\mathbf{A})| \geq \epsilon(f_{\mathbf{X}}(\mathbf{A}) + f_{\varphi}(\mathbf{A}))\right)$ as follows,

$$
\begin{aligned}
\mathbb{P} &\leq \exp\left(\frac{-\epsilon^2(f_{\mathbf{X}}(\mathbf{A}) + f_{\varphi}(\mathbf{A}))^2}{\epsilon(f_{\mathbf{X}}(\mathbf{A}) + f_{\varphi}(\mathbf{A}))^2/3r + 2(f_{\mathbf{X}}(\mathbf{A}) + f_{\varphi}(\mathbf{A}))^2/r}\right) \\
&= \exp\left(\frac{-r\epsilon^2}{(\epsilon/3 + 2)}\right)
\end{aligned}
$$

So to get the above event with at least 0.99 probability it is enough to set $r$ to be $O\left(\frac{1}{\epsilon^2}\right)$. Note that the above is guaranteed for a fixed $\mathbf{X} \in \mathbb{R}^{k \times d}$.

Now we show that coreset $(\mathbf{C}, \Omega)$ can be made strong coreset by taking a union bound over a set of queries. We take a union bound over the $\epsilon/2$-net of $\mathbb{R}^{k \times d}$ Woodruff et al. (2014); Lucic et al. (2016). Such a net will have at most $O(\epsilon^{-dk \log k})$ queries. To ensure a strong a coreset guarantee it is enough to set $r$ as $O\left(\frac{dk(\log k)\log(1/\epsilon)}{\epsilon^2}\right)$. $\qquad\square$

**Intuition for** $\log(f_{\mathbf{X}}(\mathbf{A}))$**:** Consider the following input stream — every $i$ that is a multiple of $\sqrt{n}$ has the property that $f_{\mathbf{X}}(\mathbf{a}_i) = f_{\mathbf{X}}(A_{i-1})$, i.e. this point's contribution to the current cost is more than the total contributions of all the previous input points. For all other $j$, $f_{\mathbf{X}}(\mathbf{a}_j)$ is, say, small. Any online algorithm (i.e. one that makes irrevocable decisions without looking at the future) will need to assign a constant probability to every $i$ that is a multiple of $\sqrt{n}$. Hence the resulting coreset size is at least $\sqrt{n}$. Note that $f_{\mathbf{X}}(\mathbf{A}) = O(c^{\sqrt{n}})$, and hence $\log(f_{\mathbf{X}}(\mathbf{A}))$ is a tight bound on the coreset size in this example.

**Lemma A.3.** *For a $\mu$-similar Bregman divergence, for all $i \in [n]$ every incoming points $\mathbf{a}_i$ we have,*

$$\sup_{\mathbf{X} \in \mathbb{R}^{k \times d}} \frac{f_{\mathbf{X}}(\mathbf{a}_i)}{f_{\mathbf{X}}(\mathbf{A}_{i-1}) + \tau f_{\varphi_i}(\mathbf{A}_i)} \leq \frac{2 f_{\varphi_i}^{\mathbf{M}_i}(\mathbf{a}_i)}{\mu_i \tau \sum_{j \leq i} f_{\varphi_j}^{\mathbf{M}_j}(\mathbf{a}_j)} + \frac{8}{\mu_i \tau (i-1)} \tag{17}$$

*Proof.*

$$
\begin{aligned}
\frac{f_{\mathbf{X}}(\mathbf{a}_i)}{f_{\mathbf{X}}(\mathbf{A}_{i-1}) + \tau f_{\varphi_i}(\mathbf{A}_i)} \quad &\overset{(i)}{\leq} \quad \frac{f_{\mathbf{X}}^{\mathbf{M}_i}(\mathbf{a}_i)}{f_{\mathbf{X}}(\mathbf{A}_{i-1}) + \tau f_{\varphi_i}(\mathbf{A}_i)} \\
&\leq \quad \frac{2 f_{\varphi_i}^{\mathbf{M}_i}(\mathbf{a}_i) + 2 f_{\mathbf{X}}^{\mathbf{M}_i}(\varphi_i)}{f_{\mathbf{X}}(\mathbf{A}_{i-1}) + \tau f_{\varphi_i}(\mathbf{A}_i)} \\
&\leq \quad \frac{2 f_{\varphi_i}^{\mathbf{M}_i}(\mathbf{a}_i) + \frac{4}{i-1} f_{\varphi_i}^{\mathbf{M}_i}(\mathbf{A}_{i-1}) + \frac{4}{i-1} f_{\mathbf{X}}^{\mathbf{M}_i}(\mathbf{A}_{i-1})}{f_{\mathbf{X}}(\mathbf{A}_{i-1}) + \tau f_{\varphi_i}(\mathbf{A}_i)} \\
&\overset{(ii)}{\leq} \quad \frac{2 f_{\varphi_i}^{\mathbf{M}_i}(\mathbf{a}_i) + \frac{4}{i-1} f_{\varphi_i}^{\mathbf{M}_i}(\mathbf{A}_{i-1}) + \frac{4}{i-1} f_{\mathbf{X}}^{\mathbf{M}_i}(\mathbf{A}_{i-1})}{\mu_i (f_{\mathbf{X}}^{\mathbf{M}_i}(\mathbf{A}_{i-1}) + \tau f_{\varphi_i}^{\mathbf{M}_i}(\mathbf{A}_i))} \\
&= \quad \frac{2 f_{\varphi_i}^{\mathbf{M}_i}(\mathbf{a}_i) + \frac{4}{i-1} f_{\varphi_i}^{\mathbf{M}_i}(\mathbf{A}_{i-1})}{\mu_i (f_{\mathbf{X}}^{\mathbf{M}_i}(\mathbf{A}_{i-1}) + \tau f_{\varphi_i}^{\mathbf{M}_i}(\mathbf{A}_i))} + \frac{\frac{4}{i-1} f_{\mathbf{X}}^{\mathbf{M}_i}(\mathbf{A}_{i-1})}{\mu_i (f_{\mathbf{X}}^{\mathbf{M}_i}(\mathbf{A}_{i-1}) + \tau f_{\varphi_i}^{\mathbf{M}_i}(\mathbf{A}_i))} \\
&\leq \quad \frac{2 f_{\varphi_i}^{\mathbf{M}_i}(\mathbf{a}_i) + \frac{4}{i-1} f_{\varphi_i}^{\mathbf{M}_i}(\mathbf{A}_{i-1})}{\mu_i (f_{\mathbf{X}}^{\mathbf{M}_i}(\mathbf{A}_{i-1}) + \tau f_{\varphi_i}^{\mathbf{M}_i}(\mathbf{A}_i))} + \frac{4}{\mu_i (i-1)} \\
&\leq \quad \frac{2 f_{\varphi_i}^{\mathbf{M}_i}(\mathbf{a}_i)}{\mu_i \tau f_{\varphi_i}^{\mathbf{M}_i}(\mathbf{A}_i)} + \frac{\frac{4}{i-1} f_{\varphi_i}^{\mathbf{M}_i}(\mathbf{A}_{i-1})}{\mu_i \tau f_{\varphi_i}^{\mathbf{M}_i}(\mathbf{A}_i)} + \frac{4}{\mu_i (i-1)} \\
&\overset{(iii)}{\leq} \quad \frac{2 f_{\varphi_i}^{\mathbf{M}_i}(\mathbf{a}_i)}{\mu_i \tau f_{\varphi_i}^{\mathbf{M}_i}(\mathbf{A}_i)} + \frac{4}{\mu_i (i-1)} (\frac{[}{1}] \tau + 1) \\
&\leq \quad \frac{2 f_{\varphi_i}^{\mathbf{M}_i}(\mathbf{a}_i)}{\mu_i \tau \sum_{j \leq i} f_{\varphi_j}^{\mathbf{M}_j}(\mathbf{a}_j)} + \frac{4}{\mu_i \tau (i-1)} (\frac{[}{1}] \tau + 1)
\end{aligned}
$$

$\square$

Further with an analysis similar to second claim in lemma 4.4 we get $\sum_{i \leq n} l_i \leq 8 \log n + \frac{4 \log \left( f_{\varphi}^{\mathbf{M}}(\mathbf{A}) \right) - 4 \log \left( f_{\varphi_2}^{\mathbf{M}_2}(\mathbf{a}_2) \right)}{\mu \tau}$. Now the algorithm samples point based on its $l_i$ score. Then, applying Bernstein's inequality on sum of all the random variables defined as,

$$
w_i = \begin{cases}
(1/p_i - 1) f_{\mathbf{X}}(\mathbf{a}_i) & \text{with probability } p_i \\
-f_{\mathbf{X}}(\mathbf{a}_i) & \text{with probability } (1 - p_i).
\end{cases}
$$

Finally, by taking a union bound over an $\epsilon$-net we can prove the claim in corollary 4.1.

## A.2 Coresets for Non-Parametric Clustering

### A.2.1 Proof of Lemma 5.3

*Proof.* The proof is fairly straight forward. Using simple algebra (similar to (Sherman & Morrison, 1950)) we have,

$$
\begin{aligned}
\frac{1}{q + w - ur} \quad &= \quad \frac{1}{q + w} + \frac{ur(q + w)^{-2}}{1 - ur(q + w)^{-1}} \\
&= \quad \frac{1}{q + w} + \frac{u}{1 - u}(q + w)^{-1}
\end{aligned}
$$

$$
\begin{aligned}
\frac{1}{q+w-vr} &= \frac{1}{q+w} + \frac{vr(q+w)^{-2}}{1 - vr(q+w)^{-1}} \\
&= \frac{1}{q+w} + \frac{v}{1-v}(q+w)^{-1}
\end{aligned}
$$

So we get,

$$
\mathbb{E}\left[\frac{s}{t+w} - \frac{s}{q+w}\right] = \frac{pu + (1-p)v - uv}{(1-u)(1-v)}\left(\frac{s}{q+w}\right)
$$

$\square$

### A.2.2 Proof of Lemma 5.4

*Proof.* In order to show this we define the following notations. Let $\mathbf{A}_i$ represent the set of $i$ points seen by the algorithm so far. For some $\mathbf{X}$ and for some $j \leq n$, we define two scalars $\zeta_{i,j}^u$ and $\zeta_{i,j}^l$ as follows,

$$
\zeta_{i,j}^u = \frac{\epsilon}{2}f_{\mathbf{X}}(\mathbf{A}_i) + (1 + \frac{\epsilon}{2})f_{\mathbf{X}}(\mathbf{A}_j) \qquad \text{and} \qquad \zeta_{i,j}^l = -\frac{\epsilon}{2}f_{\mathbf{X}}(\mathbf{A}_i) + (1 - \frac{\epsilon}{2})f_{\mathbf{X}}(\mathbf{A}_j)
$$

So we have $\zeta_{i,i}^u = (1+\epsilon)f_{\mathbf{X}}(\mathbf{A}_i)$ and $\zeta_{i,i}^l = (1-\epsilon)f_{\mathbf{X}}(\mathbf{A}_i)$. It is clear that for $j \leq i-1$ we have $\zeta_{i-1,j}^u \geq \zeta_{j,j}^u$ and $\zeta_{i-1,j}^l \leq \zeta_{j,j}^l$. Further two more scalars $\gamma_{i,j}^u$ and $\gamma_{i,j}^l$ are defined as follows,

$$
\gamma_{i,j}^u = \zeta_{i,j}^u - f_{\mathbf{X}}(\mathbf{C}_j, \Omega_j) \qquad \text{and} \qquad \gamma_{i,j}^l = f_{\mathbf{X}}(\mathbf{C}_j, \Omega_j) - \zeta_{i,j}^l
$$

Note that $\gamma_{i,i}^u = (1+\epsilon)f_{\mathbf{X}}(\mathbf{A}_i) - f_{\mathbf{X}}(\mathbf{C}_i, \Omega_i)$ and $\gamma_{i,i}^l = f_{\mathbf{X}}(\mathbf{C}_i, \Omega_i) - (1-\epsilon)f_{\mathbf{X}}(\mathbf{A}_i)$. For $j \leq i-1$ we get $\gamma_{i-1,j}^u \geq \gamma_{j,j}^u$ and $\gamma_{i-1,j}^l \geq \gamma_{j,j}^l$. Let, $d_{j+1} = \frac{f_{\mathbf{X}}(\mathbf{a}_{j+1})}{p_{j+1}}$. If $p_{j+1} < 1$, then we have $p_{j+1} \geq c_u \tilde{s}_{j+1}^u \geq c_u s_{j+1}^u$, and hence we have the following for upper barrier,

$$
p_{j+1} \geq \frac{c^u f_{\mathbf{X}}(\mathbf{a}_{j+1})}{\gamma_{j,j}^u + \epsilon f_{\varphi_{j+1}}(\mathbf{A}_{j+1})} \geq \frac{c^u f_{\mathbf{X}}(\mathbf{a}_{j+1})}{\gamma_{i-1,j}^u + \epsilon f_{\varphi_{j+1}}(\mathbf{A}_{j+1})} \geq \frac{c^u f_{\mathbf{X}}(\mathbf{a}_{j+1})}{\gamma_{i-1,j}^u + \epsilon f_{\varphi_i}(\mathbf{A}_i)}.
$$

Therefore,

$$
\frac{d_{j+1}}{\gamma_{i-1,j}^u + \epsilon f_{\varphi_i}(\mathbf{A}_i)} \leq \frac{1}{c^u}.
$$

Let $\frac{d_{j+1}}{\gamma_{i-1,j}^u + \epsilon f_{\varphi_i}(\mathbf{A}_i)} = h_{j+1}^u$, which is bounded by $\frac{1}{c^u}$. Similarly for the lower barrier we have,

$$
p_{j+1} \geq \frac{c^l f_{\mathbf{X}}(\mathbf{a}_{j+1})}{\gamma_{j,j}^l + \epsilon f_{\varphi_{j+1}}(\mathbf{A}_{j+1})} \geq \frac{c^l f_{\mathbf{X}}(\mathbf{a}_{j+1})}{\gamma_{i-1,j}^l + \epsilon f_{\varphi_{j+1}}(\mathbf{A}_{j+1})} \geq \frac{c^l f_{\mathbf{X}}(\mathbf{a}_{j+1})}{\gamma_{i-1,j}^l + \epsilon f_{\varphi_i}(\mathbf{A}_i)}.
$$

So we get,

$$
\frac{d_{j+1}}{\gamma_{i-1,j}^l + \epsilon f_{\varphi_i}(\mathbf{A}_i)} \leq \frac{1}{c^l}.
$$

$\square$

### A.2.3 Proof of Lemma 5.5

*Proof.* For brevity, going forward, we denote $h_{j+1}^l = \frac{d_{j+1}}{\gamma_{i-1,j}^l + \epsilon f_{\varphi_i}(\mathbf{A}_i)}$. Recall that Lemma 5.4 showed that $h_{j+1}^l \leq \frac{1}{c^l}$.

To apply Lemma 5.3 we set $q = \gamma_{i-1,j}^u, r = d_{j+1}/h_{j+1}^u, s = f_{\mathbf{X}}(\mathbf{a}_i)$ and $w = \epsilon f_{\varphi_i}(\mathbf{A}_i)$. Further let $u = h_{j+1}^u(1 - p_{j+1}(1 + \epsilon/2)), v = -h_{j+1}^u p_{j+1}(1 + \epsilon/2)$ and $p = p_{j+1}$. Note that from the above substitution we get $\frac{r}{q+w} = 1$ and $t = \gamma_{i-1,j+1}^u$. To prove the corollary we need the RHS of the lemma $\frac{pu + (1-p)v - uv}{(1-u)(1-v)} \leq 0$. After substituting every term we we get, $\frac{p_{j+1}h_{j+1}^u(h_{j+1}^u(1+\epsilon/2 - p_{j+1}(1+\epsilon/2)^2) - \epsilon/2)}{(1-u)(1-v)}$. Now if $p_{j+1} \geq 1/(1 + \epsilon/2)$

then this term is non positive, else when $c^u = \frac{2}{\epsilon} + 1$ the term remains non-positive. As $\pi_{j+1} \in \{0, 1\}$ is the sampling choice made by the algorithm for $\mathbf{a}_{j+1}$. So we have,

$$\mathbb{E}_{\pi_{j+1}} \left[ \frac{f_{\mathbf{X}}(\mathbf{a}_i)}{\gamma^u_{i-1,j+1} + \epsilon f_{\varphi_i}(\mathbf{A}_i)} \right] \leq \frac{f_{\mathbf{X}}(\mathbf{a}_i)}{\gamma^u_{i-1,j} + \epsilon f_{\varphi_i}(\mathbf{A}_i)}$$

A similar analysis also follows for the lower barrier. We use Lemma 5.3 by setting $q = \gamma^l_{i-1,j}, r = d_{j+1}/h^l_{j+1}, s = f_{\mathbf{X}}(\mathbf{a}_i)$ and $w = \epsilon f_{\varphi_i}(\mathbf{A}_i)$. Further let $u = -h^l_{j+1}(1 - p_{j+1}(1 - \epsilon/2)), v = h^l_{j+1} p_{j+1}(1 - \epsilon/2))$ and $p = p_{j+1}$. By these substitution we have, $\frac{r}{q+w} = 1$ and the random variable $t = \gamma^l_{i-1,j+1}$. Let $\pi_{j+1} \in \{0, 1\}$ is the random sampling choice made by the algorithm for $\mathbf{a}_{j+1}$. Now for $c^l = \frac{2}{\epsilon} - 1$ we get,

$$\mathbb{E}_{\pi_{j+1}} \left[ \frac{f_{\mathbf{X}}(\mathbf{a}_i)}{\gamma^l_{i-1,j+1} + \epsilon f_{\varphi_i}(\mathbf{A}_i)} \right] \leq \frac{f_{\mathbf{X}}(\mathbf{a}_i)}{\gamma^l_{i-1,j} + \epsilon f_{\varphi_i}(\mathbf{A}_i)}$$

$\square$

### A.2.4 Proof of Lemma 5.6

*Proof.* We use Lemma 5.5 to get the expected upper bound on the sensitivity scores i.e.,

$$\sup_{\mathbf{X}} \frac{f_{\mathbf{X}}(\mathbf{a}_i)}{(1 + \epsilon)f_{\mathbf{X}}(\mathbf{A}_{i-1}) - f_{\mathbf{X}}(\mathbf{C}_{i-1}, \Omega_{i-1}) + \epsilon f_{\varphi_i}(\mathbf{A}_i)}$$

$$\sup_{\mathbf{X}} \frac{f_{\mathbf{X}}(\mathbf{a}_i)}{f_{\mathbf{X}}(\mathbf{C}_{i-1}, \Omega_{i-1}) - (1 - \epsilon)f_{\mathbf{X}}(\mathbf{A}_{i-1}) + \epsilon f_{\varphi_i}(\mathbf{A}_i)}$$

Let $(1 + \epsilon)f_{\mathbf{X}}(\mathbf{A}_{i-1}) - f_{\mathbf{X}}(\mathbf{C}_{i-1}, \Omega_{i-1}) = f_{\mathbf{X}}(\mathbf{A}^u_{i-1})$ where $f_{\mathbf{X}}(\mathbf{A}^u_{i-1}) = \sum_{j \leq i-1} f_{\mathbf{X}}(\mathbf{a}^u_j)$. Here each term $f_{\mathbf{X}}(\mathbf{a}^u_j) = (1 + \epsilon - p^{-1}_j)f_{\mathbf{X}}(\mathbf{a}_j)$ if $\mathbf{a}_j$ is present in $\mathbf{C}_{i-1}$ else $f_{\mathbf{X}}(\mathbf{a}^u_j) = (1 + \epsilon)f_{\mathbf{X}}(\mathbf{a}_j)$. Now the expected upper bound on the upper barrier sensitivity score can be bounded as follows.

$$\mathbb{E}_{\Pi_{i-1}} \left[ \frac{f_{\mathbf{X}}(\mathbf{a}_i)}{f_{\mathbf{X}}(\mathbf{A}^u_{i-1}) + \epsilon f_{\varphi_i}(\mathbf{A}_i)} \right] \overset{(i)}{\leq} \mathbb{E}_{\Pi_{i-1}} \left[ \frac{f^{\mathbf{M}_i}_{\mathbf{X}}(\mathbf{a}_i)}{f_{\mathbf{X}}(\mathbf{A}^u_{i-1}) + \epsilon f_{\varphi_i}(\mathbf{A}_i)} \right]$$

$$\overset{(ii)}{\leq} \mathbb{E}_{\pi_{i-1}} \left[ \frac{\left[ 2f^{\mathbf{M}_i}_{\varphi_i}(\mathbf{a}_i) + \frac{4}{i-1}\sum_{\mathbf{a}_j \in \mathbf{A}_{i-1}}[f^{\mathbf{M}_i}_{\varphi_i}(\mathbf{a}_j) + f^{\mathbf{M}_i}_{\mathbf{X}}(\mathbf{a}_j)] \right]}{(1 + \epsilon)f_{\mathbf{X}}(\mathbf{A}_{i-1}) - f_{\mathbf{X}}(\mathbf{C}_{i-1}, \Omega_{i-1}) + \epsilon f_{\varphi_i}(\mathbf{A}_i)} \Big| \Pi_{i-2} \right]$$

$$= \mathbb{E}_{\pi_{i-1}} \left[ \frac{\left[ 2f^{\mathbf{M}_i}_{\varphi_i}(\mathbf{a}_i) + \frac{4}{i-1}\sum_{\mathbf{a}_j \in \mathbf{A}_{i-1}} f^{\mathbf{M}_i}_{\varphi_i}(\mathbf{a}_j) \right]}{(1 + \epsilon)f_{\mathbf{X}}(\mathbf{A}_{i-1}) - f_{\mathbf{X}}(\mathbf{C}_{i-1}, \Omega_{i-1}) + \epsilon f_{\varphi_i}(\mathbf{A}_i)} \Big| \Pi_{i-2} \right]$$

$$+ \mathbb{E}_{\pi_{i-1}} \left[ \frac{\frac{4}{i-1}f^{\mathbf{M}_i}_{\mathbf{X}}(\mathbf{A}_{i-1})}{(1 + \epsilon)f_{\mathbf{X}}(\mathbf{A}_{i-1}) - f_{\mathbf{X}}(\mathbf{C}_{i-1}, \Omega_{i-1}) + \epsilon f_{\varphi_i}(\mathbf{A}_i)} \Big| \Pi_{i-2} \right]$$

$$\overset{(iii)}{=} \mathbb{E}_{\pi_{i-1}} \left[ \frac{2f^{\mathbf{M}_i}_{\varphi_i}(\mathbf{a}_i) + \frac{4}{i-1}f^{\mathbf{M}_i}_{\varphi_i}(\mathbf{A}_{i-1})}{\gamma^u_{i-1,i-1} + \epsilon f_{\varphi_i}(\mathbf{A}_i)} \Big| \Pi_{i-2} \right]$$

$$+ \mathbb{E}_{\pi_{i-1}} \left[ \frac{\frac{4}{i-1}f^{\mathbf{M}_i}_{\mathbf{X}}(\mathbf{A}_{i-1})}{\gamma^u_{i-1,i-1} + \epsilon f_{\varphi_i}(\mathbf{A}_i)} \Big| \Pi_{i-2} \right]$$

$$\overset{(iv)}{\leq} \mathbb{E}_{\pi_{i-2}} \left[ \frac{2f^{\mathbf{M}_i}_{\varphi_i}(\mathbf{a}_i) + \frac{4}{i-1}f^{\mathbf{M}_i}_{\varphi_i}(\mathbf{A}_{i-1})}{\gamma^u_{i-1,i-2} + \epsilon f_{\varphi_i}(\mathbf{A}_i)} \Big| \Pi_{i-3} \right]$$

$$+ \mathbb{E}_{\pi_{i-2}} \left[ \frac{\frac{4}{i-1}f^{\mathbf{M}_i}_{\mathbf{X}}(\mathbf{A}_{i-1})}{\gamma^u_{i-1,i-2} + \epsilon f_{\varphi_i}(\mathbf{A}_i)} \Big| \Pi_{i-3} \right]$$

$$\overset{(v)}{\leq} \mathbb{E}_{\pi_0} \left[ \frac{2f^{\mathbf{M}_i}_{\varphi_i}(\mathbf{a}_i) + \frac{4}{i-1}f^{\mathbf{M}_i}_{\varphi_i}(\mathbf{A}_{i-1})}{\gamma^u_{i-1,0} + \epsilon f_{\varphi_i}(\mathbf{A}_i)} \right] + \mathbb{E}_{\pi_0} \left[ \frac{\frac{4}{i-1}f^{\mathbf{M}_i}_{\mathbf{X}}(\mathbf{A}_{i-1})}{\gamma^u_{i-1,0} + \epsilon f_{\varphi_i}(\mathbf{A}_i)} \right]$$

$$
= \frac{2f^{\mathbf{M}_i}_{\varphi_i}(\mathbf{a}_i) + \frac{4}{i-1}f^{\mathbf{M}_i}_{\varphi_i}(\mathbf{A}_{i-1})}{\epsilon/2 f_{\mathbf{X}}(\mathbf{A}_{i-1}) + \epsilon f_{\varphi_i}(\mathbf{A}_i)} + \frac{\frac{4}{(i-1)}f^{\mathbf{M}_i}_{\mathbf{X}}(\mathbf{A}_{i-1})}{\epsilon/2 f_{\mathbf{X}}(\mathbf{A}_{i-1}) + \epsilon f_{\varphi_i}(\mathbf{A}_i)}
$$

$$
\overset{(vi)}{\leq} \frac{2f^{\mathbf{M}_i}_{\varphi_i}(\mathbf{a}_i) + \frac{4}{i-1}f^{\mathbf{M}_i}_{\varphi_i}(\mathbf{A}_{i-1})}{\mu_i \epsilon (0.5 f^{\mathbf{M}_i}_{\mathbf{X}}(\mathbf{A}_{i-1}) + f^{\mathbf{M}_i}_{\varphi_i}(\mathbf{A}_i))} + \frac{\frac{4}{(i-1)}f^{\mathbf{M}_i}_{\mathbf{X}}(\mathbf{A}_{i-1})}{\mu_i \epsilon (0.5 f^{\mathbf{M}_i}_{\mathbf{X}}(\mathbf{A}_{i-1}) + f^{\mathbf{M}_i}_{\varphi_i}(\mathbf{A}_i))}
$$

$$
\leq \frac{2f^{\mathbf{M}_i}_{\varphi_i}(\mathbf{a}_i)}{\mu_i \epsilon f^{\mathbf{M}_i}_{\varphi_i}(\mathbf{A}_i)} + \frac{4f^{\mathbf{M}_i}_{\varphi_i}(\mathbf{A}_{i-1})}{\mu_i \epsilon (i-1) f^{\mathbf{M}_i}_{\varphi_i}(\mathbf{A}_i)} + \frac{8f^{\mathbf{M}_i}_{\mathbf{X}}(\mathbf{A}_{i-1})}{\mu_i \epsilon (i-1) f^{\mathbf{M}_i}_{\mathbf{X}}(\mathbf{A}_{i-1})}
$$

$$
\overset{(vii)}{\leq} \frac{2f^{\mathbf{M}_i}_{\varphi_i}(\mathbf{a}_i)}{\mu_i \epsilon f^{\mathbf{M}_i}_{\varphi_i}(\mathbf{A}_i)} + \frac{4}{\mu_i \epsilon (i-1)} + \frac{8}{\mu_i \epsilon (i-1)}
$$

$$
\leq \frac{2f^{\mathbf{M}_i}_{\varphi_i}(\mathbf{a}_i)}{\mu_i \epsilon \sum_{j \leq i} f^{\mathbf{M}_j}_{\varphi_j}(\mathbf{a}_j)} + \frac{12}{\mu_i \epsilon (i-1)}
$$

The inequality $(i)$ is by upper bounding Bregman divergence by squared Mahalanobis distance. The inequality $(ii)$ is due to applying triangle inequality on the numerator, $(a^2 + b^2) \leq 2(a^2 + b^2)$. The $(iii)$ equality is by replacing the denominator with the above definition. The $(iv)$ inequality is by applying the supporting Lemma 5.3. By recursively applying Lemma 5.3 we get the inequality $(v)$ which is independent of the random choices made by the algorithm. The inequality $(vi)$ is by using the lower bound on the denominator. The inequality $(vii)$ an upper bound on the second and the third term. In the final inequality we use the fact that for any $\mu$ similar Bregman divergence from Ackermann & Blömer (2009); Lucic et al. (2016) we have $\mathbf{M}_j \preceq \mathbf{M}_i$ for $j \leq i$ (Lemma 4.1). Further by the property of Bregman divergence we know that $f_{\varphi_i}(\mathbf{A}_{i-1}) \geq f_{\varphi_{i-1}}(\mathbf{A}_{i-1})$. Hence we have $f^{\mathbf{M}_i}_{\varphi_i}(\mathbf{A}_i) \geq \sum_{j \leq i} f^{\mathbf{M}_j}_{\varphi_j}(\mathbf{a}_j)$. Notice that the above analysis is also true for all $\mathbf{X}$. Hence we have this upper bound for all $\mathbf{X}$ with at most $n$ centers. So we have the,

$$
\mathbb{E}_{\Pi_{i-1}}\left[\frac{f_{\mathbf{X}}(\mathbf{a}_i)}{(1+\epsilon) f_{\mathbf{X}}(\mathbf{A}_{i-1}) - f_{\mathbf{X}}(\mathbf{C}_{i-1}, \Omega_{i-1}) + f_{\varphi_i}(\mathbf{A}_i)}\right] \leq \frac{2f_{\varphi_i}(\mathbf{a}_i)}{\epsilon \sum_{j \leq i} f_{\varphi_j}(\mathbf{a}_j)} + \frac{12}{\epsilon (i-1)}
$$

Now let $(f_{\mathbf{X}}(\mathbf{C}_{i-1}, \Omega_{i-1}) - (1-\epsilon) f_{\mathbf{X}}(\mathbf{A}_{i-1}) = f_{\mathbf{X}}(\mathbf{A}^l_{i-1})$ where $f_{\mathbf{X}}(\mathbf{A}^l_{i-1}) = \sum_{j \leq i-1} f_{\mathbf{X}}(\mathbf{a}^l_j)$. Here each term $f_{\mathbf{X}}(\mathbf{a}^l_j) = (p_j^{-1} - 1 + \epsilon) f_{\mathbf{X}}(\mathbf{a}_j)$ if $\mathbf{a}_j$ is present in $\mathbf{C}_{i-1}$ else $f_{\mathbf{X}}(\mathbf{a}^l_j) = (-1 + \epsilon) f_{\mathbf{X}}(\mathbf{a}_j)$. Now the expected upper bound on the lower sensitivity score is,

$$
\mathbb{E}_{\Pi_{i-1}}\left[\frac{f_{\mathbf{X}}(\mathbf{a}_i)}{f_{\mathbf{X}}(\mathbf{A}^l_{i-1}) + \epsilon f_{\varphi_i}(\mathbf{A}_i)}\right] \overset{(i)}{\leq} \mathbb{E}_{\Pi_{i-1}}\left[\frac{f^{\mathbf{M}_i}_{\mathbf{X}}(\mathbf{a}_i)}{f_{\mathbf{X}}(\mathbf{A}^l_{i-1}) + \epsilon f_{\varphi_i}(\mathbf{A}_i)}\right]
$$

$$
\overset{(ii)}{\leq} \mathbb{E}_{\pi_{i-1}}\left[\frac{\left[2f^{\mathbf{M}_i}_{\varphi_i}(\mathbf{a}_i) + \frac{4}{i-1}\sum_{\mathbf{a}_j \in \mathbf{A}_{i-1}}[f^{\mathbf{M}_i}_{\varphi_i}(\mathbf{a}_j) + f^{\mathbf{M}_i}_{\mathbf{X}}(\mathbf{a}_j)]\right]}{f_{\mathbf{X}}(\mathbf{C}_{i-1}, \Omega_{i-1}) - (1-\epsilon) f^{\mathbf{M}_i}_{\mathbf{X}}(\mathbf{A}_{i-1}) + \epsilon f_{\varphi_{ii}}(\mathbf{A}_i)}\Big|\Pi_{i-2}\right]
$$

$$
= \mathbb{E}_{\pi_{i-1}}\left[\frac{\left[2f^{\mathbf{M}_i}_{\varphi_i}(\mathbf{a}_i) + \frac{4}{i-1}f^{\mathbf{M}_i}_{\varphi_i}(\mathbf{A}_{i-1})\right]}{f_{\mathbf{X}}(\mathbf{C}_{i-1}, \Omega_{i-1}) - (1-\epsilon) f_{\mathbf{X}}(\mathbf{A}_{i-1}) + \epsilon f_{\varphi_i}(\mathbf{A}_i)}\Big|\Pi_{i-2}\right]
$$

$$
+ \mathbb{E}_{\pi_{i-1}}\left[\frac{\frac{4}{\epsilon(i-1)}f^{\mathbf{M}_i}_{\mathbf{X}}(\mathbf{A}_{i-1})}{f_{\mathbf{X}}(\mathbf{C}_{i-1}, \Omega_{i-1}) - (1-\epsilon) f_{\mathbf{X}}(\mathbf{A}_{i-1}) + \epsilon f_{\varphi_i}(\mathbf{A}_i)}\Big|\Pi_{i-2}\right]
$$

$$
\overset{(iii)}{=} \mathbb{E}_{\pi_{i-1}}\left[\frac{2f^{\mathbf{M}_i}_{\varphi_i}(\mathbf{a}_i) + \frac{4}{i-1}f^{\mathbf{M}_i}_{\varphi_i}(\mathbf{A}_{i-1})}{\gamma^l_{i-1,i-1} + \epsilon f_{\varphi_i}(\mathbf{A}_i)}\Big|\Pi_{i-2}\right]
$$

$$
+ \mathbb{E}_{\pi_{i-1}}\left[\frac{\frac{4}{i-1}f^{\mathbf{M}_i}_{\mathbf{X}}(\mathbf{A}_{i-1})}{\gamma^l_{i-1,i-1} + \epsilon f_{\varphi_i}(\mathbf{A}_i)}\Big|\Pi_{i-2}\right]
$$

$$
\overset{(iv)}{\leq} \mathbb{E}_{\pi_{i-2}}\left[\frac{2f^{\mathbf{M}_i}_{\varphi_i}(\mathbf{a}_i) + \frac{4}{i-1}f^{\mathbf{M}_i}_{\varphi_i}(\mathbf{A}_{i-1})}{\gamma^l_{i-1,i-2} + \epsilon f_{\varphi_i}(\mathbf{A}_i)}\Big|\Pi_{i-3}\right]
$$

$$
+ \mathbb{E}_{\pi_{i-3}}\left[\frac{\frac{4}{i-1}f^{\mathbf{M}_i}_{\mathbf{X}}(\mathbf{A}_{i-1})}{\gamma^l_{i-1,i-2} + \epsilon f_{\varphi_i}(\mathbf{A}_i)}\Big|\Pi_{i-3}\right]
$$

$$
\begin{aligned}
&\overset{(v)}{\leq} \quad \mathbb{E}_{\pi_0}\left[\frac{2f_{\varphi_i}^{\mathbf{M}_i}(\mathbf{a}_i) + \frac{4}{i-1}f_{\varphi_i}^{\mathbf{M}_i}(\mathbf{A}_{i-1})}{\gamma_{i-1,0}^l + \epsilon f_{\varphi_i}(\mathbf{A}_i)}\right] + \mathbb{E}_{\pi_0}\left[\frac{\frac{4}{i-1}f_{\mathbf{X}}^{\mathbf{M}_i}(\mathbf{A}_{i-1})}{\gamma_{i-1,0}^l + \epsilon f_{\varphi_i}(\mathbf{A}_i)}\right] \\
&= \quad \frac{2f_{\varphi_i}^{\mathbf{M}_i}(\mathbf{a}_i) + \frac{4}{i-1}f_{\varphi_i}^{\mathbf{M}_i}(\mathbf{A}_{i-1})}{\epsilon/2 f_{\mathbf{X}}(\mathbf{A}_{i-1}) + \epsilon f_{\varphi_i}(\mathbf{A}_i)} + \frac{\frac{4}{(i-1)}f_{\mathbf{X}}^{\mathbf{M}_i}(\mathbf{A}_{i-1})}{\epsilon/2 f_{\mathbf{X}}(\mathbf{A}_{i-1}) + \epsilon f_{\varphi_i}(\mathbf{A}_i)} \\
&\overset{(vi)}{\leq} \quad \frac{2f_{\varphi_i}^{\mathbf{M}_i}(\mathbf{a}_i) + \frac{4}{i-1}f_{\varphi_i}^{\mathbf{M}_i}(\mathbf{A}_{i-1})}{\mu_i \epsilon (0.5 f_{\mathbf{X}}^{\mathbf{M}_i}(\mathbf{A}_{i-1}) + f_{\varphi_i}^{\mathbf{M}_i}(\mathbf{A}_i))} + \frac{\frac{4}{(i-1)}f_{\mathbf{X}}^{\mathbf{M}_i}(\mathbf{A}_{i-1})}{\mu_i \epsilon (0.5 f_{\mathbf{X}}^{\mathbf{M}_i}(\mathbf{A}_{i-1}) + f_{\varphi_i}^{\mathbf{M}_i}(\mathbf{A}_i))} \\
&\leq \quad \frac{2f_{\varphi_i}^{\mathbf{M}_i}(\mathbf{a}_i)}{\mu_i \epsilon \sum_{j\leq i} f_{\varphi_j}^{\mathbf{M}_j}(\mathbf{a}_j)} + \frac{12}{\mu_i \epsilon (i-1)}
\end{aligned}
$$

The inequality $(i)$ is by upper bounding Bregman divergence by squared Mahalanobis distance. The inequality $(ii)$ is due to applying triangle inequality on the numerator. The $(iii)$ equality is by replacing the denominator with the above definition. The $(iv)$ inequality is by applying the supporting Lemma 5.3. By recursively applying Lemma 5.3 we get the inequality $(v)$ which is independent of the random choices made by the algorithm. The inequality $(vi)$ is by using the lower bound on the denominator and from here the analysis is same as the upper bound analysis. So we have the following,

$$
\mathbb{E}_{\Pi_{i-1}}\left[\frac{f_{\mathbf{X}}(\mathbf{a}_i)}{(1+\epsilon)f_{\mathbf{X}}(\mathbf{A}_{i-1}) - f_{\mathbf{X}}(\mathbf{C}_{i-1}, \Omega_{i-1}) + f_{\varphi_i}(\mathbf{A}_i)}\right] \leq \frac{2f_{\varphi_i}^{\mathbf{M}_i}(\mathbf{a}_i)}{\mu_i \epsilon \sum_{j\leq i} f_{\varphi_j}^{\mathbf{M}_j}(\mathbf{a}_j)} + \frac{12}{\mu_i \epsilon (i-1)}
$$

$\square$

### A.2.5   Proof of Lemma 5.7

*Proof.* First we bound the expected sampling probability of each $\mathbf{a}_i$ i.e., $\mathbb{E}_{\Pi_{i-1}}[p_i]$.

$$
\begin{aligned}
\mathbb{E}_{\Pi_{i-1}}[p_i] &= \quad c^u \mathbb{E}_{\Pi_{i-1}}[\tilde{s}_i^u] + c^l \mathbb{E}_{\Pi_{i-1}}[\tilde{s}_i^l] \\
&\overset{(i)}{\leq} \quad \frac{2c^u f_{\varphi_i}^{\mathbf{M}_i}(\mathbf{a}_i)}{\mu_i \epsilon \sum_{j\leq i} f_{\varphi_j}^{\mathbf{M}_j}(\mathbf{a}_j)} + \frac{12c^u}{\mu_i \epsilon (i-1)} + \frac{2c^l f_{\varphi_i}^{\mathbf{M}_i}(\mathbf{a}_i)}{\epsilon \sum_{j\leq i} f_{\varphi_j}^{\mathbf{M}_j}(\mathbf{a}_j)} + \frac{12c^l}{\mu_i \epsilon (i-1)} \\
&\leq \quad \frac{8f_{\varphi_i}^{\mathbf{M}_i}(\mathbf{a}_i)}{\mu_i \epsilon^2 \sum_{j\leq i} f_{\varphi_j}^{\mathbf{M}_j}(\mathbf{a}_j)} + \frac{48}{\mu_i \epsilon^2 (i-1)}
\end{aligned}
$$

The inequality $(i)$ is because the oracle returns a tight upper bound on the actual barrier sensitivity scores. Now we bound the total expected sample size,

$$
\begin{aligned}
\sum_{3\leq i\leq n} \mathbb{E}[p_i] &\leq \quad \sum_{3\leq i\leq n} \left(\frac{8f_{\varphi_i}^{\mathbf{M}_i}(\mathbf{a}_i)}{\mu_i \epsilon^2 \sum_{j\leq i} f_{\varphi_j}^{\mathbf{M}_j}(\mathbf{a}_j)} + \frac{48}{\mu_i \epsilon^2 (i-1)}\right) \\
&\leq \quad \frac{48\log n}{\mu_i \epsilon^2} + \sum_{3\leq i\leq n}\left(\frac{8f_{\varphi_i}^{\mathbf{M}_i}(\mathbf{a}_i)}{\mu_i \epsilon^2 \sum_{j\leq i} f_{\varphi_j}^{\mathbf{M}_j}(\mathbf{a}_j)}\right)
\end{aligned}
$$

Because of the subtlety that $\mathbb{E}[p_2] \geq 1$, hence in the above analysis we bound sum from 3 to $n$. Let the term $\frac{f_{\varphi_i}^{\mathbf{M}_i}(\mathbf{a}_i)}{\sum_{j\leq i} f_{\varphi_j}^{\mathbf{M}_j}(\mathbf{a}_j)} = q_i \leq 1$. In the following analysis we bound summation of this term i.e., $\sum_{i\leq n} q_i$. For that consider the term $\sum_{j\leq i} f_{\varphi_j}^{\mathbf{M}_j}(\mathbf{a}_j)$ as follows,

$$
\begin{aligned}
\sum_{j\leq i} f_{\varphi_j}^{\mathbf{M}_j}(\mathbf{a}_j) &= \quad \sum_{j\leq i-1} f_{\varphi_j}^{\mathbf{M}_j}(\mathbf{a}_j)\left(1 + \frac{f_{\varphi_i}^{\mathbf{M}_i}(\mathbf{a}_i)}{\sum_{j\leq i-1} f_{\varphi_j}^{\mathbf{M}_j}(\mathbf{a}_j)}\right) \\
&\geq \quad \sum_{j\leq i-1} f_{\varphi_j}^{\mathbf{M}_j}(\mathbf{a}_j)\left(1 + \frac{f_{\varphi_i}^{\mathbf{M}_i}(\mathbf{a}_i)}{\sum_{j\leq i} f_{\varphi_j}^{\mathbf{M}_j}(\mathbf{a}_j)}\right)
\end{aligned}
$$

$$
= \sum_{j \leq i-1} f_{\varphi_j}^{\mathbf{M}_j}(\mathbf{a}_j)(1 + q_i)
$$

$$
\overset{(i)}{\geq} \exp(q_i/2) \sum_{j \leq i-1} f_{\varphi_j}^{\mathbf{M}_j}(\mathbf{a}_j)
$$

$$
\exp(q_i/2) \leq \frac{\sum_{j \leq i} f_{\varphi_j}^{\mathbf{M}_j}(\mathbf{a}_j)}{\sum_{j \leq i-1} f_{\varphi_j}^{\mathbf{M}_j}(\mathbf{a}_j)}
$$

In $(i)$ we use the fact that, for $q_i \leq 1$, $(1 + q_i) \geq \exp(q_i/2)$. Now as we know that $\sum_{j \leq i} f_{\varphi_j}^{\mathbf{M}_j}(\mathbf{a}_j) \geq \sum_{j \leq i-1} f_{\varphi_j}^{\mathbf{M}_j}(\mathbf{a}_j)$ hence following product results into a telescopic product and we get,

$$
\prod_{3 \leq i \leq n} \exp(q_i/2) \leq \frac{\sum_{j \leq n} f_{\varphi_j}^{\mathbf{M}_j}(\mathbf{a}_j)}{f_{\varphi_2}^{\mathbf{M}_2}(\mathbf{a}_2)}
$$

$$
\leq \frac{f_\varphi^{\mathbf{M}}(\mathbf{A})}{f_{\varphi_2}^{\mathbf{M}_2}(\mathbf{a}_2)}
$$

Now taking log in both sides we get $\sum_{3 \leq i \leq n} q_i \leq 2 \log \left( f_\varphi^{\mathbf{M}}(\mathbf{A}) \right) - 2 \log \left( f_{\varphi_2}^{\mathbf{M}_2}(\mathbf{a}_2) \right)$. Now with $p_1 = p_2 = 1$ we have the following bound on the expected samples.

$$
\sum_{1 \leq i \leq n} \mathbb{E}[p_i] \leq 2 + \frac{32}{\mu \epsilon^2} \left( 3 \log n + \log \left( f_\varphi^{\mathbf{M}}(\mathbf{A}) \right) - \log \left( f_{\varphi_2}^{\mathbf{M}_2}(\mathbf{a}_2) \right) \right)
$$

Here we consider that $\forall i \in [n]$ we have $\mu = \mu_n \leq \mu_i$ and $\mathbf{M} = \mathbf{M}_n \succeq \mathbf{M}_i$ as the $\mu$-similar Bregaman divergence parameters for $\mathbf{A}$. Note that the coreset size is independent of $k$ and $d$. Hence the resultant coreset ensures the desired guarantee equation 1 for all $\mathbf{X}$ with at most $n$ centers in $\mathbb{R}^d$. The expected size of the coreset is $O\left( \frac{1}{\mu \epsilon^2} \left( \log n + \log \left( f_\varphi^{\mathbf{M}}(\mathbf{A}) \right) - \log \left( f_{\varphi_2}^{\mathbf{M}_2}(\mathbf{a}_2) \right) \right) \right)$. $\qquad \square$

### A.2.6 Proof of Lemma 5.2: $k$-means Clustering

*Proof.* We prove it using the Lemmas 5.2, 5.6, 5.9 and 5.8. As for $k$-means clustering we have $\mathbf{M}_i = \mathbf{I}_d$ and $\mu_i = 1$ for each $i \leq n$, hence $\forall i \in [n]$ the expected upper bound on both lower and upper barrier sensitivity scores are,

$$
\frac{2 f_{\varphi_i}(\mathbf{a}_i)}{\epsilon \sum_{j \leq i} f_{\varphi_j}(\mathbf{a}_j)} + \frac{12}{\epsilon(i-1)}
$$

It can be verified by a similar analysis as in the proof A.2.1 and A.2.4 of Lemma 5.3 and 5.6. The rest of the lemma's proof follows as it is and we get a required guarantee. The `NonParametricFilter` returns a coreset of $O\left( \frac{1}{\epsilon^2} \left( \log n + \log \left( f_\varphi(\mathbf{A}) \right) - \log \left( f_{\varphi_2}(\mathbf{a}_2) \right) \right) \right)$ expected samples. $\qquad \square$

## A.3 Uniform Deviation

### A.3.1 Proof of Lemma 6.1

*Proof.* For any $\mathbf{a} \in \mathbb{R}^d$ and $\mathbf{X} \in \mathbb{R}^{k \times d}$ we have,

$$
\frac{f_{\mathbf{X}}(\mathbf{a})}{\sigma + \mathbb{E}[f_{\mathbf{X}}(\mathbf{a})]} \overset{(i)}{\leq} \frac{f_{\mathbf{X}}^{\mathbf{M}}(\mathbf{a})}{\mu(\sigma_{\mathbf{M}} + \mathbb{E}[f_{\mathbf{X}}^{\mathbf{M}}(\mathbf{a})])}
$$

$$
\overset{(ii)}{\leq} \frac{2 f_\xi^{\mathbf{M}}(\mathbf{a}) + 2 f_{\mathbf{X}}^{\mathbf{M}}(\xi)}{\mu(\sigma_{\mathbf{M}} + \mathbb{E}[f_{\mathbf{X}}^{\mathbf{M}}(\mathbf{a})])}
$$

$$
= \frac{2 f_\xi^{\mathbf{M}}(\mathbf{a}) + 2 \mathbb{E}[f_{\mathbf{X}}^{\mathbf{M}}(\xi)]}{\mu(\sigma_{\mathbf{M}} + \mathbb{E}[f_{\mathbf{X}}^{\mathbf{M}}(\mathbf{a})])}
$$

$$\overset{(iii)}{\leq} \frac{2f_\xi^{\mathbf{M}}(\mathbf{a}) + 4\mathbb{E}[f_\xi^{\mathbf{M}}(\mathbf{a})] + 4\mathbb{E}[f_{\mathbf{X}}^{\mathbf{M}}(\mathbf{a})]}{\mu(\sigma_{\mathbf{M}} + \mathbb{E}[f_{\mathbf{X}}^{\mathbf{M}}(\mathbf{a})])}$$

$$= \frac{2f_\xi^{\mathbf{M}}(\mathbf{a}) + 4\sigma_{\mathbf{M}} + 4\mathbb{E}[f_{\mathbf{X}}^{\mathbf{M}}(\mathbf{a})]}{\mu(\sigma_{\mathbf{M}} + \mathbb{E}[f_{\mathbf{X}}^{\mathbf{M}}(\mathbf{a})])}$$

$$\leq \frac{2f_\xi^{\mathbf{M}}(\mathbf{a})}{\mu\sigma_{\mathbf{M}}} + \frac{8}{\mu}$$

Here $(i)$ is by using upper bound (on the numerator) and lower bound (on the denominator) of the Bregman divergence using squared Mahalanobis distance. In $(ii)$ and $(iii)$ we use the fact that $(a+b)^2 \leq 2(a^2 + b^2)$. Next we bound $\mathbb{E}[s(\mathbf{a})^2]$.

$$\mathbb{E}[s(\mathbf{a})^2] = \mathbb{E}\left[\left(\frac{2f_\xi^{\mathbf{M}}(\mathbf{a})}{\mu\sigma_{\mathbf{M}}} + \frac{8}{\mu}\right)^2\right]$$

$$= \mathbb{E}\left[\left(\frac{4f_\xi^{\mathbf{M}}(\mathbf{a})^2}{\mu^2\sigma_{\mathbf{M}}^2} + \frac{64}{\mu^2} + \frac{32f_\xi^{\mathbf{M}}(\mathbf{a})}{\mu^2\sigma_{\mathbf{M}}}\right)\right]$$

$$= \frac{\mathbb{E}[4f_\xi^{\mathbf{M}}(\mathbf{a})^2]}{\mu^2\sigma_{\mathbf{M}}^2} + \frac{64}{\mu^2} + \frac{32\mathbb{E}[f_\xi^{\mathbf{M}}(\mathbf{a})]}{\mu^2\sigma_{\mathbf{M}}}$$

$$\leq \frac{4\mathbb{E}[f_\xi^{\mathbf{M}}(\mathbf{a})^2]}{\mu^2\sigma_{\mathbf{M}}^2} + \frac{96}{\mu^2} < t$$

We get the last equality because $\mathbb{E}[f_\xi^{\mathbf{M}}(\mathbf{a})] = \sigma_{\mathbf{M}}$ and by the assumption equation 14. $\qquad\square$

### A.3.2   Proof of Theorem 6.2

*Proof.* The proof of is same as the proof of Theorem 5 in Bachem et al. (2017a). Hence, here we only present proof sketch.

As $\mathbb{E}[s(\mathbf{a})] \leq t$, hence on $2m$ i.i.d. samples $\{\mathbf{a}_i, \ldots, \mathbf{a}_{2m}\}$ by Markov we get $\frac{1}{2m}\sum_{i=1}^{2m} s(\mathbf{a}_i)^2 < O(t)$ with at least a constant probability.

The rest of the proof is based on a double sampling approach. Let $\mathbf{a}_{m+1}, \mathbf{a}_{m+2}, \ldots, \mathbf{a}_{2m}$ be an additional $m$ independent samples from $\mathcal{D}$ and let $h_1, h_2, \ldots, h_m$ be independent random variables uniformly sampled from $\{-1, 1\}$. If $\mathbb{E}[s(\mathbf{a})^2] \leq t$, the probability of equation 16 not holding can be bounded by the probability that there exists a $g_{\mathbf{X}}(\cdot) \in \mathcal{G}$ such that

$$\left| \frac{1}{m} \sum_{i \leq m} h_i \cdot (g_{\mathbf{X}}(\mathbf{a}_i) - g_{\mathbf{X}}(\mathbf{a}_{m+i})) \right| \geq \epsilon \tag{18}$$

We first provide the intuition for some function $g \in \mathcal{G}$ and then show how we extend it to all $g \in \mathcal{G}$. While the function $g(\mathbf{a})$ is not bounded, for a given sample $\mathbf{a}_1, \mathbf{a}_2, \ldots, \mathbf{a}_{2m}$, each $g(\mathbf{a}_i)$ is contained within $[0, s(\mathbf{a}_i)]$. Given the sample $\mathbf{a}_1, \mathbf{a}_2, \ldots, \mathbf{a}_{2m}$, the random variable $h_i \cdot (g(\mathbf{a}_i) - g(\mathbf{a}_{i+m}))$ is bounded in $0 \pm \max\{s(\mathbf{a}_i), s(\mathbf{a}_{i+m})\}$ and has zero mean. Hence, given independent samples $\mathbf{a}_1, \mathbf{a}_2, \ldots, \mathbf{a}_{2m}$, the probability of equation 18 occurring for a single $g \in \mathcal{G}$ can be bounded using Hoeffding's inequality by,

$$\mathbb{P}\left(\left| \frac{1}{m} \sum_{i \leq m} h_i \cdot (g_{\mathbf{X}}(\mathbf{a}_i) - g_{\mathbf{X}}(\mathbf{a}_{m+i})) \right| \leq \epsilon\right) \leq 2\exp\left(\frac{-2m\epsilon^2}{\frac{1}{m}\sum_{i \leq m}\max\{s(\mathbf{a}_i), s(\mathbf{a}_{i+m})\}}\right)$$

$$\leq 2\exp\left(\frac{-m\epsilon^2}{\frac{1}{2m}\sum_{i \leq 2m} s(\mathbf{a}_i)}\right)$$

From lemma 6.1 as we know $\mathbb{E}[s(\mathbf{a})^2] \leq t$, so for $m \in \Omega\left(\frac{t}{\epsilon^2}\right)$ we can ensure above event with at least 0.99 probability. Finally using $\text{Pdim}(\mathcal{G}) \leq \rho$ we take a union to ensure that the above event with at least 0.99 probability. So we get $m \in \Omega\left(\frac{t\rho}{\epsilon^2}\right)$. $\qquad\square$

### A.4 Experiments on MNIST and Song Data

Here we discuss some more experimental results. We run our algorithms to compare with other baseline coreset creation algorithms. Once the coreset is obtained from each of the sampling methods, we run weighted $k$-means++ clustering (Arthur & Vassilvitskii, 2007) on them for various values of $k$ and get the centers. These centers are considered as initial centres while running $k$-means clustering on the coreset and finally obtain the centres. Once these centers are obtained, we compute the quantization error on the entire dataset $C_s$, with respect to the corresponding centers. We also run a similar $k$-means clustering on the entire data for the same values of $k$ and get the quantization error from those centres, i.e., $C_f$. Finally we report the relative error $\eta$, i.e., $\eta = \frac{|C_s - C_f|}{C_f}$.

#### A.4.1 `BregmanFilter`

We compare the performance of `BregFilter` (our `BregmanFilter`) with `Uniform` and `TwoPass` on the following datasets.

1. `MNIST`: $60,000$ points in $784$ dimension digits dataset.

2. `KDD(BIO-TRAIN)`: $145,751$ points with $74$ features.

3. `SONGS`: $515,345$ songs from the Million song dataset with $90$ features.

In the figure 3 we show the change in relative error ($\eta$) with respect to the change in the coreset size (% of data). Here we consider relative entropy (or KL divergence) as Bregman divergence and run the sampling methods on the `MNIST` dataset. As the data has a natural clustering of 10 digits, hence we use $k = \{5, 10\}$. We run the sampling algorithms for various coreset sizes. We run 5 random instances for each each coreset size and here we report the average of their $\eta$.

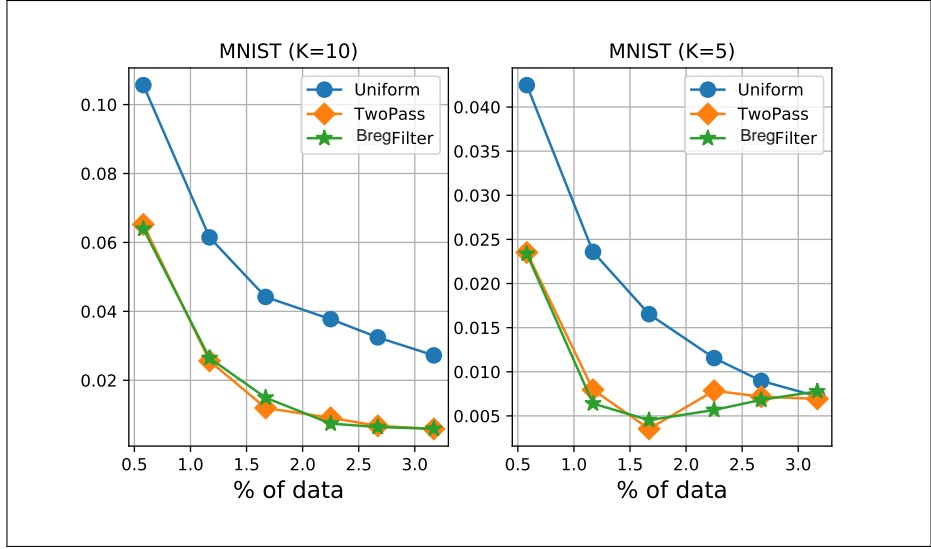

Figure 3: Relative error v/s coreset size for KL divergence.

We also run the sampling methods on `KDD(BIO-TRAIN)` and `SONGS`, considering squared Euclidean distance as the Bregman divergence. In the figure 4 we report the average $\eta$ of the 5 runs, for $k = \{100, 200\}$. The plot shows the change in relative error ($\eta$) with change in coreset size (% of data).

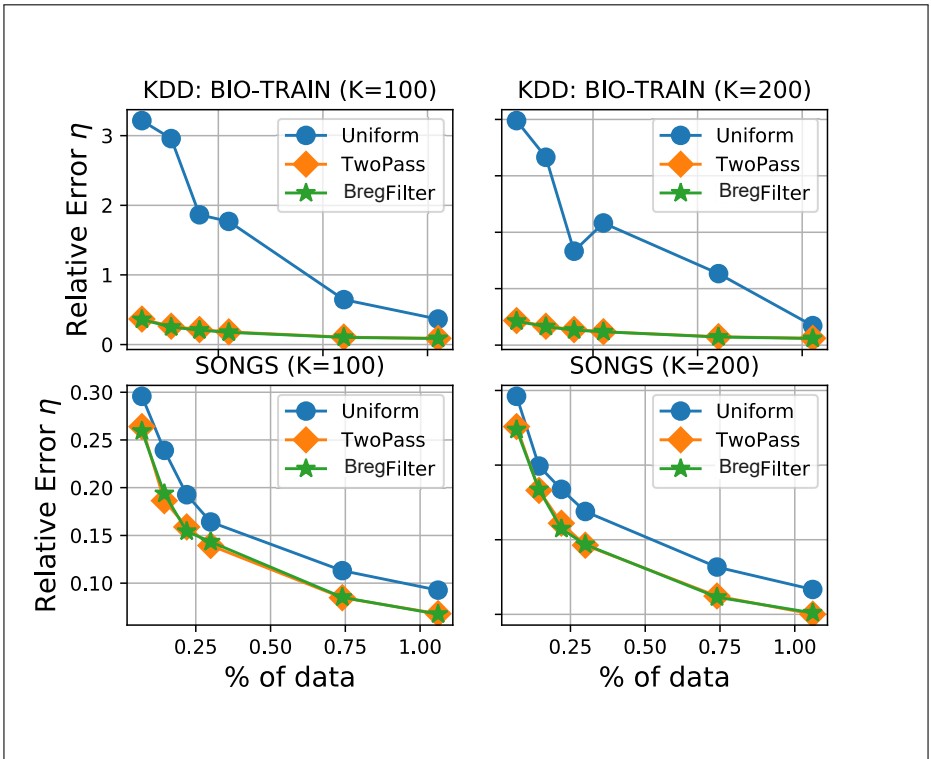

Figure 4: Relative error v/s coreset size. Squared Euclidean Distance as Bregman Divergence.

In all these cases, as per the expectation we do see an improvement in $\eta$ as coreset size increases. Further we also notice that the performance of the `BregFilter` is equivalent to `TwoPass` and outperforms the performance of `Uniform`.

**Bad example for `Leverage`:**  In the datasets considered, the empirical performance of the leverage score is similar to that of the provable sampling strategy that we propose. It is useful to recall that this is not always the case. The following example shows when leverage score can perform really badly.

Consider 4 points in $R^1$ as $(1000), (1000), (1000)$ and $(1)$ coming in this order. On these we are interested in a 2-means clustering, i.e., $k = 2$. Clearly the cluster centers are $(1000)$ and $(1)$. Now the online leverage scores (as defined in Algorithm 1 in Chhaya et al. (2020a)) of these points will be $\frac{1000^2}{1000^2}, \frac{1000^2}{2 \cdot 1000^2}, \frac{1000^2}{3 \cdot 1000^2}$, and $\frac{1}{3 \cdot 1000^2 + 1}$. Let $r$ be the parameter that controls the expected coreset size. So the points are sampled with probability $r \cdot 1, r \cdot 0.5, r \cdot 0.33$ and $\frac{r}{3 \cdot 1000^2 + 1}$ respectively. Now based on this, if we build a coreset, say with $r = 3$, then with a very high probability we will not sample any representative from the $(1)$ cluster (this example can easily be generalized to coresets of a general size). Now with our sensitivity upper bound (refer $l_i$ in algorithm 1) depends on how much a current point is far from the previous points. In this example, the online sensitivity score of the first point $(1000)$ will be 1 and point will be sampled with probability $r \cdot 1$. The second point $(1000)$ is just a copy of the first point, and hence the mean of the first two points is equal to the second point itself. As a result its online sensitivity will be 1 (which is due to the second term in the algorithm) and even this point will be sampled with probability $r \cdot 1$. For the next point $(1000)$, also the mean does not move, so its online sensitivity score will be $1/2$ and its sampling probability will be $r \cdot 0.5$. Now for the final point $(1)$ the mean moves from $(1000)$ to $(750.25)$. So its online sensitivity scores will be $\frac{749.25}{749.25} + \frac{1}{2}$ and its sampling probability be $r \left( \frac{749.25}{749.25} + \frac{1}{2} \right)$, which is bigger than 1. So for any r the final point will be sampled in our coreset with higher probability than the third point in the stream. As a result our coreset, even if it is of size three, will have representatives from each cluster.

The main insight from the above toy example is that leverage score sampling only tries to preserve the rank of the data, and not the cluster structure. Hence, if the data spans only a low rank space but has a large number of clusters then `Leverage` sampling will likely perform worse compared to our `BregFilter`.

### A.4.2 `NonParametricFilter`

Now we compare the performance of a heuristic version of `NonParametricFilter` called `NP-Filter` with `Uniform`, `Offline` and `TwoPass` on KDD(BIO-TRAIN) dataset. The `Offline` is the lightweight coreset from (Bachem et al., 2018a). Here for the comparison we do not consider the assumption 5.1. Instead use simply use the expected upper bounds as shown in lemma 5.6. Further for `NP-Filter` the coreset size is controlled by $\epsilon$. Now once we get a coreset from `NP-Filter`, we set the desired parameters of other sampling methods to get similar coreset size. Here once the coreset is obtained from each of the sampling methods, we get the relative error $\eta$ as described above. Now on each coreset, we run $k$-clustering for various values of $k = \{50, 100, 200, 300\}$ to capture the non-parametric nature of the coreset.

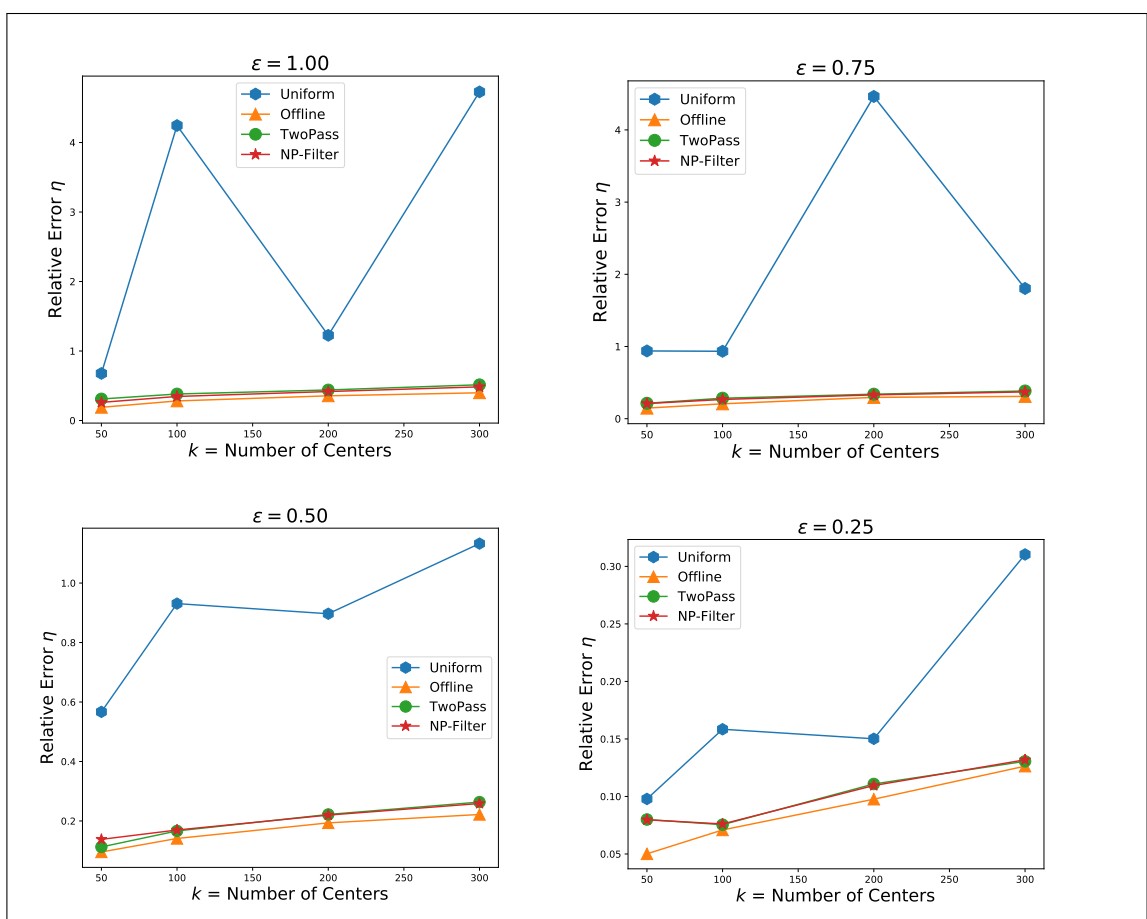

Figure 5: Change in Relative Error $\eta$ with respect to number of centers $k$ for various values of $\epsilon$.

For each of the algorithms and for each value of $\epsilon$ we run 5 random instances, compute $\eta = \frac{|C_S - C_F|}{C_F}$ and report the average $\eta$ value. We consider $\epsilon = \{1.0, 0.75, 0.5, 0.25\}$, for which we have $\{500, 850, 1650, 5500\}$ expected samples. Figure 5 shows the change in the value of Relative Error $\eta$ with respect to the change in number of centers $k$, for various values of $\epsilon$. With decrease in $\epsilon$ we can note that the value of $\eta$ also decreases. This is because as the $\epsilon$ increases, the coreset size reduces, which results to a high $\eta$. Now an interesting point to note is that the $\eta$ remains significantly smaller than $\epsilon$ for the same coreset across various values of $k$. Even though our algorithm is heuristic, but this plot reflects the non-parametric nature of our

coreset from importance sampling. Note that, even though `Offline` outperforms `NP-Filter`, but they are very close.

