# OpenReview forum: "Online Coresets for Parameteric and Non-Parametric Bregman Clustering"
_TMLR — Accepted by TMLR_

### Review · Reviewer_QNv9 · 2022-05-21

**Summary Of Contributions:**

The authors study the problem of constructing coresets for Bregman Clustering in an online setting. The approximation guarantee is in both relative and additive errors with high probability. Compare to the previous works, the size of the coreset constructed by the authors has better dependency on the parameters $d,k,\epsilon$ and independent of $n$. The authors also consider the case where the optimal cluster size is unknown, which they refer to as non-parametric clustering.

**Requested Changes:**

1. Fix the typo and notation issues as mentioned in the previous section.

2. Either provide more evidence in practice that BregFilter outperforms baselines, or tone down the claim in the experiment section. Even without strong experiment results the theory part seems convincing.

3. Add more online methods as baselines, such as those mentioned in the related work section.

4. For Figure 1, since the method proposed by the authors will either keep a streaming point in coreset or discarding it, I wonder how can the authors make all methods have the same coreset size? It seems to me that all points in Figure 1 are aligned, which implies them having the same coreset size. Please explain this issue.

5. For experiment in non-parametric case (Figure 2), I wonder if Uniform and TwoPass are the only two possible baselines? Does that mean all previous methods can not take care of this case? Please confirm.

6. In the first paragraph (page 1), the authors write “A canonical definition of the clustering problem is via the **k-median**, in which k possible centers need to be proposed such that the **sum** of distances of every point to its closest center is minimized.” Should not k-means corresponds to sum/mean and k-median corresponds to the median?

7. If possible, can the authors design some experiments to actually verify their theorem? So far it I did not notice any of them. One of such example can be comparing with the theoretical bound or trends.


**Strengths And Weaknesses:**

Strengths:

(+) The authors provide a thorough study of online coreset problem for Bregman clustering. The new theoretical results show the better (smaller) coreset size in expectation at a cost of newly introduced additive error in the approximation. These are supported by the theories and analysis in Section 4 and 5.

Weaknesses:

(-) The proposed algorithm shows little or no improvement compare to the baselines in the experiment. Also, claims in the experiment section are not well supported. For example, the authors claim "Our online sampling method BregFilter shows competitive performance with the best known offline methods". However, this is not true by examining Figure 1. I do not consider the results of BregFilter (proposed by the authors) as competitive to offline methods such as LWCS. In fact, I feel the performance of BregFilter is roughly the same as Leverage, which is an online method. Also, I have some questions about Figure 1, see Requested Changes section below.

(-) The writing of the paper can be improved. There are typos and undefined notations in the paper. For example, in page 4, what is the notation of $A_{i-i}$? Also, in page 5 definition 2.3, the notation $\leftrightarrow$ is undefined. Furthermore, it is unclear whether ${above; below}$ is a set of two reals or not. If it is, then should not it be using comma "," instead of semicolon ";"?

(-) The baseline to be compared in the experiment section is very limited. The authors mention many more of the online methods in the related work section. Why not also compare with them?

---

> ### Author Response · Authors · 2022-06-05
> **Response to reviewer QNv9 1/n**
>
> We thank the reviewer for the detailed review and feedback.
>
> Q1. Fix the typo and notation issues as mentioned in the previous section
>
> A1. Thank you for pointing them out, we will proofread the paper and we will correct all the typos and missing notations.
>
>
> Q2. Either provide more evidence in practice that BregFilter outperforms baselines, or tone down the claim in the experiment section. Even without strong experiment results the theory part seems convincing.
>
> A2. Yes we agree, we are not clearly outperforming the baseline (i.e., leverage score).  We would also like to point out that leverage scores do not come with any guarantees for the clustering problem. ** The following example illustrates this issue.**
>
> Consider 4 points in $R^1$ as $(1000), (1000), (1000)$ and $(1)$ coming in this order. On these we are interested in a 2-means clustering, i.e., $k = 2$. Clearly the cluster centers are $(1000)$ and $(1)$. Now the online leverage scores (as defined in Algorithm 1 in [Chhaya et al. "Streaming coresets for symmetric tensor factorization." ICML 2020]) of these points will be $1000^2/1000^2, 1000^2/(2\cdot1000^2)$, $1000^2/(3\cdot1000^2)$, and $1/(3\cdot1000^2+1)$. Let $r$ be the parameter that controls the expected coreset size. So the points are sampled with probability $r\cdot1. r\cdot0.5, r\cdot0.33$ and $r/(3\cdot1000^2+1)$ respectively. Now based on this, if we build a coreset, say with $r = 3$,  then with a very high probability we will not sample any representative from the $(1)$ cluster (this example can easily be generalized to coresets of a general size). Now with our sensitivity upper bound (refer $l_i$ in algorithm 1) depends on how much a current point is far from the previous points. In this example, the online sensitivity score of the first point $(1000)$ will be $1$ and point will be sampled with probability $r\cdot1$. The second point $(1000)$ is just a copy of the first point, and hence the mean of the first two points is equal to the second point itself. As a result its online sensitivity will be 1 (which is due to the second term in the algorithm) and even this point will be sampled with probability $r\cdot1$. For the next point $(1000)$, also the mean does not move, so its online sensitivity score  will be $1/2$ and its sampling probability will be $r\cdot0.5$. Now for the final point $(1)$ the mean moves from $(1000)$ to $(750.25)$. So its online sensitivity scores will be $749.25/749.25 + 1/2$ and its sampling probability be $r(749.25/749.25 + 1/2)$, which is bigger than $1$. So for any r the final point will be sampled in our coreset with higher probability than the third point in the stream. As a result our coreset, even if it is of size three, will have representatives from each cluster.
>
> The main insight from the above toy example is that leverage score sampling only tries to preserve the rank of the data, and not the cluster structure. Hence, if the data spans only a low rank space but has a large number of clusters then the online leverage score sampling will likely perform worse compared to our BregFilter.
> Also, note that the update time of online leverage scores sampling is $O(d^{2})$ compare to our BregFilter which is $O(d)$. This is reflected in our experiments too, and we will report this.
>
>
> Q3. Add more online methods as baselines, such as those mentioned in the related work section.
>
> A3. Note that the online algorithms mentioned in the related work give a bicriterial approximation for the online k-means clustering problem and they do not return a coreset. We will clarify this in the related work section. We will also add experiments to compare the performance of BregFilter with the online clustering (algorithm 3) from [Liberty, E 2016]  in the following manner. The online k-means algorithm return a bicriteria approximation, i.e. given a value of k, it returns about $\tilde{O}(k)$ centers. We treat these centers as the coreset and apply an (offline) k-means clustering on these centers to get the final k centers. We then report the performance of this solution to the solution obtained via running an offline algorithm on the coreset obtained by BregFilter.
>
> The update time of these algorithms is at least $O(kd)$ since every incoming point needs to be compared with all existing centers before deciding whether it will be sampled as a center or not. Note that this can be made more efficient by running the online clustering algorithms in a pipeline with BregFilter, where the output of BregFilter is fed to these online k-means algorithms. As the update time of BregFilter is $O(d)$, overall we get a better amortized update time of $O(d)$.

---

> > ### Author Response · Authors · 2022-06-05
> > **Response to reviewer QNv9 2/n**
> >
> > Q4. For Figure 1, since the method proposed by the authors will either keep a streaming point in coreset or discarding it, I wonder how can the authors make all methods have the same coreset size? It seems to me that all points in Figure 1 are aligned, which implies them having the same coreset size. Please explain this issue.
> >
> > A4. For every sampling process, we have run multiple instances of the technique in such a way that over expectation the coreset size of every technique is the same. In the plot, the X-axis represents the expected coreset size, which is a parameter of the algorithm that is used to scale the sampling probabilities to achieve the expected sample size. The actual sample size is concentrated around the expectation. We will clarify this in our updated version.
> >
> >
> > Q5. For experiment in non-parametric case (Figure 2), I wonder if Uniform and TwoPass are the only two possible baselines? Does that mean all previous methods can not take care of this case? Please confirm.
> >
> > A5. To the best of our knowledge we do not know of any online algorithm that builds coreset for the non-parametric case.
> >
> >
> > Q6. In the first paragraph (page 1), the authors write “A canonical definition of the clustering problem is via the k-median, in which k possible centers need to be proposed such that the sum of distances of every point to its closest center is minimized.” Should not k-means correspond to sum/mean and k-median corresponds to the median?
> >
> > A6.  In both k-means and k-median, the total cost is always the sum of the individual point costs. Traditionally, k-means refers to the problem when the cost of every point is the square of the distance to the closest center. The ''**means**'' terminology refers to the fact that if the distance is Euclidean, then the optimal center, under the squared distance cost, is truly the mean (i.e. average) of the cluster points. This is not true if the distance is not Euclidean.
> > Similarly, **k-median** refers to the problem where the cost of each point is the distance to the assigned center. Again, the term ``**median**'' is (perhaps) a reference to the fact that if the distance function is $\ell_1$, then the optimal center is really the (distributional) median of the points (taken coordinate-wise if points are in $R^d$). However, this is not true for the k-median problem under other distance measures.
> > Since for Bregman divergences we do not square the divergence of a point from the center, using the k-median terminology is more appropriate. Note that $\ell_2^2$ itself is a Bregman divergence, and k-median with $\ell_2^2$ is exactly the traditional k-means for Euclidean distance.
> >
> >
> > Q7. If possible, can the authors design some experiments to actually verify their theorem? So far it I did not notice any of them. One of such example can be comparing with the theoretical bound or trends.
> >
> > A7. Please note that our existing plots already show that the actual coreset size needed is smaller than the theoretical bound that we obtained. Note that in the figure 1 for $k = 100$, our desired coreset size is $\tilde{O}(dk/\epsilon^2)$ (see theorem 4.6). The experiments are based on the Biotrain data whose n = 145751 and d = 74. Now even to get a 2-factor approximation, i.e., $\epsilon = 1$, theoretically we will need $\tilde{O}(dk)$ which is at least 7400 samples. However, in the plot notice that even with just 3% of the data, i.e., around $4500$ samples we are achieving a relative error approximation of at most $0.1$. The plot also shows the expected trend of decreasing error with increasing coreset size. A similar trend can also be verified from our other plots.
> >
> > If the reviewer had other experiments in mind to illustrate the theorems, we will be glad to discuss them.

---

> > > ### Comment · Reviewer_QNv9 · 2022-06-18
> > > **Re**
> > >
> > > I thank the authors for their patient and detailed rebuttal. I have no further questions or requests. Please make sure to address the issues mentioned by all reviewers in the revision.

---

### Review · Reviewer_XJ2S · 2022-05-29

**Summary Of Contributions:**

The paper suggests the first coreset for online non-parametric clustering based on Bergman divergencies and offers improved coresets for the online parametric clustering based on Bergman divergencies.

**Broader Impact Concerns:**

There are none.

**Requested Changes:**

* Please fix the following typos:
    - the dimension of the input points are -> the dimension of the input points is (first page)
    - The comparison are done on real -> The comparison is done on real (third page)
    - One can also define A clustering problem is called -> One can also define a clustering problem as (fourth page)
    - Note that this algorithm is online is nature because for -> Note that this algorithm is online in nature because for (sixth page)
    - We present BregmanFilter as algorithm equation 1. -> remove equation (sixth page)
    - add a comma when trying to explain the transitions in any equation, and a dot to the end of equations when ending without any explanations.
    - sampling enough points based on $l_i$, we get equation 1 ->  add a dot to the end. (ninth page)
    - What is $t$ in equation 12? is it the total sensitivity? please add a brief explanation.
    - where BregFilter will out perform Leverage -> where BregFilter will outperform Leverage.
    - Page 26, in the appendix, please remove the additional $($ from the definition of $\mathbb{P}$
    - Please make Figure 5 more understandable, i.e., make it bigger for instance.

* More experimental results:
    - Please compare against the online versions of the coresets you have chosen to compete against by using the merge-and-reduce tree.

* Discuss more regarding how hard is to approximate an upper bound on the barrier sensitivity, to better elevate the use of empirical bounds on this type of sensitivity.

**Strengths And Weaknesses:**

* Strengths:
    1. The paper provides improved online coreset construction for parametric and non-parametric clustering via Bergman divergences. The coresets admit deterministic guarantees whereas the randomness is with respect to the coreset size.
    2. The coreset sampling attribute is analyzed via Bernestien's inequality leading to better and more interesting results.
    3. The notion of "barrier sensitivity" is certainly interesting and quite useful in the analysis of the paper. I anticipate that such a notion will be very useful in the literature of coresets as it provides leanness and as well as interesting, out-of-the-norm, provable guarantees.
    4. Elegant proofs throughout the paper.

* Weaknesses:
    1. Written typos
    2. Insufficient experimental comparison against online coresets for clustering problems based on Bergman divergences. The authors claim insufficiency towards the existence of online coreset that are applicable for comparison against their methods. Offline coresets can be made online rather easily through the use of the merge-and-reduce tree where coresets will be made applicable towards handles stream of points. In addition, such coresets can also adapt to the dynamic setting where points can be removed from the current stream/offline batch of points.

---

> ### Author Response · Authors · 2022-06-05
> **Response to reviewer XJ2S**
>
> We thank the reviewer for the detailed reviewer and feedback.
>
> Q1. Please compare against the online versions of the coresets you have chosen to compete against by using the merge-and-reduce tree.
>
> A1. The reviewer raised a very good point, but please note that our online setup is focused to a very resource constrained setting, where for every input point we only allow an irrevocable decision of including it in the coreset or now. The construction that uses “merge-and-reduce” to convert any offline coreset algorithm into a **streaming** one does not fit this model, since points that were chosen before can be discarded later. Hence we have not compared against the merge-and-reduce style algorithms. As mentioned in the response to other reviewers, we will add online clustering algorithms as other baselines.
>
> Q2. Discuss more regarding how hard is to approximate an upper bound on the barrier sensitivity, to better elevate the use of empirical bounds on this type of sensitivity.
>
>
> A2. The cost function of the k-median clustering problem is decomposable, i.e., it can expressed as sum of costs of individual points. The existing sensitivity framework is well defined for decomposable function which are sum of non-negative terms. Notice that in the case of our upper barrier sensitivity function for $a_i$, the cost $(1+\epsilon)f(A_{i-1}) - f(C_{i-1},\Omega_{i-1})$ is not sum of non-negative terms. In fact, for points which are sampled in $C_{i-1}$ with probability higher than $1/(1+\epsilon)$ will account a negative weight. This makes it difficult to analyze the upper bound with any specific instace of $C_{i-1}$.
> However, since the denominator $(1+\epsilon)f(A_{i-1}) - f(C_{i-1},\Omega_{i-1}) + \epsilon\cdot f_{\varphi_{i}}(A_i)$ is non-negative by induction, hence we are only able to ensure an upper bound over expectation. A similar argument can be also made for the lower barrier sensitivity scores.
> We currently have no theoretical proof that getting a good approximation to the upper bound of barrier sensitivity is hard. We think this is an interesting open problem.
>
> Q3. Fix typos
>
> A3. Thank you for giving a thorough review. We will incorporate all the requested changes. We will proofread the complete paper and correct all the typos.

---

### Review · Reviewer_3Wo5 · 2022-05-29

**Summary Of Contributions:**

The paper describes new coresets for k-assignment-based clustering under Bregman divergences.  These include non-parametric variants (where k is not know a-priori) and streaming algorithms.  What distinguishes this work from prior work is the streaming nature of the coresets and the additive error, instead of the more common and stronger relative error.  This is a quite reasonable trade-off, and reflects the sort of methods users of such methods may want to actually do, for instance, if only restricted to one pass.

The paper is clearly in scope for TMLR.
No correctness issues were detected.

The writing is generally good and clear.  The proofs and arguments appear rigorous and careful.  The comparison to prior work is fair, and the paper clearly discusses how its ideas build on and relate to those used in other paper.  The assumptions under which the main results are stated are made clear, and motivated.  The experimental sections shows that the coreset sort of results in space/error trade-offs are on par with other methods and heuristics, and supersede random sampling.

I have two (small) suggestions for the authors to chance.  Given the care taken in the rest of the paper, I assume they can do these without being checked.


1. The function f_psi shows up in equation (1), and is used throughout the paper.  While f_q and psi are both defined individually, this function is so prevalent and essential to interpretation of the bounds that I would suggest a sentence or two specifically discussing how to think about it (the sum of divergences from the average point).  Perhaps even for the special case of k-means clustering this term could be replaced by something not relying on the f functions, since we just have Euclidean distance then.

2.  On page 20 between equations (12) and (13) there is partial sentence
  "Now for a point a~D and X in R^{kxd} we define"

**Broader Impact Concerns:**

no broader impact concerns.

**Requested Changes:**

I have two (small) suggestions for the authors to chance.  Given the care taken in the rest of the paper, I assume they can do these without being checked.


1. The function f_psi shows up in equation (1), and is used throughout the paper.  While f_q and psi are both defined individually, this function is so prevalent and essential to interpretation of the bounds that I would suggest a sentence or two specifically discussing how to think about it (the sum of divergences from the average point).  Perhaps even for the special case of k-means clustering this term could be replaced by something not relying on the f functions, since we just have Euclidean distance then.

2.  On page 20 between equations (12) and (13) there is partial sentence
  "Now for a point a~D and X in R^{kxd} we define"

**Strengths And Weaknesses:**

(S) in score of TMLR
(S) no correctness issues detected
(S) well-written paper with rigorously proven bounds.

(W) the experiments are small, and do not show that these methods will have a large impact.  [This paper addresses an interesting space in the score of TMLR, and is mostly theoretical, so not a big concern.]

---

> ### Author Response · Authors · 2022-06-05
> **Response to Reviewer 3Wo5**
>
> We thank the reviewer for the detailed review and feedback.
>
> Q1. The function f_psi shows up in equation (1), and is used throughout the paper. While f_q and psi are both defined individually, this function is so prevalent and essential to interpretation of the bounds that I would suggest a sentence or two specifically discussing how to think about it (the sum of divergences from the average point). Perhaps even for the special case of k-means clustering this term could be replaced by something not relying on the f functions, since we just have Euclidean distance then.
>
> A1. We thank the reviewer for the suggestion. We will put a discussion about the function $f_{\varphi}()$ in the paper.
>
> Q2. On page 20 between equations (12) and (13) there is partial sentence "Now for a point a~D and X in R^{kxd} we define"
>
> A2. We will remove the incomplete sentence as the definition has been stated in the following lemma (Lemma 6.1). We will proofread the entire paper and correct all the typos.

---

### Review · Reviewer_CfBV · 2022-05-31

**Summary Of Contributions:**

The authors propose an online methods for finding coresets for clustering with respect to miu-similar Bregman divergences. These are coresets with respect to additive error (as opposed to coresets that guarantee a 1+eps approximation guarantee). The authors provide two methods, one for usual clustering with k centers, and one for what they call non-parametric (where finding the number of centers is part of the optimization problem). The authors instantiate their general results for special cases of k-means and DP-means (a nonparametric version of k-means) clustering. Based on my understanding, the main contribution compared to previous work is proposing an "online" method.



**Broader Impact Concerns:**

No concerns here.

**Requested Changes:**

+ Adding an outline of different sections can improve the readability a lot.
+ The size of corsets in your main theorems depend on a number of factors such as log[f(A)] and log[n]. Do we know if these dependencies necessary? Can you add a discussion? [there is some discussion in section 6; perhaps you can link to those]
+ It is mentioned in the related work that there are online corsets for k-means in the literature. Can you compare your (theoretical) results with those in more details? What is their coreset size? In what parameter regimes you get an improvement? Perhaps you can compare with them in details after stating your own theorems?
+ The discussion of the uniform deviation bound in section 6 relies on an upper bound. But it is not clear if that upper bound itself is tight and therefore it is not clear if it shows that the dependence on f[A] is necessary. In the absence of some kind of lower bound, the claims should be somewhat weakened.
+ Was there a reason that in section 6 you didn't use classical deviation bounds such as those in this? Biau, Gérard, Luc Devroye, and Gábor Lugosi. "On the performance of clustering in Hilbert spaces." IEEE Transactions on Information Theory 54, no. 2 (2008): 781-790.
+ In section 5.2.1, why do we need a non-parametric method for k-means? isn't k fixed?
+ page 5: the reference for pseudo-dim is not correct
+ please fix the typos through out the paper (including the abstract: a non-parametric coresets..)

**Strengths And Weaknesses:**

Strengths:

+ Overall well-written paper with clear goals
+ Solid technical (theoretical) contributions
+ Addressing a natural question in the clustering literature

Weaknesses:

+ without the knowledge of the previous work it is not easy to say which technical ideas are new
+ theoretical results are not sufficiently contrasted with previous work
+ the presentation of the paper can be improved (see below suggestions)

---

> ### Author Response · Authors · 2022-06-07
> **Response to Reviewer CfBV**
>
> We thank the reviewer for the detailed review and feedback.
>
> Q1. Adding an outline of different sections can improve the readability a lot.
>
> A1. Thank you for the suggestion. We will add an outline section.
>
> Q2. The size of corsets in your main theorems depend on a number of factors such as log[f(A)] and log[n]. Do we know if these dependencies necessary? Can you add a discussion? [there is some discussion in section 6; perhaps you can link to those]
>
> A2. We will add more intuition why a $\log(f(A))$ and $\log(n)$ terms come into our analysis. Here we give the intuition for the $\log(f(A))$ term with a simple example. Recall that our online algorithm takes an irrevocable decision for each point $a_i$ without knowledge about future points. Consider the following input stream — every $i$ that is a multiple of $\sqrt{n}$ has the property that $f(a_i) = f(A_{i-1})$, i.e. this point’s contribution to the current cost is more than the total contributions of all the previous input points. For all other $j, f(a_j)$ is, say, small. Any online algorithm (i.e. one that makes irrevocable decisions without looking at the future) will need to assign a constant probability to every $i$ that is a multiple of $\sqrt{n}$. Hence the resulting coreset size is $\Omega(\sqrt{n})$. Note that $f(A) = O(c^\sqrt{n})$, and hence $\log(f(A))$ is a tight bound on the coreset size in this example.
>
> Q3. It is mentioned in the related work that there are online corsets for k-means in the literature. Can you compare your (theoretical) results with those in more details? What is their coreset size? In what parameter regimes you get an improvement? Perhaps you can compare with them in details after stating your own theorems?
>
> A3. The online algorithms mentioned in the related work do not create coresets. We would like to clarify that these online algorithms give a bicriteria approximation to the  k-means clustering problems and have an update time of $O(dk)$. On the other hand, we built a online coreset for the problem in $O(d)$ update time. Further it is important to note that our online coreset algorithm can be considered in a pipeline where the output coreset can be fed to the online k-means clustering algorithm (mentioned in the related work). This will improve the overall amortized update time from $O(kd)$ to $O(d)$. Please see discussion of this in the reply to Reviewer QNv9.
>
> Q4. The discussion of the uniform deviation bound in section 6 relies on an upper bound. But it is not clear if that upper bound itself is tight and therefore it is not clear if it shows that the dependence on f[A] is necessary. In the absence of some kind of lower bound, the claims should be somewhat weakened.
>
> A4. This is a very good point. We believe it is beyond the scope of the current paper. We will modify the claims accordingly. We also plan to explore in this direction for future work.
>
> Q5. Was there a reason that in section 6 you didn't use classical deviation bounds such as those in this? Biau, Gérard, Luc Devroye, and Gábor Lugosi. "On the performance of clustering in Hilbert spaces." IEEE Transactions on Information Theory 54, no. 2 (2008): 781-790.
>
> A5. Note that both the suggested paper (1) “On the performance of clustering in Hilbert spaces” as well as the paper that we used (2) “Uniform deviation bounds for k-means clustering” talk about clustering in Hilbert spaces. Since Bregman divergences do not form Hilbert spaces, we need to extend the guarantees in our setting.
>
> The guarantee in the paper (1) shows that the difference between the optimal empirical loss and the optimal clustering loss (i.e., the expected excess risk) can be bounded in terms of the radius of the input data. However, the result in our section 6, is in the line of strong coreset guarantee, which requires a bound on the difference between empirical loss and clustering loss for all possible queries. As the paper (2) shows in section 3, such a guarantee is possible only with the combination of relative and additive error terms. In (2) the results were proven only for k-means clustering under \ell_2^2 distance. Hence we chose to extend the analysis in (2) to \mu-similar Bregman divergences. This is what is described in section 6.
>
> We will add this discussion and a corresponding citation to the draft.
>
> Q6. In section 5.2.1, why do we need a non-parametric method for k-means? isn't k fixed?
>
> A6. We agree that the section name k-means is misleading for a non-parametric clustering. We will change it to non-parametric clustering via squared Euclidean distance.
>
> Q7. page 5: the reference for pseudo-dim is not correct
>
> A7. Thank you for pointing this out. We will correct its bibliography input.
>
> Q8. please fix the typos through out the paper (including the abstract: a non-parametric coresets..)
>
> A8. We will proofread the complete paper and we will fix all the typos.

---

### Author Response · Authors · 2022-06-10
**General response**

Dear AE and Reviewers,

We thank all for their feedback. We have incorporated all the requested changes and submitted the new pdf.
We will be happy to discuss if you have any other queries.

Thanks
Authors

---

### Decision · Action_Editors · 2022-06-28

**Recommendation:** Accept with minor revision

**Comment:**

The paper presents novel algorithms for finding coresets for clustering in an online setting. The focus is on k-clustering problems with respect to general Bregman divergences and developing coresets with small additive error. This allows for coresets with a size that is independent of k.

Overall a good paper with solid technical contributions. I recommend that the paper be accepted with minor revisions. These include fixing the typos and providing clarifications as requested by the reviewers, but also changes that the authors have agreed to incorporate in the final version; some of the notable ones are as follows:

a) Add an outline of different sections to improve the readability;

b) Discuss how the size of coresets depends on various factors and if such dependencies are necessary. In general, please comment on the tightness of your results;

c) Add further discussion of related work on online coresets for k-means clustering, and how existing work is different from your work here (building on your response to reviewers CfBV and QNv9);

d) Add empirical comparisons for BregFilter with the online clustering algorithm of Liberty, Edo (2016).